# Is Lindblad for me?

**Martino Stefanini†[1]** ⓘ, **Aleksandra A. Ziolkowska‡[1]** ⓘ, **Dmitry Budker [1,6,7,8]** ⓘ,
**Ulrich Poschinger [1], Ferdinand Schmidt-Kaler [1,6], Antoine Browaeys [2], Atac Imamoglu [3],**
**Darrick Chang [4,5] and Jamir Marino [1]** ⓘ

**1** Institute for Physics, Johannes Gutenberg-Universität Mainz, 55099 Mainz, Germany
**2** Université Paris-Saclay, Institut d'Optique Graduate School, CNRS, Laboratoire Charles
Fabry, 91127, Palaiseau, France
**3** Institute for Quantum Electronics, ETH Zurich, CH-8093 Zurich, Switzerland
**4** ICFO — Institut de Ciencies Fotoniques, The Barcelona Institute of Science and Technology,
Castelldefels (Barcelona) 08860, Spain
**5** ICREA — Institució Catalana de Recerca is Estudis Avançats, Barcelona 08015, Spain
**6** Helmholtz-Institut Mainz, D-55128 Mainz, Germany
**7** GSI Helmholtzzentrum für Schwerionenforschung, 55128 Mainz, Germany
**8** Department of Physics, University of California, Berkeley, California 94720, USA

† mstefa@uni-mainz.de ,    ‡ aaz1@st-andrews.ac.uk
These two authors contributed equally.

## Abstract

**The Lindblad master equation is a foundational tool for modeling the dynamics of open quantum systems. As its use has extended far beyond its original domain, the boundaries of its validity have grown opaque. In particular, the rise of new research areas including open quantum many body systems, non-equilibrium condensed matter, and the possibility to test its limits in driven-open quantum simulators, call for a critical revision of its regimes of applicability. In this pedagogical review, we re-examine the folklore surrounding its three standard approximations (Born, Markov, and Rotating Wave Approximation), as we build our narrative by employing a series of examples and case studies accessible to any reader with a solid background on the fundamentals of quantum mechanics. As a synthesis of our work, we offer a checklist that contrasts common lore with refined expectations, offering a practical guideline for assessing the breakdown of the Lindblad framework in the problem at hand.**

# 1   Introduction

This review centers around a critical re-examination of the assumptions and limitations underlying the Lindblad quantum master equation, the standard framework for modeling the Markovian dynamics of quantum systems weakly coupled to their environments. Since its formalization in the 1970s, it has permeated virtually every subfield of quantum physics with ap-

plications encompassing atomic and molecular physics [1] as well as quantum information [2], including attempts to model problems in quantum gravity and high energy physics [3].

However, with its broad use over the past several decades, the boundaries delineating when the Lindblad equation faithfully captures physical dynamics and when it does not have grown blurred. When its validity is questioned, it is customary to invoke the triumvirate of approximations—Born, Markov, and Rotating Wave (or secular)— highlighting only partially their connection with the system at hand. As a result, the field has seen a gradual erosion in the clarity regarding when, where, and how the Lindblad framework breaks down.

This review aims to address this gap. Our central objective is to provide a critical re-assessment of the Lindblad equation's limitations, grounded in concrete, physically motivated examples. We do not seek to replace canonical references [4–7]—on the contrary, we intend to complement them by offering an example-driven perspective that helps clarify when Lindblad dynamics should be trusted. There are both conceptual and practical reasons why this task is timely. First, the advent of open quantum simulators—engineered platforms that allow controlled access to system-environment interactions [8–11] — has shifted the study of open quantum systems from a theoretical exercise to empirical science. These platforms make it possible to experimentally test the assumptions behind Lindblad dynamics and to witness their breakdowns firsthand. Second, and perhaps more importantly, the systems under investigation today are no longer limited to a few degrees of freedom, as is the case in traditional quantum optics. The community is increasingly focusing on open quantum systems with many constituents, where the interplay between strong correlations, driving, and dissipation generates regimes of behavior far from those contemplated by the original Lindblad framework, which was perfectly suited for quantum systems of a few particles coupled to an environment.

Such regimes are not only of interest in atomic, molecular, and optical (AMO) physics but have also emerged in non-equilibrium condensed matter systems [12], driven quantum materials [13], spintronics [14], optomechanics [15], etcetera. In these settings, environmental effects can entangle with many-body dynamics in ways that defy the assumptions of separability, memorylessness, or timescales separation, which underpin the Lindblad framework.

We have therefore organized this review around a series of representative case studies, drawn both from the literature and from our own original contribution, which aim to make the limitations of the Lindblad approach more tangible. These examples are chosen to be broadly accessible across fields and are intended to serve as a practical resource for researchers entering the field, particularly those working with driven-dissipative quantum matter, complex environments, or beyond-Markovian regimes. To support this goal, we also include technical sections that introduce key concepts in a self-contained manner, enabling newcomers to follow the review without requiring them to go back and forth to the canonical references.

We are aware that some of the examples discussed here may be familiar to established researchers trained in AMO theory. This review is not primarily written for them. Rather, our aim is to support the new generation of scientists studying open many-body systems and to offer a critically engaged entry point for researchers from adjacent fields.

## 1.1 Scope and limitations of this review

This review is not intended as a comprehensive survey of Lindblad master equations in their full generality—a task that would be far too vast, given the many contexts in which such equations arise. The Lindblad form appears across a broad spectrum of frameworks, including weak-coupling limit, singular-coupling limit, continuous measurement theory, collisional models, and even in so-called universal Lindblad constructions. Each of these contexts brings its own assumptions, technicalities, and subtleties. Instead, we focus on what is arguably the most common and foundational scenario: an autonomous quantum system (i.e., one without explicit time-dependent driving) weakly coupled to a large thermal or otherwise struc-

tured bath, both governed by time-independent Hamiltonians. In this setting, the joint system–environment evolution is unitary and energy-conserving. This setup is often the starting point for applications of the weak-coupling limit between system and bath, and it leads to the canonical derivation of the Lindblad equation.

Our goal is not to re-derive this result in a textbook fashion, but to re-express the derivation with an eye toward exposing potential pitfalls, misapplications, and limitations—especially in light of the numerous (traditional and more recent) applications of Lindblad across different areas of physics. To keep the presentation focused and accessible, we deliberately do not discuss more advanced or specialized Lindbladian frameworks, such as those involving continuous monitoring or explicit time-dependent driving.

This is not a review of the Lindblad equation, but rather a critical examination of the assumptions and conditions under which one particular and widely used Lindblad form is valid. We emphasize this boundary because it may not be obvious: readers encountering phenomena that appear to fall outside the scope of our discussion should consider whether those phenomena lie beyond the assumptions we have set. Even within the specialized setting we consider, a truly exhaustive treatment would require a dedicated textbook. Thus, the absence of certain examples in this review is hopefully not an oversight, but a reflection of the deliberate scope we have chosen.

## 1.2 Guide to the reader

For a student or researcher new to the subject, this review is a self-contained explanation of the Lindblad equation. Throughout the article, we assume that the reader has a knowledge of the fundamentals of quantum mechanics, including its formulation in the Schrödinger and interaction pictures. Necessary for its understanding is also familiarity with the density matrix formalism and time-independent perturbation theory. Second quantization is used in some parts. Sections marked with an asterisk (*) contain additional discussion on the nuances of the Lindblad equation as well as some finer mathematical details of the formalism, and can be skipped at a first reading.

The article is organized as follows. In section 2 we introduce the Lindblad equation, focusing on the separation of timescales as a gateway to accessing its physical content. Sections 3 - 5 are devoted to each of the three approximations in the derivation of the Lindblad equation. These are, in order, Born, Markov, and Rotating-Wave approximations. Because of the intertwined nature of the Born and Markov approximations, many aspects of the former are covered in section 4, together with the latter.

The breakdown of the Markov approximation is presented by inquiring which properties the bath spectral density should satisfy to permit a description within the Lindblad framework (Sec. 4.3). As a highlight, in Sec. 4.4.1 we introduce a toy model to explain the breakdown of the Markovian approximation during dynamics. The model is based on a simple ordinary differential equation with a memory kernel, which illustrates how non-Markovianity is expected to be a general feature both at short and long times, while at intermediate times a Markovian description becomes feasible. The implications for Lindblad dynamics are discussed thoroughly. As applications, we cover the role of temperature in dictating Markovian vs non-Markovian conditions for environments (Sec. 4.4.4), and present a case of study inspired by photonic crystals (Sec. 4.5.2). The failure of the Born approximation, which is traditionally tied to Markovianity, is presented independently through the example of the Kondo model (Sec. 4.5.1). This is the paradigmatic case of renormalization of system-bath coupling growing quickly into non-perturbative regimes as a result of bath correlations dressing the coupling. The sections on breakdown of RWA (Sec. 5.3) are those mostly relevant for applications (or failures) of the Lindblad description to open quantum many-body systems.

We conclude the review with a practical guide in the form of a table, summarizing the applicability of the Lindblad equation to a problem at hand in section 6.

## 2   What is the Lindblad equation?

The Lindblad quantum master equation reads

$$\frac{\mathrm{d}}{\mathrm{d}t}\rho_S(t) = -\frac{i}{\hbar}[H, \rho_S] + \sum_a \gamma_a \Big[L_a \rho_S L_a^\dagger - \frac{1}{2}\{L_a^\dagger L_a, \rho_S\}\Big]. \tag{1}$$

It defines the time evolution of the reduced density matrix of the system $\rho_S$, in the form of an exponential relaxation towards a stationary state (for more details see the discussion in the appendix A). It comprises the coherent dynamics governed by the system's Hamiltonian $H$—possibly shifted by the coupling to the bath—and the dissipation induced by the environment, encoded via jump operators $L_a$ acting with corresponding positive rates $\gamma_a$. The term "Markovian" refers to the property of (1) that the derivative of $\rho_S(t)$ at a given time $t$ is completely determined by its value at the same time—in other words, the dynamics have no memory. The motivation behind equation (1) is to model the system's dynamics without explicitly computing the evolution of the environment. Indeed, the effect of the environment is fully encapsulated into the jump operators $L_a$ and in the rates $\gamma_a$, as well as in a correction to the system's Hamiltonian. This simplification makes it easier to model and understand dissipation and is beneficial on the computational side since it is much easier to simulate equation (1) than to follow the full state of the system and environment.

The form of equation (1) is dictated both by general properties of quantum mechanics and by the requirement of Markovianity. In general, the time evolution of the density matrix of an open quantum system is described by a map $\Lambda_{t,0} : \rho_S(0) \to \rho_S(t)$, and the postulates of quantum mechanics constrain it to be CPT: Completely Positive and Trace-preserving linear map [2, 16]. These properties express the requirement that $\Lambda_{t,0}$ has to map a physical state $\rho_S(0)$ to a physical state $\rho_S(t)$, hence it must preserve the trace $\mathrm{Tr}\,\rho_S(0) = \mathrm{Tr}\,\rho_S(t) = 1$ and yield a positive semidefinite matrix $\rho_S(t)$ (i.e. whose eigenvalues are positive or zero). The "complete" positivity further requires that the dynamics yield a physical state also in the presence of an additional, arbitrary inert system (ancilla), which is possibly initially entangled with the physical system. On this basis, the Lindblad equation was originally derived by Gorini, Kossakowski, Sudarshan [17], and Lindblad [18] as the most general[1] CPT map that is also Markovian, in the sense mentioned before.

Since the dynamics generated by equation (1) is guaranteed to respect the physicality of the state $\rho_S(t)$, the Lindblad equation is often used in a phenomenological fashion: one guesses the form of the jump operators $L_a$ on the basis of the kind of processes that the environment is expected to induce, while the rates $\gamma_a$ are parameters to be fitted to experiments. However, the mathematical derivation of the master equation from the requirements of CPT property and Markovianity does not elucidate its microscopic origins. Indeed, the full dynamics of the system and environment are usually described by a Hamiltonian[2], and a natural question is under which circumstances the dynamics of the system alone are well-described by equation (1). This question is relevant for many practical applications in which one would like to identify

---

[1]Strictly speaking, this statement has been formally proven only if $\sum_a L_a^\dagger L_a$ is a bounded operator, and for time-independent coefficients.

[2]An exception is provided by an otherwise isolated system that is continuously undergoing (weak) measurements. It turns out that the state of the system, averaged over all measurement results, obeys equation (1) with $L_a$ being the operators corresponding to observables that are being monitored [2,16]. While the topic of measurement-induced dynamics is fascinating on its own, it is outside the scope of this review.

the appropriate jump operators and decay rates from a known system-environment Hamiltonian. Many authors have addressed this problem [17–23], and it is well established that a number of approximations and assumptions are needed when deriving equation (1) from underlying unitary dynamics. However, the regimes of validity of these approximations are often not universally clear to researchers, and the purpose of this review is precisely to offer a short and self-contained account of the limits in which the Lindblad equation can be applied. For open dynamics to warrant a Lindblad treatment, the properties of the system, bath, and their interaction must all be considered. Their physical properties find their equivalence in the mathematical approximations carried out to arrive at the Lindblad equation. We will examine the relevant physical requirements in detail throughout this review. Here, we provide a sketch of the properties of interest. We will elaborate on those in detail in the following sections of this review.

The strength of the interaction, roughly defined by the coefficients $\gamma$ in equation (1), needs to be weak in order to apply the *Born approximation*. More precisely, if $\lambda$ is the coupling constant and $\tau_B$ the decay time of the bath, the small parameter of the theory is $\lambda \tau_B / \hbar \ll 1$ and the typical decay rate is $\gamma \sim \lambda^2 \tau_B / \hbar$. This condition ensures the validity of the perturbation theory on which the Lindblad framework is based. Consequently, the timescale for the system relaxation $\tau_R \sim \gamma^{-1}$ is the longest timescale in the problem. Notice that the assumption of weak coupling is naturally needed to separate the system from its environment.

The *Markovian approximation* implies that the environment has fast-decaying correlation functions and quickly "forgets" the information acquired due to the interaction with the system, so that there is no possibility of back-action—namely, one excludes that the bath modifications induced by the system might affect the system at later times. In other words, the bath should be essentially a good thermodynamic bath, large enough to act on the system without being significantly affected by it. In more detail, the correlation functions of the bath operator $B$ coupled to the system, $\langle B(t)B(0)\rangle$, define another timescale, $\tau_B$, over which they decay to zero. For the system dynamics to be Markovian, the relaxation of the system should not be sensitive to the internal bath dynamics and $\tau_B \ll \tau_R$. The requirement that $\langle B(t)B(0)\rangle$ decays to zero implies that the bath has to be thermodynamically large in the sense that its spectrum should be sufficiently dense (and ideally continuous)[3].

Lastly, the system's spectrum calls for scrutiny. The interaction with the bath induces a certain set of transitions between the system's eigenstates, with transition frequencies $\Omega$ (these are the unperturbed transition frequencies of the system). Their differences, $\Delta\Omega$, define yet another timescale[4], $\tau_S \sim (\Delta\Omega)^{-1}$. Consistently with the requirement that the dissipation is a perturbation to the system rather than the dominant process, $\Delta\Omega$ is a larger energy scale than the one defined by the dissipation strength $\gamma$. Hence, the opposite relation holds for the corresponding timescales, and $\tau_S \ll \tau_R$ - the system's dynamics must be much faster than the relaxation rate[5]. One is then justified in applying the *rotating-wave approximation*, also known as the *secular approximation*. This approximation amounts to neglecting the processes in which the bath induces simultaneously different transitions with a large frequency mismatch $\Delta\Omega \gg \tau_R^{-1}$. This step guarantees the positivity of the time evolution, namely that all rates

---

[3]In a finite system, $\langle B(t)B(0)\rangle$ generically exhibits recurrences, i.e., comes back close to its initial value $\langle B(0)B(0)\rangle$ at certain times $T_R^{(1)} < T_R^{(2)} < \ldots$ which usually grow together with the bath size. Then, Markovianity requires that the minimal recurrence time $T_R^{(1)}$ should be much larger than the decay time $\tau_R$.

[4]More rigorously, every pair of transitions defines a timescale for a system dynamics. Each of these, in turn, should then be compared against the *corresponding* timescale for dissipation, i.e., we require a separation of timescale between each dissipative process, and the pairs of transitions it is linking. As we will show later, there is no need to consider transitions between degenerate levels with $\Delta\Omega = 0$ here, since they do not require any further approximation.

[5]Although the weak dissipation is necessary for the rotating-wave approximation, it is not a sufficient condition. We discuss this point in detail in section 5.2

$\gamma_a$ are positive. It is a somewhat technical assumption, and its necessity has been debated extensively [24–27].

The hierarchy of timescales relevant in the standard derivation of the Lindblad equation is visualized in Figure 1. The notation follows the convention introduced in Ref. [5]. On the basis of the approximations sketched, one can expect that the Lindblad equation models well the dynamics of a few-body system with discrete energy levels coupled to a large, unstructured[6] bath. A paradigmatic physical example of such a problem is that of the optical transitions of an atom in an electromagnetic field.

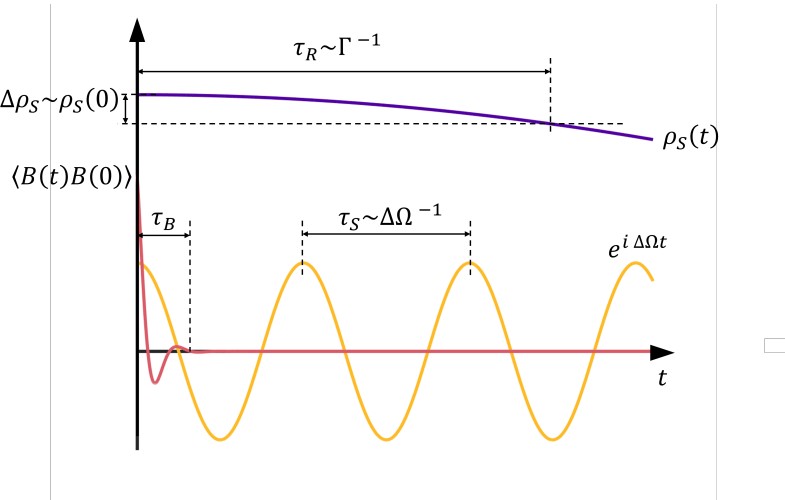

Figure 1: Comparison of the relevant timescales in the derivation of the Lindblad equation. $\tau_B$ is the decay time of the bath correlation functions $\langle B(t)B(0)\rangle$, $\tau_S$ the timescale of the internal system dynamics inversely proportional to the difference in the system's transitions $\Delta\Omega$, and $\tau_R$ is the relaxation time of the system over which its state changes appreciably due to dissipation. $\Gamma$ represents a typical relaxation rate of the system (see equation (13)), which is directly related to the rates $\gamma_a$ in equation (1).

## 3   The Born Approximation

- The Born approximation relies on the assumption of a weak system-bath coupling to neglect the effect of their entanglement on the dynamics of the system, i.e., consider the total density matrix separable within the master equation [see equation (6)].

- If $\lambda$ is the coupling and $\tau_B$ is the typical decay time of the bath correlation functions [see equation (7)], the approximation is valid as long as $\lambda\tau_B/\hbar \ll 1$. Hence, strong coupling or long bath correlation times invalidate the approximation.

### 3.1   General Discussion

Consider a system in contact with a bath represented by a combined density matrix $\rho(t)$. The dynamics are described by a system Hamiltonian $H_S$, bath Hamiltonian $H_B$, and the system-

---

[6]The term "unstructured" refers more specifically to its density of states, that should not have sharp features like narrow resonances close to the system's transition energies. This point will be discussed in section 4.

bath interaction of strength $\lambda$ is governed by $H_I = \sum_\alpha A_\alpha B_\alpha$, where the operators $A$ and $B$ act only on the Hilbert spaces of the system and bath, respectively. For the simplicity of notation, we assume that the operators $A_\alpha$ and $B_\alpha$ are Hermitian. This does not make the derivation any less general, as any form of coupling can be represented as a linear combination of Hermitian operators. The full system-bath Hamiltonian under consideration is

$$H = H_S + H_B + \lambda H_I \,. \tag{2}$$

To begin with, we assume that the initial state is separable $\rho(0) = \rho_S(0) \otimes \rho_B(0)$. This condition implies that, at the start of evolution, there are no pre-existing correlations between the system and the bath, which is usually a good approximation in experiments. Then, all correlations will be a consequence of system-bath interactions. For simplicity, we also assume that the initial state of the bath is stationary with respect to its own Hamiltonian, i.e., $[H_B, \rho_B(0)] = 0$. For the clarity of notation, we will use $\rho_B(0) = \rho_B$ in the following. This requirement is not strictly needed for deriving the Lindblad equation, but it makes the derivation clearer, and it is often met in practical applications. For instance, a common case is that of a bath that is initially at thermal equilibrium at a temperature $T$, $\rho_B \propto e^{-\beta H_B}$, where $\beta = (k_B T)^{-1}$. It is convenient[7] to go to the interaction picture, in which density matrices evolve according to $\rho(t) = e^{-\frac{i}{\hbar} H_0 t} \rho_0(t) e^{\frac{i}{\hbar} H_0 t}$ while operators evolve with $O(t) = e^{\frac{i}{\hbar} H_0 t} O e^{-\frac{i}{\hbar} H_0 t}$, where $H_0 = H_S + H_B$ and $\rho_0(t)$ is the state in the Schrödinger picture. Then, the time evolution of the total density matrix in the interaction picture follows the equation

$$\frac{\mathrm{d}}{\mathrm{d}t} \rho(t) = -\frac{i}{\hbar} \lambda [H_I(t), \rho(t)] \,. \tag{3}$$

The main strategy will be to make a clever approximation of the dynamics up to second order in $\lambda$. It is easier to do so if the equation for $\rho(t)$ is explicitly of the second order, so we integrate the above equation as $\rho(t) = \rho(0) - \frac{i}{\hbar} \lambda \int_0^t \mathrm{d}s\, [H_I(s), \rho(s)]$ and we substitute it on the right-hand side of the equation (3) to obtain

$$\frac{\mathrm{d}}{\mathrm{d}t} \rho(t) = -\frac{i}{\hbar} \lambda [H_I(t), \rho(0)] - \frac{\lambda^2}{\hbar^2} \int_0^t \mathrm{d}s\, [H_I(t), [H_I(s), \rho(s)]] \,. \tag{4}$$

In order to arrive at the description of the reduced dynamics of the system, the bath degrees of freedom need to be traced out: we seek an equation that involves only $\rho_S(t) \equiv \mathrm{Tr}_B \rho(t)$. The first term on the right-hand side of equation (4) vanishes under the trace, as, for the stationary bath, the one-point correlations are zero and $\mathrm{tr}_B[B_\alpha(s)\rho_B] = 0$.[8] To take the partial trace of the second term on the right-hand side of equation (4), we need to consider the conditions for separability of the density matrix. While we assumed that the initial state is separable, $\rho(0) = \rho_S(0) \otimes \rho_B$, the system and environment interactions generally induce correlations— $\rho(t)$ is generally not factorized at $t > 0$. However, the corrections to the separable form of the density matrix can be at most of the same order as the interaction term in the time evolution. Consequently, we can write

$$\rho(t) = \rho_S^{(0)}(t) \otimes \rho_B + \lambda^2 \rho_{\mathrm{corr}}(t) \,, \tag{5}$$

---

[7]As it will become clear in the next section, the interaction picture is actually crucial in the approximations that follow because it allows the separation of the time scales of the dynamics. While in the Schrödinger picture $\rho_0(t)$ evolves on the "fast" timescales of $H_0$, the interaction picture $\rho(t)$ is "slow" as its evolution is mostly determined by the interactions. Indeed, $\mathrm{d}\rho(t)/\mathrm{d}t \propto \lambda^2$.

[8]For a descriptive explanation see [28]. In general, if $\langle B_\alpha \rangle_B \equiv \mathrm{tr}_B[B_\alpha(s)\rho_B] \neq 0$, it can always be made to vanish by shifting $B_\alpha \to B_\alpha - \lambda \langle B_\alpha \rangle_B$ and including $\lambda \sum_\alpha \langle B_\alpha \rangle_B A_\alpha$ as a "driving" term in $H_S$. The physical intuition behind this mathematical trick is that we want to separate the "classical" field $\langle B_\alpha \rangle_B$, that would just contribute to the unitary dynamics of the system, from the actual dissipation which, as we will see, comes from the "noise" of the bath—namely, the correlation functions of the $B_\alpha$ operators.

where $\rho_S^{(0)}(t)$ is the leading-order system density matrix contribution, $\rho_B$ is the *initial* bath density matrix[9], and $\rho_{\text{corr}}(t)$ is a non-separable contribution to $\rho(t)$. Here, we assume that the coupling strength $\lambda$ is weak; hence, we can use it to perform a perturbative expansion[10]. The reduced density matrix of the system is then $\rho_S(t) = \rho_S^{(0)}(t) + \lambda^2 \operatorname{tr}_B \rho_{\text{corr}}(t) \approx \rho_S^{(0)}(t)$. If we substitute equation (5) into the right-hand side of equation (4), we see that the contribution of $\rho_{\text{corr}}(t)$ to $d\rho(t)/dt$ is of order $\lambda^4$ and therefore negligible with respect to the second-order contribution from the factorized term. Therefore, we keep only the latter; this is known as the *Born approximation*[11],

$$
\begin{aligned}
\frac{\mathrm{d}}{\mathrm{d}t}\rho_S(t) &= -\frac{\lambda^2}{\hbar^2}\int_0^t \mathrm{d}s\,\operatorname{tr}_B\left[H_I(t),[H_I(s),\rho(s)]\right] \\
&\approx -\frac{\lambda^2}{\hbar^2}\int_0^t \mathrm{d}s\,\operatorname{tr}_B\left[H_I(t),[H_I(s),\rho_S(s)\otimes\rho_B]\right].
\end{aligned}
\tag{6}
$$

The Born approximation is usually phrased as $\rho(t) \approx \rho_S(t)\otimes\rho_B$, similar to a mean-field Ansatz. We emphasize that this writing is meaningful only on the right-hand side of the equation (4). In fact, $\rho_S(t)\otimes\rho_B$ is generally a poor approximation to $\rho(t)$—for instance, see Ref. [30] for a comparison of the two. The heart of the Born approximation is that while the system-bath correlations grow in time, their contribution to the dynamics of the system remains subleading. Then, the system dynamics can be modeled *as if* the total density matrix is factorizable. An important example where the Lindblad equation allows for entanglement between system and bath is in the generation of remote entanglement [31]. Here, an atom can emit a photon and produce an atom-photon entangled state. The photon can interact with a distant second atom and generate remote entanglement between two atoms. Despite the onset of entanglement, a Lindblad equation perfectly well describes the dynamical process.

We also remark that the Born approximation goes beyond simple perturbation theory in the sense that we are not simply expanding $\rho(t)$ in powers of $\lambda$. Such an approach would yield an unstable (secular) dynamics. The Born approximation is a self-consistent approximation, in the sense that it approximates $d\rho_S(t)/dt$ in terms of the interacting, time-evolving system density matrix $\rho_S(t)$ itself. This approach implicitly retains all powers of $\lambda$ in $\rho_S(t)$—in other words, $\rho_S^{(0)}(t)$ in equation (5) contains corrections of order higher than $\lambda^2$. Still, this approximation is perturbative because the decay rates $\gamma_a$ will be obtained to order $\lambda^2$ only. A clarification of the difference between simple and self-consistent perturbation theory can be found in the literature, see for instance [32].

The Born master equation is still non-Markovian, as the time evolution of the density matrix depends on its history. Nevertheless, some of the assumptions made to arrive at this equation go hand in hand with the Markov approximation, which we introduce in section 4.

The key elements that allow us to examine the convergence of the perturbative expansion, and thus the validity of the Born approximation, are the correlation functions of the environment. They play the role of coefficients of different dissipative channels within the master

---

[9]If $[\rho_B(0), H_B] \neq 0$, one should use $\rho_S^{(0)}(t) \otimes \rho_B(t)$ as the separable part, where $\rho_B(t) = e^{-\frac{i}{\hbar}H_B t}\rho_B(0)e^{\frac{i}{\hbar}H_B t}$ is the bath's density matrix evolving in the absence of the system. The rationale behind this decomposition is that we do not want to track the effect of the system on the bath, and it is justified by the fact that the latter is higher order in $\lambda$.

[10]The actual perturbative parameter should include the norm of the bath coupling operators $\|B\|$. Moreover, a careful reader will realize that throughout this section, we assumed $\lambda$ to have a dimension of energy, and thus it is not a proper perturbation parameter. A rigorous restatement of equation (5) would involve an expansion in $\tilde{\lambda} = \lambda\tau_B\|B\|/\hbar$.

[11]The formal justification of the approximate factorizability of $\rho(t)$ is quite subtle: in [29], it is shown that if the bath is made of many independent components (i.e., it is many body), it can be related to a hierarchy within its n-point correlation functions.

equation. Substituting in the explicit form of the interaction Hamiltonian into equation (6), we get

$$\frac{\mathrm{d}}{\mathrm{d}t}\rho_S(t) = -\frac{\lambda^2}{\hbar^2}\sum_{\alpha\beta}\int_0^t \mathrm{d}s \left\langle B_\alpha(t)B_\beta(s)\right\rangle \left[A_\alpha(t)A_\beta(s)\rho_S(s) - A_\beta(s)\rho_S(s)A_\alpha(t)\right] + \mathrm{H.c.}\,, \quad (7)$$

where $\left\langle B_\alpha(t)B_\beta(s)\right\rangle = \mathrm{tr}_B\left[B_\alpha(t)B_\beta(s)\rho_B\right]$. By construction, the bath correlation functions decay rapidly for $t - s > \tau_B$ [12], providing an effective cut-off for the integral, in the sense that $\int_0^t \mathrm{d}s\cdots \approx \int_{t-\tau_B}^t \mathrm{d}s\dots$. If the typical timescale of the system evolution, $\tau_R$, is much slower than $\tau_B$, the state of the system can be considered constant over the decay time of the bath correlation functions. Consequently, we can set $\rho_S(s) \to \rho_S(0)$ for the integrated terms. As a result, we can estimate the magnitude of the integral term in equation (6) to be $\sim \lambda^2 \tau_B/\hbar^2$. The higher-order corrections to the dynamics involve an increasing number of time integrals. These can be examined analogously to the second-order correction to find the scaling $\sim (\lambda/\hbar)(\lambda\tau_B/\hbar)^{n-1}$, where $n$ is the order of the correction. The perturbative series converges if $\lambda\tau_B/\hbar \ll 1$, which is consistent with the weak coupling assumption.

The second-order correction, by definition, also indicates the relaxation rate of the system $\tau_R^{-1} = \lambda^2 \tau_B//\hbar^2$. Combining this condition with the requirement of convergence of the perturbative expansion, we see that the validity of the perturbation theory is synonymous with the condition for the fast bath relaxation $\tau_B \ll \tau_R$ also required for the memory loss characterizing the Markovian dynamics. The Born master equation (6) requires the knowledge of the density matrix at every time step. It is computationally costly, which can be remedied by applying the Markov approximation, the subject of section 4.

## 4   The Markov Approximation

- The Markov approximation uses the short bath correlation time $\tau_B$ (with respect to the decay time of the system $\tau_R$) to neglect the "memory" of the bath, replacing the integro-differential Born master equation (7) with the simpler Redfield master equation (11).

- The system probes the bath spectral function $J(\omega)$ (section 4.3) around the energies of the transitions induced by the bath. The Markov approximation is valid if $J(\omega)$ does not have sharp features in this region—see Fig. 2.

- The Markov approximation is valid for times larger than $\tau_B$ until a crossover time, which is usually much larger than $\tau_R$. Therefore, in most physical systems, the Markovian behavior appears only for a finite window of evolution.

### 4.1   Properties of the Markovian approximation

Markovian dynamics refers in both classical and quantum mechanics to a time evolution independent of the history of interactions. The discussion on what exactly constitutes a Markovian system is complicated by a lack of agreement on how to quantify quantum Markovianity [33–37]. Still, for practical purposes, if the time-evolution generator, such as an experimental apparatus or a quantum circuit, has no memory of the initial conditions of the evolution, we can treat it as Markovian. More precisely, we can write that for a Markovian map

---

[12]This is the "fast decay of the bath" requirement, implying that any correlations separated temporally by more than $\tau_B$ are (essentially) zero.

$\Lambda_{t,t'} : \rho(t') \rightarrow \rho(t)$ evolving a system between the two time arguments, we have a decomposition

$$\Lambda_{t,t'} = \Lambda_{t,s} \circ \Lambda_{s,t'} \quad \text{for} \quad t \leq s \leq t' \, . \tag{8}$$

This is a semigroup property [38], which indicates that Markovian dynamics has no notion of absolute time—only time-evolution intervals.

Although conceptually Markovianity seems straightforward, this idea is not apparent mathematically, and multiple implementation schemes have been suggested in the literature [17, 19, 20]. The nuance here is the different order of the interaction strength and the intrinsic system dynamics. It is not immediately clear which energy scales are subleading in a Markovian equation, and their separation can lead to quantitatively different results. In particular, applying the Markov approximation to the differential form of the Born master equation (6) or its integrated version yields different time-evolution generators [22]. Fortunately, the differences between mathematically sound implementations of the Markov approximation are quantitatively small and introduce an error of the order of the coupling strength $\lambda^2$. Some Markovian master equations reduce to the same standard Lindblad form after applying RWA [39]. In this review work, we will concern ourselves with the common form of the Markov approximation, which involves the most physically intuitive steps [5].

In order to make progress with the Born master equation (7) we need to know how the coupling operators $A_\alpha$ evolve in time. Introducing the eigenstates of $H_S$, $|n\rangle$, with eigenenergies $\varepsilon_n$, we can use $e^{-iH_S t/\hbar} = \sum_n e^{-i\varepsilon_n t/\hbar} |n\rangle\langle n|$ and write the Heisenberg evolution of the operator $A_\alpha$ under $H_S$ as $A_\alpha(t) = e^{iH_S t/\hbar} A_\alpha e^{-iH_S t/\hbar} = \sum_{m,n} e^{-i(\varepsilon_n - \varepsilon_m)t/\hbar} |m\rangle\langle m| A_\alpha |n\rangle\langle n|$. A little rearrangement of this decomposition shows that it is of the form $A_\alpha(t) = \sum_\Omega e^{-i\Omega t} \mathcal{A}_\alpha(\Omega)$, where the operators $\mathcal{A}_\alpha(\Omega)$ are known as eigenoperators [5, 22], and are associated with all the possible "Bohr frequencies" (i.e., frequencies of transitions) $\Omega = (\varepsilon_n - \varepsilon_m)/\hbar$ of the system. The eigenoperators are given explicitly by expression

$$\mathcal{A}_\alpha(\Omega) \equiv \sum_{m,n} \delta_{\varepsilon_n - \varepsilon_m, \hbar\Omega} |m\rangle\langle m| A_\alpha |n\rangle\langle n| = \sum_\varepsilon \Pi(\varepsilon - \hbar\Omega) A_\alpha \Pi(\varepsilon) \, , \tag{9}$$

where we have introduced the projector $\Pi(\varepsilon) = \sum_n |n\rangle\langle n| \delta_{\varepsilon, \varepsilon_n}$ on the subspace[13] of eigenstates with energy $\varepsilon$. From the above equation, it can be verified that operators $\mathcal{A}_\alpha(\Omega)$ have several convenient properties. First, they obey an "eigenvalue" equation $[H_S, \mathcal{A}_\alpha(\Omega)] = -\hbar\Omega\mathcal{A}_\alpha(\Omega)$ (whence the name eigenoperators), which guarantees that $\mathcal{A}_\alpha(\Omega)$ evolves with a simple phase under $H_S$, $\mathcal{A}_\alpha(\Omega, t) = \mathcal{A}_\alpha(\Omega) e^{-i\Omega t}$. Second, if $A_\alpha$ is Hermitian, $\mathcal{A}_\alpha^\dagger(\Omega) = \mathcal{A}_\alpha(-\Omega)$. Physically, these two properties indicate that the operators $\mathcal{A}_\alpha(\Omega)$ $(\mathcal{A}_\alpha^\dagger(\Omega))$ behave like annihilation (creation) operators, in the sense that their application on a state reduces (increases) its energy by an amount $\hbar\Omega$. Finally, any operator $A_\alpha$ can always be decomposed in terms of the corresponding eigenoperators, $A_\alpha = \sum_\Omega \mathcal{A}_\alpha(\Omega) = \sum_\Omega \mathcal{A}_\alpha^\dagger(\Omega)$. At a practical level, the eigenoperators can be found either by direct application of equation (9) or by computing the Heisenberg time evolution of $A_\alpha$ and singling out the coefficients of the $e^{-i\Omega t}$ terms. We are going to further comment on the eigenoperators in section 4.2.

Rewriting $A_\alpha$ in terms of the eigenoperators and substituting it into the equation (7), we get

$$\frac{\mathrm{d}}{\mathrm{d}t}\rho_S(t) = -\frac{\lambda^2}{\hbar^2} \sum_{\Omega,\Omega'} \sum_{\alpha\beta} \int_0^t \mathrm{d}s \left\langle B_\alpha(t)B_\beta(s) \right\rangle e^{i\Omega'(t-s)} e^{-i(\Omega' - \Omega)t}$$

$$\left[ \mathcal{A}_\alpha^\dagger(\Omega)\mathcal{A}_\beta(\Omega')\rho_S(s) - \mathcal{A}_\beta(\Omega')\rho_S(s)\mathcal{A}_\alpha^\dagger(\Omega) \right] + \text{H.c.} \, . \tag{10}$$

Equation (10) makes it apparent that the actual objects governing the perturbation theory in the Born master equation are the factors $\left\langle B_\alpha(t)B_\beta(s) \right\rangle e^{i\Omega'(t-s)}$. Their magnitude is limited by

---

[13]In the absence of degeneracies, $\Pi(\varepsilon_n) = |n\rangle\langle n|$ become simple projectors.

two independent timescales. One of them is the typical decay time of the bath correlation functions $\tau_B$, which makes $\left\langle B_\alpha(t)B_\beta(s)\right\rangle$ highly localized in time. The other is a timescale[14] arising from the system's transition frequencies, $\tau_H = 1/\Omega'$ [40]. This timescale defines the period of coherent oscillations between two levels separated by an energy $\hbar\Omega'$ in the unperturbed system. The exponential factor $e^{i(t-s)/\tau_H}$ contributes to the convergence of the integral on the right-hand side of equation (10) by averaging out to zero the contributions coming from times $t-s \gg \tau_H$. Hence, the smaller the $\tau_H$, the shorter the support of the time-integral, which is not suppressed by the oscillations. Alternatively, one can consider $\tau_H$ in the context of perturbation theory—the larger the energy scales of the unperturbed system dynamics $\Omega'$, the more of an actual perturbation is the dissipative coupling. As a result, the actual magnitude of the dissipative term is indicated by the smallest between the two timescales $\tau_B$ and $\tau_H$. The existence of the latter timescale is often overlooked, but it is important in cases in which the bath correlation functions $\left\langle B_\alpha(t)B_\beta(s)\right\rangle$ decay slowly in time $t-s$, yielding a large $\tau_B$. We are going to comment more on these cases in 4.3.1.

The essentially non-Markovian element of the Born master equation (10) is that the time-evolution of the density matrix at time $t$ depends on its state at previous times, $s < t$. Crucial for the physical justification of the Markovian approximation and its subsequent mathematical implementation are the assumptions made on the bath correlation functions. As discussed in section 3.1, $\left\langle B_\alpha(t)B_\beta(s)\right\rangle$ decays rapidly for $t - s > \tau_B$. The bath correction time $\tau_B$ is very short, so the correlation functions are strongly peaked around time $t$, acting almost like a delta function and "picking out" the terms under the integral close to time $t$. Hence, the actual contribution from the density matrix is from $\rho(s) \approx \rho(t)$. This approximation can be refined for large system transition frequencies $\Omega$, which may further narrow the integration support due to fast oscillations beyond the timescale $\tau_H$. Substituting $\rho_S(s) \to \rho_S(t)$ into the Born master equation (10) we get

$$\frac{d}{dt}\rho_S(t) = -\sum_{\Omega,\Omega'}\sum_{\alpha\beta}\Gamma_{\alpha\beta}(\Omega',t)e^{-i(\Omega'-\Omega)t}\left[\mathcal{A}_\alpha^\dagger(\Omega)\mathcal{A}_\beta(\Omega')\rho_S(t) - \mathcal{A}_\beta(\Omega')\rho_S(t)\mathcal{A}_\alpha^\dagger(\Omega)\right] + \text{H.c.} ,$$

$$(11)$$

with a time-dependent coefficient matrix

$$\Gamma_{\alpha\beta}(\Omega',t) = \frac{\lambda^2}{\hbar^2}\int_0^t ds \left\langle B_\alpha(t)B_\beta(s)\right\rangle e^{i\Omega'(t-s)} . \tag{12}$$

To arrive at a fully Markovian master equation, we need to perform another approximation. We rely again on the fact that the bath correlation functions decay rapidly outside of short support of size $\min(\tau_B, \tau_H)$. Hence, the limit of the integral in the expression for the coefficient matrix (12) can be extended to infinity without altering any physical properties. Furthermore, for a stationary bath, the correlation functions are time-translationally invariant, so they depend only on the difference of time arguments, and we have $\left\langle B_\alpha(t)B_\beta(s)\right\rangle = \left\langle B_\alpha(t-s)B_\beta(0)\right\rangle$. Using a substitution $\tau = t-s$, the coefficient matrix becomes time-independent

$$\Gamma_{\alpha\beta}(\Omega') = \frac{\lambda^2}{\hbar^2}\int_0^\infty d\tau \left\langle B_\alpha(\tau)B_\beta(0)\right\rangle e^{i\Omega'\tau} . \tag{13}$$

Using the time-independent coefficients in equation (11) gives a fully Markovian master equation called the Redfield equation. Although it encodes Markovian dynamics, it is not a CPT map, as discussed further below in sections 4.1.1 and 5.

---

[14]If we take $\hbar\Omega'$ to be the average level spacing of the system spectrum, then $\tau_H$ is the Heisenberg time. Note that $\tau_H$ is distinct from the timescale $\tau_S \sim 1/(\Omega' - \Omega)$ that governs the RWA, although the two are related. Let us illustrate the difference between these timescales on an example of a two-level system with a Hamiltonian $H = \frac{\Delta}{2}\sigma^z$, coupled to a bath via jump operators proportional to $\sigma^\pm$. Then, we have $\tau_H = \hbar/\Delta$ and $\tau_S = \hbar/2\Delta$.

### 4.1.1 The question of positivity *

By considering only general properties of dynamical maps, it was shown that the most general Markovian CPT map with time-independent coefficients is of the Lindblad form [17, 18]. Hence, one could have expected that imposing the Markovian condition at the microscopic level would automatically generate a positivity-preserving map. This is not the case, and an additional step in the form of the RWA is needed. This raises questions about the standard implementations of the Markovian approximation in open systems and gives rise to an on-going search for a microscopic derivation that does not require RWA to achieve the Lindblad form, such as the Universal Lindblad Equation (ULE) [27, 41] or the Unified Lindblad master equation [39].

In the standard derivation presented in this work, we break positivity for the first time already in the truncation of the perturbation theory justified by the Born approximation—the first line of equation (6) is exact and positivity preserving, whereas the second violates positivity. Yet, positivity breaking is not an inherent feature of the Born approximation—a common formulation of the Born master equation based on the cumulant expansion is completely positive [28, 42]. The positivity is only necessarily broken when implementing the Markov approximation (unless the bath spectrum has an infinite width and infinite temperature, i.e., it is perfectly Markovian on any timescales). The positivity problem arises from different orders of magnitude of the coherent and incoherent dynamics and our truncation of the small but finite non-Markovian contributions at the second order in perturbation theory[15]. This results in a departure from the rigorously perturbative approximation—at the cost of physically motivated simplification of the master equation, we made an uncontrolled truncation in the perturbative sense. Physically, although the Markov approximation does not discard any physical processes, it changes their relative contribution to the dynamics and may render states with unphysical, negative probabilities [44]. As we will discuss in section 5.1, positivity is restored by applying RWA, which is a standard way of arriving at the Lindblad equation.

### 4.1.2 Singular coupling limit *

It is worth noting that, in exceptional cases, the Lindblad equation can be reached directly in the so-called *singular coupling limit* [45, 46]. It applies only in a highly restricted situation when the system-bath Hamiltonian has a structure

$$H = H_S + \lambda^2 H_B + \lambda H_I \,, \tag{14}$$

and allows for the computation of the master equation in the strong coupling $\lambda \to \infty$ and high-temperature $T \to \infty$, with $\lambda^2/T$ kept constant [47]. Although superficially different than our previous considerations, the combination of infinite temperature with an infinitely extended bath spectrum implies a perfectly Markovian bath, which dissipates any information instantly, leading to a relaxation time approaching zero, $\tau_B \to 0$. It also implies bath correlation functions are $\langle B_\alpha(t) B_\beta(s) \rangle \propto \delta(t-s)$, which immediately reduces a Redfield-type master equation into a Lindblad form.

## 4.2 Jump operators

In the previous derivation, the introduction of eigenoperators $\mathcal{A}_\alpha(\Omega)$ of $H_S$, equation (9), may be just seen as a technical step to write down the dynamics of the corresponding coupling operators $A$ and simplify the subsequent treatment of the Born master equation. While their

---

[15]The same problem occurs in classical systems subject to weak stochastic time-dependent perturbations [22]. There, an averaged generator of the dynamics is taken to preserve positivity, in analogy with the coarse-graining procedure. More mathematical details of the positivity condition can be found in [43].

mathematical convenience is a relevant advantage, they also play an important role in shaping and guiding the physical intuition on the Lindblad equation, because each of the $\mathcal{A}_\alpha(\Omega)$ represents the set of possible transitions with a given frequency $\Omega$ that can be induced in the system by the environment—they define *dissipative channels*. For example, if $H_S$ has a non-degenerate spectrum with eigenstates $|n\rangle$ (with energies $\varepsilon_n$), then the eigenoperators will have the form $\mathcal{A}_\alpha(\Omega) = \langle m|A_\alpha|n\rangle |m\rangle\langle n|$, with $\hbar\Omega = \varepsilon_n - \varepsilon_m$. The physical interpretation of such an operator is that the coupling to the bath is able to induce transitions from the eigenstate $|n\rangle$ to $|m\rangle$, accompanied by the emission (or absorption) of an energy $\hbar\Omega$. The detailed information about the system spectrum contained in the eigenoperators allows one to make quantitative estimates of the limits of Markovian and rotating wave approximations, and is responsible for important properties of the final master equation, such as yielding the correct thermalization of the system when the environment is initially at equilibrium. Ultimately, the jump operators will be given by linear combinations of the eigenoperators—see 5.1. For these reasons, in the next paragraphs we will provide further details on the eigenoperators.

### 4.2.1  Coupling operators vs. jump operators

Consider a two-level system with states $|\pm 1\rangle$ governed by a Hamiltonian $H_S = \frac{1}{2}\Delta\sigma^z$, so that the energies of the two levels are $H_S|\pm 1\rangle = \pm\frac{1}{2}\Delta|\pm 1\rangle$. This spin-1/2 system couples to the environment via a transverse field $H_I = \sigma^x B$, where $B$ is an (Hermitian) operator acting on the bath degrees of freedom. The coupling operator of the system, $\sigma^x$, is not an eigenoperator of the Hamiltonian. To determine its time-evolution in the interaction picture, it has to be projected onto the energy levels of $H_S$, $|1\rangle\langle 1|\sigma^x|-1\rangle\langle -1| = \sigma^+$, and $|-1\rangle\langle -1|\sigma^x|1\rangle\langle 1| = \sigma^-$. As a result, the associated Lindblad master equation has two dissipative channels $\sigma^\pm$ governed by the jump operators, which are the eigenoperators of the Hamiltonian $[H,\sigma^\pm] = \pm\Delta\sigma^\pm$. These correspond to the *transition* energies of the Hamiltonian $\pm\Delta$, rather than just the energy levels $\pm\frac{1}{2}\Delta$. Conversely, if the initial coupling to the environment is $H_I = \sigma^+ B_1 + \sigma^- B_1^\dagger$, it is already in the eigenoperator basis of the Hamiltonian. In that case, $\sigma^\pm$ will also become the jump operators in the corresponding Lindblad problem. While the set of jump operators is the same for the two cases, the difference in the underlying system-bath coupling will generally manifest in the difference of their coefficients, namely in the Lamb shift and decay rates[16].

### 4.2.2  Hermiticity

The above example of a two-level system illustrates that deriving the Lindblad equation from the underlying Hamiltonian microscopic model generally leads to non-Hermitian jump operators. Since $\mathcal{A}_\alpha^\dagger(\Omega) = \mathcal{A}_\alpha(-\Omega)$, the only possibility for the microscopic derivation to yield a Hermitian eigenoperator with $\mathcal{A}_\alpha^\dagger(\Omega) = \mathcal{A}_\alpha(\Omega)$ is that $\Omega = 0$, which through $[H_S, \mathcal{A}_\alpha(\Omega)] = -\hbar\Omega\mathcal{A}_\alpha(\Omega)$ implies that any Hermitian eigenoperator must commute with $H_S$—i.e., it must be a constant of motion, $[H_S, \mathcal{A}_\alpha(0)] = 0$. This finding has an important physical consequence, since if the resulting Hermitian jump operator is conserved by the system dynamics, then it can only induce pure decoherence—i.e., expressing $\rho_S(t)$ in the basis of the eigenstates of $H_S$, the off-diagonal elements of $\rho_S(t)$ decay exponentially, while the diagonal components do not evolve. Vice versa, a Hermitian jump operator that does not commute with the Hamiltonian drives the system to an infinite temperature state regardless of the state of the bath[17]. If this

---

[16]Although in the common case in which $B = B_1 + B_1^\dagger$, with $B_1$ containing only bosonic annihilation operators in the bath, then the second interaction is a truncation of the first via the Hamiltonian RWA (see 5.1.1), the two resulting master equations coincide for a bath at equilibrium. This situation is specific of two-level systems.

[17]For an infinite-temperature state to be a stationary state of a Lindblad dynamics the minimal requirement is that the jumps are normal operators, i.e., $[L_a, L_a^\dagger] = 0$. Then, we have that the Lindblad generator is equal to its adjoint and $\mathcal{L}(\mathbb{1}) = 0$, where $\mathcal{L}$ is the time-evolution generator in the Lindblad form. While this condition is satisfied in the

were true even for a finite-temperature bath, thermodynamics would be violated. Lindblad equations with Hermitian jump operators can be obtained from non-Hamiltonian microscopic evolutions, such as measurements [16, 49] or coupling to classical stochastic fields [50, 51]. Hermitian jump operators are often introduced phenomenologically for describing dephasing (meaning, loss of quantum-mechanical coherence without energy exchange with the environment), but this description can only be accurate at early times, since at later times equilibration with the environment is to be expected [18] (see also [40]).

### 4.2.3 Thermalization

The form of the jump operators is tied to the spectral properties (eigenergies and eigenstates) of the system Hamiltonian. If any perturbation to the system Hamiltonian is introduced so that $H'_S = H_S + V$, the system spectrum is altered, and the jump operators need to change accordingly. As long as $V$ can be considered a perturbation with respect to $H_S$, the new jump operators will remain "close" to the unperturbed ones, and employing the original jump operators will introduce only a small error in the dynamics. In general, however, one needs to diagonalize the full Hamiltonian $H'_S$ and find the new eigenoperators that become the new dissipative channels. Although this step is straightforward for models with few degrees of freedom, it may become an issue for many-body systems whose Hamiltonians are not trivially solvable (e.g., those constituted by non-interacting particles or spins), as numerical diagonalization of $H_S$ becomes rapidly intractable as the number of constituents increases [52].

The correct relation between the jump operators and the system Hamiltonian ensures thermodynamically accurate behavior of the open system. If the environment is at thermodynamic equilibrium, a physical master equation should lead the system to thermalize at the same temperature as the bath. Indeed, the Lindblad equation that we are going to derive has the property that if the bath is at temperature $T = (k_B \beta)^{-1}$, then the Gibbs state $\rho_S \propto e^{-\beta H_S}$ is a stationary state[19], and that this state is unique under broad assumptions—namely, the absence of strong symmetries and corresponding conserved quantities [5, 40, 55]. Moreover, a number of statements expected from thermodynamics can be proved, such as the second law—namely, that the evolution under (1) increases the thermodynamic entropy of the system [40]. Intuitively, these properties can be achieved only because the jump operators "know" the spectrum of $H_S$ through the eigenoperators, while the temperature of the bath is encoded in the fluctuation-dissipation relation obeyed by the correlation functions—i.e., in the $\Gamma_{\alpha\beta}(\Omega)$ coefficients. As an example, let us consider again the two-level system from section 4.2.1, $H_S = \frac{1}{2}\Delta\sigma^z$, coupled to a bath at zero temperature via an interaction $H_I = \sigma^x B$. Then, the Lindblad construction yields a single jump operator $L = \sigma^-$ with a certain rate $\gamma = \gamma(\Delta)$, corresponding to spontaneous emission[20], and it is straightforward to verify that

---

case of Hermitian, conserved jumps, the conservation of the diagonal elements of $\rho_S(t)$ prevents the system from reaching the infinite-temperature state starting from any other state. Even in the case of non-conserved, Hermitian jumps, the infinite-temperature state needs not be the unique steady state, as strong symmetries can protect other steady states [48].

[18]Naturally, in concrete physical scenarios (like modeling experiments), extra jump operators associated to incoherent losses or pumps are unavoidably present (both non-hermitian), including possibly time-dependent drive (not covered in this review). Both would guarantee that the system relaxes into a more physical steady state.

[19]An attentive reader might notice that this is not the exact thermal state for a system and a bath at a common temperature $T$, since in general $\text{Tr}_B[e^{-\beta(H_S + \lambda H_I + H_B)}]$ is not proportional to $e^{-\beta H_S}$. Indeed, it can be proven that a master equation at order $\lambda^{2n+2}$ can predict stationary states only to order $\lambda^{2n}$ [53]. The above state is nevertheless the one that is usually considered in thermodynamics [54], namely the state of a small subsystem whose interaction with the rest of the bath, although nonvanishing to ensure thermalization, has a negligible effect on thermodynamic observables.

[20]The other possible jump operator $\sigma^+$ would correspond to absorption of energy from the bath, but this cannot happen if the latter is in the ground state. This property is ensured by the vanishing of the corresponding absorption rate $\gamma(-\Delta) = e^{-\beta\Delta}\gamma(\Delta) \to 0$ for $\beta \to \infty$.

the only stationary state is the ground state, with $\langle \sigma^z \rangle = -1$, as expected from thermodynamics. However, let us introduce a small coupling $V = h\sigma^x/2$ between the ground and excited states. Thermodynamically, we expect the system to go to the new ground state, in which $\langle \sigma^z \rangle = -1 + h^2/(2\Delta^2) + \mathcal{O}(h^4)$ is a little larger than $-1$. However, if we keep the old jump operator $\sigma^-$, we reach a different state, with $\langle \sigma^z \rangle = -1 + h^2/(2\Delta^2 + \gamma^2/2) + \mathcal{O}(h^4)$, which does not correspond to thermal equilibrium, but only approximates it for $\gamma \ll \Delta$.

### 4.2.4  Many body systems

The dependence of the jump operators on the eigenstates and spectrum of the system Hamiltonian has important conceptual consequences in the case of many body systems. For models consisting of multiple interacting parts, the dissipative part in the Lindblad equation (1) is not the sum of the corresponding expressions for each part taken individually. In particular, the correct jump operators will generally be *nonlocal*, in the sense that they will act on more than one part simultaneously. The eigenstates of a system composed of parts A and B, which are mutually interacting, have a weight on both subsystems, and the eigenoperators, and thus the jumps inherit this property through their definition (9). Although the nonlocal nature of jump operators might sound counterintuitive (after all, the interaction with the bath is generally local), it is once again dictated by the requirement of thermalization — the jump operators must "know" the correct eigenstates of the full $H_S$ to drive the system into the appropriate Gibbs state. These considerations apply even if the two subsystems are coupled to independent reservoirs, as long as the latter are at the same temperature [40]. Indeed, if the two parts A and B were not interacting, $H_S = H_A + H_B$, and each one coupled to its own reservoir, the density matrix would factorize exactly $\rho_S(t) = \rho_A(t) \otimes \rho_B(t)$ and the two parts would evolve according to their own Lindbladian dynamics. Nevertheless, the stationary state would still be given by the collective Gibbs state $e^{-\beta H_A} \otimes e^{-\beta H_B} = e^{-\beta(H_A + H_B)} \equiv e^{-\beta H_S}$. However, if we introduce an interaction between the two parts, $H'_S = H_A + H_B + V_{AB}$, then the stationary state $e^{-\beta H'_S}$ is no longer factorizable, and cannot be reproduced by two independent Lindblad equations for $\rho_{A,B}$. Thus, in this case, the jump operators must act on both subsystems[21]. The same conclusion can be reached even without invoking thermodynamics, as the dissipative processes caused by the two baths can become correlated through the interaction between the subsystems [57]. On the other hand, if the two subsystems are coupled to a common bath, the jump operators may become nonlocal also through the bath itself, due to the structure of the coefficients $\Gamma_{\alpha\beta}(\Omega)$ in the $\alpha$, $\beta$ indices—jumps acting on different subsystems can share a common coefficient linking them together within the master equation. This is what happens in correlated emission, i.e., the collective decay of closely spaced atoms [58–63].

In general, employing local jump operators in multipartite (or many body) systems is at best an approximation—although often a computationally convenient one—that may be valid for weak interactions between the subsystems and for times before the timescale required to fully thermalize the system. Whether the approximation is accurate or not depends on the particular case at hand [55, 57, 64–70], although there are some general attempts [52, 71]. Nevertheless, the resulting lack of (complete) thermalization can lead to inconsistent physical predictions [72, 73]. See also [74] for a discussion on the relation between perturbation theory, thermalization and conservation laws in various master equations.

We remark that the various properties of the Lindblad master equation listed in the previous paragraphs are related to the main setup considered in this review, namely that of a system

---

[21]It should be evident that in the present formalism, thermalization is enforced by the dissipative part of the Lindblad dynamics (the jump operators), and not by the system dynamics (as it happens for isolated quantum systems obeying the eigenstate thermalization hypothesis [56]). Hence, the mere presence of the interaction $V_{AB}$, which acts on both parts of the system, cannot be sufficient to bring any initial state to the thermal one if the jumps still act on the two subsystems independently.

that at $t = 0$ is put in contact with a reservoir at thermodynamic equilibrium. The range of possible kinds of behavior increases significantly if the bath is initially not at equilibrium, and by considering special choices of the coupling operators $B_\alpha$, which are reflected in the structure of the $\Gamma_{\alpha\beta}(\Omega, t)$ coefficients (equation (12)) in the $(\alpha, \beta)$ indices.

## 4.3 The protagonist of Markovianity: the bath spectral density

In this section we introduce the main object determining the Markovian properties of a bath—the Fourier transform of its correlation function(s) $G_{\alpha\beta}(t) \equiv \langle B_\alpha(t)B_\beta(0) \rangle$, known as spectral density or spectral function.

A fundamental condition for a bath to provide dissipation is that it has to be large enough (in the thermodynamic sense) that its spectrum can be considered to be continuous. Roughly speaking, the level spacing of the bath should be much smaller than the dissipative decay rate $\hbar/\tau_R$. If this condition is not fulfilled—for instance, in a small bath—the information of the previous states of the system will be able to feed back on it, providing memory and thus breaking Markovianity. This finite-size effect is known as recurrence, and we refer to [30,75] for explicit examples in the context of open systems.

It is convenient to specify the properties of a bath with a continuous spectrum by working in the frequency domain. Moreover, we will show that this point of view also facilitates the assessment of Markovianity of the system's dynamics. Let us consider for simplicity the correlation function for a single bath operator, $G(t) \equiv \langle B(t)B(0) \rangle$, and its Fourier transform $J(\omega)$

$$ G(t) = \int_{-\infty}^{\infty} \frac{d\omega}{2\pi} e^{-i\omega t} J(\omega) \,, \tag{15} $$

that we will call spectral density[22]. We are considering the bath to be in thermodynamic equilibrium, $\rho_B \propto e^{-\beta H_B}$. The reader familiar with signal processing will recognize $J(\omega)$ as the quantum equivalent of the power spectrum, as introduced by the Wiener-Khinchine theorem [1,6,77]. An appealing feature of the spectral density is that it can be easily interpreted, as it identifies the energies of the excitations coupled to the system and the intensity of their coupling to the system. In the signal-processing analogy mentioned above, $J(\omega)$ quantifies the (possibly quantum) "noise" of the bath. We can obtain a formal expression for $J(\omega)$ by working in the basis of exact eigenstates of $H_B$, $H_B |a\rangle = E_a |a\rangle$ and performing a Lehmann-type decomposition[23],

$$ J(\omega) = 2\pi\hbar \sum_{a,b} p_b |\langle b|B|a\rangle|^2 \delta(\hbar\omega - E_a + E_b) \,, \tag{16} $$

where $p_b = e^{-\beta E_b}/Z_B$ is the Boltzmann weight of the state $b$ ($Z_B = \text{tr } e^{-\beta H_B} = \sum_a e^{-\beta E_a}$ being the partition function). In the above equation, we have considered a bath with a large but finite size, so that its spectrum is discrete. In the thermodynamic limit the sums converge to integrals and $J(\omega)$ becomes continuous. The expression (16) shows that $J(\omega)$ is essentially a weighted density of states for the transitions mediated by the operator $B$—it is nonzero (and positive[24]) only for frequencies $\omega$ corresponding to possible *excitation energies* $E_a - E_b$.

---

[22]The use of the term "spectral density" is not homogeneous in the literature, and some authors use "spectral function" instead. We will reserve the latter to the part of the function $J(\omega)$, which describes the structure of the bath's spectrum without the statistical occupation factors (e.g., Bose-Einstein or Fermi-Dirac distributions) that are included in the full object considered here. This usage is in line with the field theory literature [76]. We will provide specific examples in section 4.4.4.

[23]The condition $\text{tr}[B(t)\rho_B] = 0$ can be used to restrict the sum to $a \neq b$. This ensures that $J(\omega)$ does not contain a peak $\propto \delta(\omega)$, barring the presence of degenerate states.

[24]For correlation functions of different operators, $G_{\alpha\beta}(t)$, the individual spectral densities $J_{\alpha\beta}(\omega)$ are not necessarily positive for $\alpha \neq \beta$, but form a positive semidefinite matrix in the $(\alpha, \beta)$ indices [5,55]. This property ensures that the decay rates $\gamma_a$ in the Lindblad equation (1) are positive.

Temperature has a strong influence on $J(\omega)$ as well, as it constrains the available excitations, and we will show in more detail that it contributes to determining to which extent a bath can be considered to be considered Markovian in section 4.4.4. For example, at zero temperature $J(\omega) = 2\pi\hbar \sum_a |\langle \text{gs}|B|a\rangle|^2 \delta(\hbar\omega - E_a + E_{\text{gs}})$ is nonvanishing only for positive frequencies[25] corresponding to the excitation energies $E_a - E_{\text{gs}} > 0$ above the ground state $|\text{gs}\rangle$. More in general, $J(\omega)$ for $\omega > 0$ quantifies the availability of bath excitations for absorption of energy from the system. For finite temperatures, $J(\omega)$ generally acquires weight at negative frequencies, which signals the availability of bath excitations that can be absorbed *by* the system. At frequencies $|\omega| \to \infty$, $J(\omega)$ is usually taken to vanish. This might be because the energies $E_a$ have an upper bound (namely, the excitations created by $B$ have a finite bandwidth, as, for instance, in a spin system) or because the matrix elements $\langle b|B|a\rangle$ decrease. The exact behavior of $J(\omega)$ at large frequencies might not be well-known, so a cutoff function is introduced as to obtain finite results while encapsulating our ignorance of the exact high-energy behavior into a cutoff parameter, in the spirit of the renormalization group. Often, the exact shape of $J(\omega)$ is chosen phenomenologically (constrained by known limits of high and low frequency), while for some scenarios, it can be derived from the knowledge of $H_B$ and $B$.

An abstract limiting case that is useful to consider is that of a completely flat spectral density, $J(\omega) = \text{const.}$, which translates to $G(t) \propto \delta(t)$—namely, a function with a vanishing correlation time $\tau_B$. In this scenario, the Markovian approximation is exact[26]. While a flat $J(\omega)$ might be a reasonable approximation in some situations, it is generally unphysical because it would imply that the bath has states with arbitrarily low energy, i.e., no ground state. A $G(t) \propto \delta(t)$ would correspond for instance to the case of a quantum system driven by a noisy classical drive (e.g. the intensity or phase noise of a laser), which is perfectly Markovian. This is however clearly an approximation since this will always have some finite correlation time resulting into non-Markovian effects.

### 4.3.1 Spectral density vs. Correlation function *

The shape of $J(\omega)$ is generally more informative than the real-time behavior of $G(t)$ in understanding whether the Markovian approximation is accurate or not. The main reason is that the late-time behavior of $G(t)$ is sensitive to features of $J(\omega)$ that are often irrelevant to the Markovian approximation. In this regard, it is helpful to clarify a point in the connection between the correlation function and its Fourier transform. When dealing with the Born and the Markov approximations, one often has in mind an exponential behavior $G(t) \sim e^{-|t|/\tau_B}$. While this picture might be conceptually and even practically useful, it is generally false as a statement for the strict limit $t \to \infty$ and may apply only during an intermediate-time transient—see, for example, [78]. In fact, in most physical scenarios, the late-time decay of $G(t)$ is algebraic, namely, $G(t)$ decays as a power of time. This conclusion can be reached from different perspectives. If the exponential behavior $G(t) \sim e^{-|t|/\tau_B}$ continued for all times, the Fourier transform of $G(t)$ would be a Lorentzian function $\propto [(\omega\tau_B)^2 + 1]^{-1}$, which would imply a bath with an infinite bandwidth. However, physical systems cannot have an infinite bandwidth—their spectrum needs to be at least bounded from below (in the sense that there are no states with lower energy than the ground state). In baths whose degrees of freedom have a finite Hilbert space (e.g., fermions and spins), one usually has a spectrum that is also bounded from above

---

[25]Notice that we are working in the canonical ensemble. If $H_B$ conserves the number of particles $N_B$, while $B$ does not—i.e., system and bath exchange particles—then, it is convenient to work in the grand-canonical ensemble, with $H_B \to H_B - \mu N_B$, where $\mu$ is the chemical potential of the bath. Then, $J(\omega) > 0 \iff \omega > 0$ is recovered at zero temperature. Otherwise, in the canonical ensemble $J(\omega)$ will have jumps in correspondence to $\omega = \pm\mu$.

[26]Only if the bath is noninteracting and Wick's theorem applies so that all higher-order correlation functions are delta-shaped as well. Otherwise, the Markovian approximation is exact only at the level of the second-order Born master equation (7).

(i.e., the bath spectrum terminates at a maximum energy). Some bath spectra can even display gaps, as in the case of photonic crystals, to be discussed in 4.5.2. In general, whenever the bath spectrum has a band edge at a certain energy $\hbar\omega_e$, in the sense that there are no levels in an energy range below (or above) $\hbar\omega_e$, one can expect that $J(\omega)$ or its derivatives will be discontinuous at $\omega = \omega_e$. For instance, in the previous section we have mentioned that at zero temperature $J(\omega)$ vanishes for negative frequencies, while it is nonzero for positive ones, and in most physical scenarios this transition does not happen smoothly. We will present concrete examples of band edges in 4.4.4. The important point of the presence of such discontinuities is that each of them contributes to a power-law asymptotic behavior in the correlation function, due to the asymptotic properties of Fourier transforms[27] [83, 84]. For instance, if $J(\omega) \sim \omega^\alpha$ for $\omega \to 0+$ and vanishes for $\omega \leq 0$ (a common scenario for bosonic baths at zero temperature [5, 30, 85]), then $G(t) \propto \int_0^\infty d\omega J(\omega) e^{-i\omega t} \sim t^{-\alpha-1}$ for $t \to \infty$. These band-edge contributions to $G(t)$ are always present in physical systems, but they usually come with a small amplitude, so that they become predominant only at very late times after the correlation function has already decayed to a small value. Thus, the effect of these slow-decaying tails is usually heavily suppressed in the Born master equation. Nevertheless, the error introduced by the Markovian approximation is dependent on the algebraic tails of $G(t)$ and requires that they decay sufficiently fast [28, 78]. Although the general mathematical framework for quantifying this condition is rather unwieldy to apply[28], in the representative case of a reservoir consisting of noninteracting (or weakly interacting [86]) fermions, it is sufficient (but not necessary) that $G(t)$ decays strictly faster than $1/t$. The behavior of $G(t)$ at intermediate times is sensitive to other features of $J(\omega)$ away from its edges, but it is generally hard to pinpoint exactly which regime of $G(t)$ is most relevant to the Markovian approximation[29]. Another conceptual problem of power-law tails in $G(t)$ is that they lack a typical time scale—namely, there is apparently no bath correlation time $\tau_B$. The intuitive definition [28] $\tau_B = \int_0^\infty dt \, t \, |G(t)| / \int_0^\infty dt |G(t)|$ can still apply if $G(t) \sim t^{-1-\alpha}$ with $\alpha > 1$, and it is dominated by the short-time behavior of the correlation function—i.e., $\tau_B$ is of the order of the high-frequency cutoff in $J(\omega)$. However, this definition of $\tau_B$ diverges for the common scenario of an Ohmic bath with $\alpha = 1$, while in most cases the Markovian approximation works rather well even for this category of baths [5, 30, 88]. In the next section, we will show that looking at the spectral density $J(\omega)$ bypasses these issues and provides a more natural way of understanding Markovianity.

## 4.4 Limitations of the Markov approximation

The Markov approximation (11) provides a big computational simplification for the master equation. However, as with any approximation, it also introduces some errors with respect to the original dynamics. In this section, we will discuss an exactly solvable toy model of non-Markovian dynamics, which will help us to illustrate the kind of errors that the Markov approximation introduces and its consequent limits of validity.

---

[27]One can also make a stronger statement: whenever $J(\omega)$ vanishes below some frequency, a general theorem by Paley and Wiener [79–81] guarantees that the Fourier transform of $J(\omega)$ must decay more slowly than an exponential at late times. Even more generally, only the Fourier transform of a smooth function (i.e., a function whose derivatives of any order are continuous) decays faster than any power law [82] (e.g., decays exponentially).

[28]The early work of Davies [19–21] provided mathematical conditions on the validity not only of the Born-Markov approximation but of the whole weak-coupling Lindblad description. We refer the interested reader to the original literature for more details.

[29]For common shapes of $J(\omega)$ for fermionic and bosonic baths, one can show that for finite but very low temperatures, far smaller than the bandwidth, the initial decrease of $G(t)$ is exponential, with $\tau_B^{-1} = \xi \pi k_B T/\hbar$, where $\xi = 1$ for fermions (see e.g., section IV of [87]) and $\xi = 2$ for bosons ([78], section 5.2)—in other words, $\tau_B$ is the inverse of the first Matsubara frequency, as also noticed in section 3.6.2 of [5]. However, our numerical analysis for bosons with $J(\omega) \propto \omega^\alpha e^{-\omega/\Lambda}$ suggests that this behavior only occurs for odd integer values of $\alpha$.

### 4.4.1 A toy model of non Markovian dynamics

Let us re-write the Born master equation (7) in the Schrödinger picture

$$\frac{\mathrm{d}}{\mathrm{d}t}\rho_S(t) = -\frac{i}{\hbar}[H_S, \rho_S(t)] - \frac{\lambda^2}{\hbar^2}\sum_{\alpha\beta}\int_0^t \mathrm{d}s \left\langle B_\alpha(t)B_\beta(s)\right\rangle \left[A_\alpha e^{-\frac{i}{\hbar}H_S(t-s)}A_\beta\rho_S(s)e^{\frac{i}{\hbar}H_S(t-s)}\right.$$

$$\left. - e^{-\frac{i}{\hbar}H_S(t-s)}A_\beta\rho_S(s)e^{\frac{i}{\hbar}H_S(t-s)}A_\alpha\right] + \text{H.c.} , \tag{17}$$

where, with a slight abuse of notation, we are using the same symbol $\rho_S(t)$ for the system's density matrix as in the interaction picture. We will focus on the most common case of a stationary bath, in which, as discussed in the previous sections, the correlation functions depend on the difference of times only, $\left\langle B_\alpha(t)B_\beta(s)\right\rangle = \left\langle B_\alpha(t-s)B_\beta(0)\right\rangle$. Let us consider the eigenstates $|n\rangle$ of the system Hamiltonian $H_S$, which satisfy $H_S|n\rangle = \hbar\omega_n|n\rangle$. If we take the matrix element of the above equation between two eigenstates $|n\rangle$ and $|m\rangle$ of $H_S$, we obtain an equation in the form

$$\frac{\mathrm{d}}{\mathrm{d}t}[\rho_S(t)]_{nm} = -i(\omega_n - \omega_m)[\rho_S(t)]_{nm} + \sum_{n'm'}\int_0^t \mathrm{d}s K_{nm;n'm'}(t-s)[\rho_S(s)]_{n'm'} , \tag{18}$$

where the integral kernel $K_{nm;n'm'}(t-s)$ is proportional to the bath correlation functions. The above equation is a system of coupled, linear integro-differential equations for the functions $[\rho_S(t)]_{nm}$. In principle, it can be solved for any kernel using Laplace transforms, but its matrix structure in the $(m, n)$ indices makes it difficult to gain some intuition on its solutions. Since we want to understand the essential qualitative consequences of a non-Markovian evolution, we will focus on a toy model that is the simplest equation in the form (18), namely

$$\frac{\mathrm{d}}{\mathrm{d}t}f(t) = -i\omega_0 f(t) + \int_0^t \mathrm{d}s K(t-s)f(s) , \tag{19}$$

where $\omega_0$ is a real angular frequency and $K(t)$ is a generic integral kernel. The above equation might describe the time evolution of an off-diagonal element of $\rho_S(t)$, $f(t) = [\rho_S(t)]_{nm}$ that is completely decoupled from all the others, and $\omega_0$ would correspond to the transition frequency $\omega_n - \omega_m$. In the spirit of the Born equation (17), we will assume that $K(t)$ is "small" in the appropriate perturbative sense with respect to the "Hamiltonian" term $-i\omega_0 f(t)$. By analogy with equation (17) we take $K(t) \propto \lambda^2$. We will show that these assumptions underpin the Markovian approximation, and are not related to the existence of an underlying perturbation theory. Despite its apparent simplicity, equation (19) already contains many of the interesting properties of non-Markovian dynamics.

An explicit derivation of equation (18) from equation (17) shows that the integral kernel $K_{nm;n'm'}(t-s)$ will generally depend on the set of system transition frequencies $\omega_a - \omega_b$—namely, $K(t-s)$ in equation (19) should in principle depend on $\omega_0$. In the following discussion, we are going to drop this complication and assume that $K(t-s)$ can be chosen freely. The ideas that we will present in this simplified scenario can be easily generalized to the more complicated one[30].

We notice that much of what will be written is simply a generalization of the well-known problem of the spontaneous decay in a two-level atom [5, 6, 77, 89] (see also the discussion in [90]). Since in the following it might be useful to have this concrete physical example in mind, we now quickly introduce the problem. The Hamiltonian of the system and bath is

$$H = \hbar\omega_0 |e\rangle\langle e| + \hbar\sum_\lambda(g_\lambda |e\rangle\langle g| b_\lambda + g_\lambda^* |g\rangle\langle e| b_\lambda^\dagger) + \sum_\lambda \hbar\omega_\lambda b_\lambda^\dagger b_\lambda , \tag{20}$$

---

[30]The $\omega_0$ dependence of the kernel becomes relevant for the dynamics of diagonal matrix elements $[\rho_S(t)]_{nn}$, for which there is no Hamiltonian term $-i\omega_0 f(t)$ to provide the unperturbed dynamics.

where $|g\rangle$ and $|e\rangle$ are the ground and excited states of the atom, and $b_\lambda$ are bosonic modes describing the photon excitations of the electromagnetic (EM) field (i.e., $\lambda$ specifies a three-dimensional momentum and a polarization), with frequencies $\omega_\lambda$. The couplings $g_\lambda$ depend in a known way on the electric dipole of the atom and on fundamental constants such as the speed of light. From a physical perspective, the above Hamiltonian is already the result of various approximations, such as the restriction to two levels, the dipole approximation and the "Hamiltonian" rotating-wave approximation (HRWA; see 5.1.1). We will not comment on the regimes of validity of these. If the atom is initially in its excited state while the EM field is in the vacuum $|0\rangle_S$, $|\psi(0)\rangle = |e\rangle_S |0\rangle_B$, then the subsequent dynamics will bring the atom to the ground state while generating at most one photon[31]

$$|\psi(t)\rangle = f(t) |e\rangle_S |0\rangle_B + \sum_\lambda f_\lambda(t) |g\rangle_S b_\lambda^\dagger |0\rangle_B \ . \tag{21}$$

In the above equation, $f(t)$ is the probability amplitude of finding the atom in its excited state at time $t$. By substituting the above Ansatz into the Schrödinger equation with Hamiltonian (20) and solving for $f(t)$, one can show by standard procedures [5,77] that $f(t)$ is determined by the toy model equation (19), with kernel $K(t) = -\sum_\lambda |g_\lambda|^2 e^{-i\omega_\lambda t}$. While in the case of spontaneous decay the shape of $K(t)$ is dictated by fundamental properties of the EM field, the following observations are valid for (almost[32]) any kernel $K(t)$. To make contact with the master equation for the atom only, we consider a slightly more general initial state that has an amplitude on the atom-field ground state, $|g\rangle_S |0\rangle_B$, namely $|\psi(0)\rangle = f_g |g\rangle_S |0\rangle_B + f(0) |e\rangle_S |0\rangle_B$. The first term does not evolve in time, while the second evolves according to equation (21). Then, the exact density matrix for the atom has components $\langle e|\rho_S(t)|e\rangle = 1 - \langle g|\rho_S(t)|g\rangle = |f(t)|^2$ and $\langle e|\rho_S(t)|g\rangle = \langle g|\rho_S(t)|e\rangle^* = f_g^* f(t)$ [5].

### 4.4.2 The Markov approximation in the toy model *

We are going to derive the Markovian approximation for the toy model (19), whose approximate solution will be compared with the exact one. We proceed in perfect analogy with the derivation of the Lindblad equation. First of all, we introduce the interaction picture $\tilde{f}(t) = e^{i\omega_0 t} f(t)$, that solves

$$\frac{\mathrm{d}}{\mathrm{d}t} \tilde{f}(t) = \int_0^t \mathrm{d}s\, e^{i\omega_0(t-s)} K(t-s) \tilde{f}(s) \ . \tag{22}$$

We employ the same approach as for the master equation. We assume that $K(t)$ decays on a much faster timescale $\tau_B$ than the typical timescale of the evolution of $\tilde{f}(t)$, $\tau_R$, so that we can bring the latter out of the integral[33]:

$$\frac{\mathrm{d}}{\mathrm{d}t} \tilde{f}(t) \approx \tilde{f}(t) \int_0^t \mathrm{d}s\, e^{i\omega_0(t-s)} K(t-s) = \tilde{f}(t) \int_0^t \mathrm{d}s\, e^{i\omega_0 s} K(s) \ , \tag{23}$$

where in the second equality, we changed the integration variable $s \to t-s$. We call the above step the "time-local approximation", in the sense that it replaces equation (22) with a memoryless equation, where the $\frac{d}{dt}\tilde{f}(t)$ does not depend on the history of $\tilde{f}(t)$. The equation that is

---

[31]This is a consequence of the HRWA, that causes the Hamiltonian (20) to conserve the number of excitations $N = |e\rangle\langle e| + \sum_\lambda b_\lambda^\dagger b_\lambda$.

[32]Mathematically, one can put enough pathological features in $K(t)$—i.e., in the spectral density $J(\omega)$—to yield vastly different dynamics than the one that will be considered here. While the solution to equation (19) presented in Appendix B is completely general, inferring its properties requires some assumption on the shape and regularity of $J(\omega)$ which are found in the most common physical scenarios.

[33]We are following the "standard" Markov approximation. One can follow [19, 20] and [22] and employ a different approximation scheme. The result would be a slightly different equation with a rate $\Delta(t)$ differing from the one below by terms of order $\lambda^2$. This approach would not change the qualitative discussion that follows.

obtained in this way is not yet fully Markovian, since it has a notion of absolute time through the time dependence of the rate $\Delta(t) = \int_0^t ds\, e^{i\omega_0 s} K(s)$. This property is a weak form of non-Markovian behavior [91]. The solution to the ordinary differential equation equation (23) is

$$\tilde{f}_{\text{tl}}(t) = f(0)e^{\int_0^t ds\Delta(s)} , \tag{24}$$

where the subscript "tl" stands for time-local and is a reminder that the above solution is approximate. The full Markov approximation is obtained by invoking once more the rapid decrease in time of $K(s)$, which implies that the rate $\Delta(t)$ will quickly converge to its asymptotic value[34] $\Delta_\infty \equiv \lim_{t\to\infty} \Delta(t)$ on the timescale $\tau_B$. Then, for later times we can replace the time-dependent rate with its asymptotic value,

$$\frac{d}{dt}\tilde{f}(t) \approx \Delta_\infty \tilde{f}(t) = \tilde{f}(t) \int_0^{+\infty} ds e^{i\omega_0 s} K(s). \tag{25}$$

The solution of the above equation yields the Markovian approximation to the original toy model (22), $\tilde{f}_{\text{M}}(t) = f(0)e^{\Delta_\infty t}$ or, reverting to the "Schrödinger picture",

$$f_{\text{M}}(t) = f(0)e^{-i\omega_0 t + \Delta_\infty t} . \tag{26}$$

The simplicity of the toy model can be useful to appreciate the role of the interaction picture in deriving the correct Markovian approximation. The crucial point is that in the perturbative regime that we are interested in, we have $|\Delta_\infty| \ll |\omega_0|$—meaning that the effect of the bath on the system becomes relevant on timescales $\sim 1/|\Delta_\infty|$ that are much longer than the intrinsic timescale $1/|\omega_0|$. The interaction picture removes this fast time scale to reveal only the slow dynamics that we are interested in approximating. The Markovian approximations (23) and (25) make sense only for the slow $\tilde{f}(t)$. Indeed, pulling the fast $f(s)$ out of the integral (19) would yield a very poor approximation [22].

To gain a better understanding of the Markovian solution we need to compute the rate $\Delta_\infty$. In analogy with the discussion in section 4.3.1, the kernel $K(t)$ will be defined by the spectral density $J(\omega)$, namely the negative of its Fourier transform:

$$K(t) = -\int \frac{d\omega}{2\pi} J(\omega) e^{-i\omega t} . \tag{27}$$

The minus sign in front of the integral comes from the interpretation of $J(\omega)$ as a *positive* density of excitations in a physical system, while $K(t)$ should give rise to a decay of $f(t)$. Substituting the above expression into the definition of $\Delta_\infty$ we obtain

$$\begin{aligned}
\Delta_\infty &= -\int \frac{d\omega}{2\pi} J(\omega) \int_0^{+\infty} ds e^{-i(\omega-\omega_0)s - 0^+ s} \\
&= \int \frac{d\omega}{2\pi} J(\omega) \frac{i}{\omega - \omega_0 - i0^+} = -i\mathcal{P}\int \frac{d\omega}{2\pi} \frac{J(\omega)}{\omega_0 - \omega} - \frac{1}{2}J(\omega_0) ,
\end{aligned} \tag{28}$$

where in the first line, we have added $e^{-0^+ s}$ to ensure convergence even after exchanging the order of integrals. The symbol $\mathcal{P}$ indicates the principal part of an integral. If we substitute the above result into equation (26) we find that

$$f_{\text{M}}(t) = f(0)e^{-i(\omega_0 - \text{Im}\,\Delta_\infty)t - J(\omega_0)t/2} \equiv f(0)e^{-i\tilde{\omega}_0 t - t/\tau_R} . \tag{29}$$

We recognize that imaginary part of $\Delta_\infty$ defines the "Lamb shift" $\Delta\omega_{\text{LS}} \equiv -\text{Im}\,\Delta_\infty$ that renormalizes the system frequency into $\tilde{\omega}_0 \equiv \omega_0 + \Delta\omega_{\text{LS}}$, while its real part defines the decay

---

[34]The skeptical reader can verify this statement with $K(t) \propto e^{-t/\tau_B}$, for which $\Delta(t) = \Delta_\infty[1 - e^{-(1-i\omega_0\tau_B)t/\tau_B}]$.

rate $\tau_R^{-1} = J(\omega_0)/2$. In particular, the positivity of the spectral density for all frequencies guarantees that $\tau_R > 0$, i.e. that the $f_M(t)$ decays. It is interesting to notice that, while the decay rate depends only on the spectral density at the system's frequency, the Lamb shift is sensitive to all frequencies of the bath. In particular, we can write the latter as

$$\Delta\omega_{LS} = \mathcal{P} \int \frac{d\omega}{2\pi} \frac{J(\omega)}{\omega_0 - \omega} = \lim_{\epsilon \to 0^+} \left[ \int_{-\infty}^{\omega_0 - \epsilon} \frac{d\omega}{2\pi} \frac{J(\omega)}{\omega_0 - \omega} - \int_{\omega_0 + \epsilon}^{\infty} \frac{d\omega}{2\pi} \frac{J(\omega)}{\omega - \omega_0} \right], \qquad (30)$$

which shows that the value of $\Delta\omega_{LS}$ is the result of a "tug of war" between level repulsions, with bath states at energies $\hbar\omega < \hbar\omega_0$ below the system energy yielding a positive contribution (first term on the right-hand side of the above equation), i.e., pushing $\tilde{\omega}_0$ above $\omega_0$, while those at energies above $\hbar\omega_0$ yield a negative contribution (second term).

Equation (29) (or (26)) provides the blueprint of Markovian dynamics: a linear, Markovian equation like (25) has only exponential solutions, which generally describe damped oscillations. Indeed, this is what we showed in Appendix A for the Lindblad equation—compare with the general solution (A.2). This statement remains valid also for the Redfield equation (11), if employed with the constant coefficients (13), since it is still a linear ODE for the density matrix elements. In a nutshell, if the system has a Hilbert space of dimension $d$, the matrix elements of the general solution to the Lindblad equation have the form[35]

$$\rho_{\alpha\beta}(t) = \sum_{\mu=0}^{D-1} c_{\mu;\alpha\beta} e^{\lambda_\mu t}, \qquad (31)$$

where $c_{\mu;\alpha\beta}$ are certain coefficients (determined by the Hamiltonian and jump operators, as well as the initial conditions) and the complex rates $\lambda_\mu$ have negative or vanishing real part (this is no longer true for the Redfield equation), so that each term describes damped oscillations. The number of terms $D$ in the above equation is at most $d^2$, in the sense that there can be at most $d^2$ distinct rates $\lambda_\mu$. Our toy model corresponds to the case in which $D = d = 1$. The effects of the non-Markovian nature of the full dynamics (19) can be manifested as deviations from the exponential behavior (26) (or (17)), which include quantitative corrections to the Markovian rates $\lambda_\mu$, and qualitative corrections, such as the presence of more than $d^2$ exponential terms[36] or completely non-exponential behaviors such as algebraic decay. We highlight that all these statements refer to the situation in which the system has a finite-dimensional Hilbert space, as usual in traditional AMO physics. If $d \to \infty$, such as in many body systems, then the sum (31) can converge to an algebraic decay even for purely Markovian dynamics (see, e.g., [8, 92, 93] and references therein). We also remark that, while Markovian dynamics implies exponential evolution, the converse is not true—see, for instance, the example discussed in 4.5.1.

It is worth noticing a peculiarity of non-Markovian dynamics like equation (17): if a solution converges in time to a stationary state $\rho(t) \to \rho_\infty \neq 0$, then—unlike what happens with a Markovian equation—$\rho(t) = \rho_\infty$ is not a solution for all times. In other words, $\rho(t) = 0$ is the only constant solution[37]. This behavior can be observed only with a higher-dimensional generalization of the toy model (19), since $f(t)$ decays to 0.

---

[35]With the possible modification that some of the exponential terms might be multiplied by polynomial functions of time, as already noted in that section. This exception does not alter the main point here.

[36]Namely, more than $d^2$ distinct rates $\lambda_\mu$. A particularly simple example of this phenomenon can be found in the toy model ($d = 1$) with $K(t) \propto e^{-t/\tau_B}$ (e.g., see [5], paragraph 10.1.2), in which two rates (the Markovian one, as well as a faster one $\lambda_{NM} \sim \tau_B^{-1}$) appear.

[37]The fact that it is a solution is guaranteed by the linearity of equation (17).

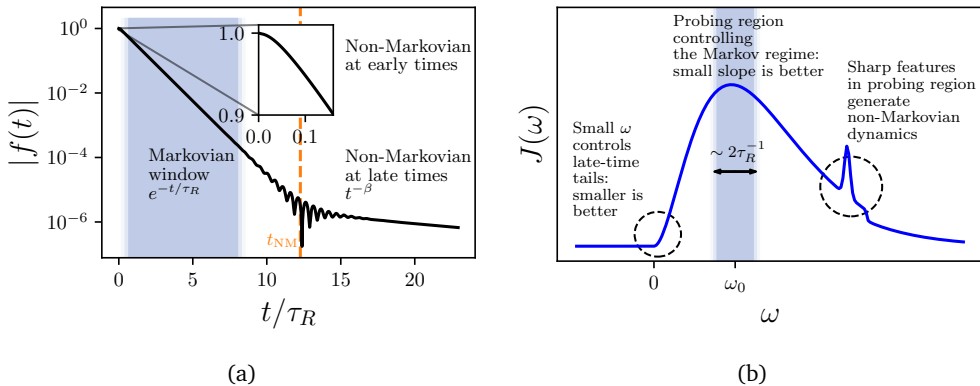

Figure 2: (a) Sketch of a typical solution to the toy model corresponding to a representative spectral density, similar to the one shown in (b). The plot has been generated by direct numerical integration of equation (19), assuming an Ohmic spectral density $J(\omega) = 2\pi\eta\omega e^{-\omega/\Lambda}\theta(\omega)$, with $\eta = 0.05$ and $\omega_0 = 0.7\Lambda$. (b) Cartoon of a spectral density summarizing the role of different frequency regions in the real-time behavior of the toy model. Notice that the axes in the plot have different scales, so the width of the probing region is actually the same as $J(\omega_0)$.

### 4.4.3  The full solution: Markovian and non Markovian regimes

In the following paragraphs, we will give an overview of non-Markovian effects and connect them to the shape of the spectral density $J(\omega)$. We will see that some of the main features of $J(\omega)$ that control the emergence of non-Markovian behavior are discontinuities and other non-analytic points. We will exemplify such cases with the common situation of a band edge, namely, we shall assume that $J(\omega)$ vanishes below some minimal energy $\omega_{\min}$, which we take to be zero. In general, this assumption corresponds to the physically relevant situation in which the bath providing the dissipation has a ground state, as mentioned in the discussion in 4.3.1. For instance, in the case of spontaneous decay, equation (20), the stability of the bosonic bath requires all bath frequencies $\omega_\lambda$ to be positive, so that $J(\omega) = 2\pi\sum_\lambda |g_\lambda|^2\delta(\omega-\omega_\lambda)$ vanishes for $\omega \le 0$. We are going to provide more examples of spectral densities for different baths in 4.4.4.

Equation (19) can be solved quite generally by means of the Laplace transform, and we present a detailed derivation in Appendix B. The salient features of the solution will be presented here. For rather generic spectral densities (like the one shown in Fig. 2b), the exact solution to the toy model (19) has the appearance shown in Fig. 2a. There are three main regimes, which are controlled by different properties of the spectral density.

**Non-Markovian behavior at early times**  The behavior of $\tilde{f}(t)$ at times shorter than the kernel (bath) decay time $\tau_B$ is generally non-Markovian. If $K(t)$ is finite at $t = 0$, the initial behavior is

$$\tilde{f}(t) = f(0)\left[1 - \tfrac{1}{2}|K(0)|t^2 + \mathcal{O}(t^3)\right], \tag{32}$$

where we used $K(0) = -\int d\omega J(\omega)/(2\pi) < 0$. The above expression can be obtained by computing the derivatives of $\tilde{f}(t)$ at $t = 0$ via equation (22). The parabolic behavior of $\tilde{f}(t)$ is related to the typical decay of quantum-mechanical amplitudes at short times[38] [5, 16, 77]—

---

[38]We expect the Born approximation to perform well at very short times since we assumed the initial state to be exactly factorized and the establishment of system-bath entanglement requires a time comparable to $\tau_R$ and thus much larger than $\tau_B$. Therefore, for very short times, the Born equation (6) is essentially exact.

indeed, in the example of spontaneous emission (20), $f(t)$ is the probability amplitude of the initial excited state. The time scale of the decay of equation (32) is set by $K(0)$, which probes the spectral density at all frequencies. The parabolic behavior of equation (32) is non-Markovian in the sense that it cannot come from the expansion of an exponential $e^{\Delta t}$, for the latter would have a finite first derivative at $t = 0$. Intuitively, the reason for the lack of Markovianity at such short times is that "there is no time" for the kernel $K(t)$ to fully decay in the convolution in the right-hand side of equation (19). Consequently, the Markovian approximation (25) leads to an overshoot for the value of $d\tilde{f}(t)/dt$ at $t = 0$—while the exact value is zero, (23) predicts a finite value $\Delta_\infty f(0)$. If we re-trace the the steps leading to the Markovian approximation (25), we can see that it cannot hold for times that are so short that the time-dependent rate $\Delta(t)$ is not yet saturated to its long-time value $\Delta_\infty$. In fact, at least for the lowest orders in $t$, the non-Markovian behavior above can still be obtained from the time-local equation (23).

The Lindblad equation (1) is similarly inadequate at very early times, because it predicts $d\rho(t)/dt$ to be finite at at $t = 0$, while the exact dynamics, as well as the Born equation (6), predicts a vanishing derivative. The physical origin of this discrepancy is that at times of the order of $\tau_B$ or shorter, the bath correlations that are responsible for dissipation have not fully developed yet. Keeping the time dependence in the coefficients (12) can amend this unphysical behavior and increase the accuracy of the Lindblad equation at short times, at the cost of employing a mildly non-Markovian master equation with time-dependent coefficients [24, 94] (see also [26, 44, 95, 96] for the alternative approach of slippage initial conditions). This does not mean any form of back-action of the bath on the system, but merely a sensitivity to the buildup of bath correlations [97].

**Markovian window at intermediate times**   For times much longer than $\tau_B$ the solution to equation (19) assumes an exponential form, meaning that the Markovian approximation is accurate. This behavior corresponds to the linear behavior on the logarithmic scale of Fig. 2a. The Markovian evolution persists until a crossover time regime characterized by oscillations. After this crossover, the decay is generally no longer exponential, hence non-Markovian. The crossover region occurs because of interference between the Markovian component of the solution and the non-Markovian component that becomes dominant at late times. The timescale $t_{NM}$ at which the crossover occurs is controlled by the bare frequency $\omega_0$, by the coupling $\lambda^2$ and by the way in which the spectral density vanishes at the band edge $\omega \to 0^+$. In more detail, $t_{NM}$ grows as $\omega_0$ increases or $\lambda^2$ decreases, and if $J(\omega) \sim \omega^\alpha$ for $\omega \to 0^+$, then $t_{NM}$ increases with $\alpha$. In the perturbative regime that we are interested in, $t_{NM}$ is much larger than the decay time $\tau_R$. Hence, most of the decay of $f(t)$ occurs during the Markovian regime, so that for practical purposes equation (25) can be used to predict the decay properties and even the stationary state[39]. The full solution in appendix B shows that the properties of the function in the Markovian regime are controlled by the shape of the spectral density at frequencies around $\omega_0$—more precisely, by the shape of $J(\omega)$ in the "probing interval"[40] $[\tilde{\omega}_0 - \tau_R^{-1}, \tilde{\omega}_0 + \tau_R^{-1}]$ around the shifted frequency $\tilde{\omega}_0 = \omega_0 - \text{Im}\,\Delta_\infty$. For simplicity, we can take the interval to be around the unperturbed frequency $\omega_0$ rather than $\tilde{\omega}_0$—it is usually a harmless approximation. This interval is depicted as a shaded area in Fig. 2b. Notice that $\tau_R^{-1} = J(\omega_0)/2$, so the *height* of the spectral density at the unperturbed frequency determines the *width* of the probing in-

---

[39]For comparison, in the 780 nm optical transition of $^{87}$Rb, $t_{NM}$ can be estimated to be about $46\tau_R$ [77], which for all practical purposes corresponds to a complete decay of the atom. To the best of the authors' knowledge, the non-exponential tail of spontaneous decay of any atom or subatomic particle has never been measured. The first experimental detection of the non-exponential decay at late times was reported in [98] for the luminescence decay of various organic pigments, which feature a rather short $t_{NM} \sim 9 \div 17\tau_R$.

[40]The boundaries of this interval are to be understood as somewhat blurred— an interval of length $4\tau_R^{-1}$ or $6\tau_R^{-1}$ could be equally taken.

terval. A large slope $|dJ(\omega)/d\omega|$ in the probing interval causes the true decay rate to depart from the Markovian prediction[41] $J(\omega_0)/2$. In the limiting case in which $J(\omega)$ displays extreme variation around $\omega_0$, such as narrow peaks or discontinuous jumps, the Markov approximation will be broken entirely. The spectral density in Fig. 2 shows an example of such sharp features in the high-frequency region. A peak with width $w$ smaller than the probing region (in other words, a peak whose height is much larger than its width) will induce an exponential decay at a rate $w \ll J(\omega_0)/2$—an example of a quantitative breakdown of the Markovian approximation, since the qualitative behavior of the function remains exponential but the exponent is not given by $\Delta_\infty$. A non-analytic behavior like a discontinuity will give rise to a qualitative breakdown, with $\tilde{f}(t)$ decaying as a power-law $t^{-1}$ rather than an exponential.

We notice that we are assuming that the system frequency $\omega_0$ is well within the band of available bath excitations, so that the decay rate $J(\omega_0)/2$ is non-vanishing. In other words, dissipation is provided by bath excitations that are resonant with the system transition, a condition that should be familiar from the derivation of Fermi's Golden Rule [99]. If $J(\omega_0) = 0$, dissipation will be provided by the nearest region with finite $J(\omega)$, i.e., by frequencies close to a band edge. In the next paragraph we will see that band edges induce non-Markovian dynamics. Physically relevant examples of this situation will be provided in sections 4.5.1 and 4.5.2.

**Non-Markovian behavior at late times**   After a crossover region at times around $t_{\mathrm{NM}}$ (the oscillating region in Fig. 2a) the function $\tilde{f}(t)$ decays more slowly than an exponential. This late-time regime is entirely controlled by the behavior of the spectral density at its lower edge $\omega = 0$—see Fig. 2b. In the common case of an algebraic behavior $J(\omega) \sim \omega^\alpha$ (with $\alpha = 1$ being the so-called Ohmic case [85]), the decay of the function is algebraic as well, $|\tilde{f}(t)| \sim t^{-1-\alpha}$. The existence of a non-exponential regime at late times might seem surprising at first glance, but it is a well-established phenomenon in the study of the decay of metastable states [80, 81, 100]. Indeed, it can be shown that unitary evolution prevents any state amplitude from decaying exactly as an exponential, both for closed [81] and open [101] systems. In the former case, the proof is rather direct [81]. Let us assume that the initial state $|\psi_0\rangle$ is unstable (i.e., not an eigenstate of the Hamiltonian), and that it evolves under the unitary dynamics $U(t) = \exp(-iHt/\hbar)$. Then, we can decompose the time-evolved state $|\psi_t\rangle = U(t)|\psi_0\rangle$ as $|\psi_t\rangle = A(t)|\psi_0\rangle + |\psi_\perp(t)\rangle$ in terms of the original state $|\psi_0\rangle$ and a un-normalized state $|\psi_\perp(t)\rangle$ which is orthogonal to the initial state and describes the decay products. Then, the amplitude $A(t)$ quantifies the survival probability of the initial state, and corresponds to $f(t)$ (compare with equation (21)). If the decay was Markovian, we would expect $A(t) \propto e^{-\gamma t}$. If we express $U(t)|\psi_0\rangle = U(t-\tau)U(\tau)|\psi_0\rangle$ and replace the above decomposition on both sides, we obtain an equation for $A(t)$:

$$A(t) = A(t-\tau)A(\tau) + \langle \psi_0 | U(t-\tau) | \psi_\perp(\tau) \rangle \ . \tag{33}$$

If the last term was vanishing, then the equation $A(t) = A(t-\tau)A(\tau)$ would indeed be solved by an exponential function. However, it is generally nonzero, unless the initial state is an eigenstate of $H$, so that $A(t) = e^{-i\varepsilon_0 t/\hbar}$ is an oscillating exponential. In all other cases in which the initial state is not stationary, the above equation shows that the decay cannot be purely exponential, because of the presence of the last term. The latter can give an intuition on why the decay is not exponential, since it is the amplitude for forming back the initial state after the decay products $|\psi_\perp(\tau)\rangle$ were formed at the intermediate time $\tau$—a non-Markovian process. We emphasize that nowhere in the above derivation we made assumptions on the size of the system or bath, so this mechanism is not a finite-size effect—in fact, it generically occurs even

---

[41]Here we focus on the decay rate, but the Lamb shift is affected by such non-Markovian effects, too.

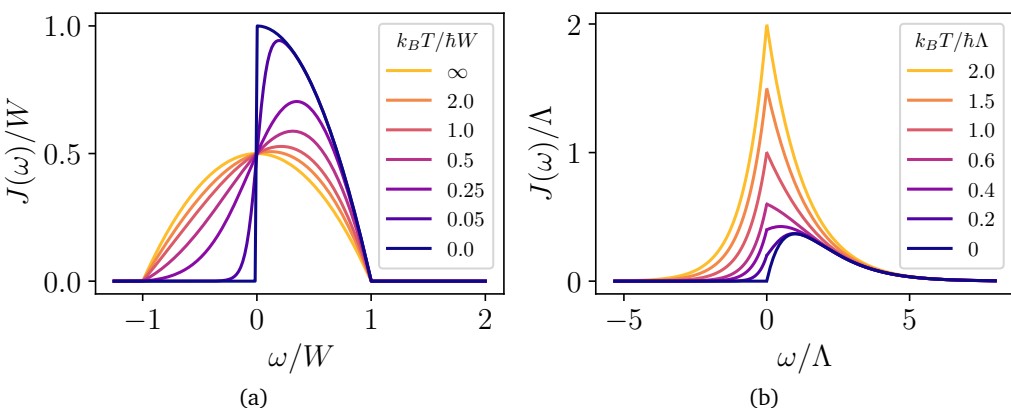

Figure 3: Typical behavior of the spectral density $J(\omega)$ of the bath as a function of temperature. (a) Case of a spin-1/2 or fermionic bath with a half-bandwidth $W$, given by equation (34) with $\rho(\omega) = 1 - \omega^2/W^2$. This bath becomes generally "more Markovian" as the temperature increases, since the zero-temperature discontinuity at $\omega = 0$ is smeared out. (a) Case of a bosonic bath with an Ohmic spectral function $\rho(\omega) \propto \omega e^{-\omega/\Lambda}$, where $\Lambda$ is a high-frequency cutoff. The non-analytic behavior at $\omega = 0$ changes nature at $T > 0$ but still persists, and the accuracy of the Markovian approximation is non-monotonic in temperature.

for a continuous bath[42]. As the argument above indicates, the loss of Markovian behavior is of quantum-mechanical origin.

The solution to the toy model (and similar analyses of unstable systems [80, 81, 98, 100, 101]) indicates that the decay is generally controlled only by the spectral density at the edges of the bath spectrum (which, in our case, is $\omega \to 0^+$). This observation suggests that the non-Markovian behavior at late times is caused by the excitation of the modes in the bath residing at the edges of the spectrum (or, more in general, by points in the bath spectrum at which the spectral density has a singular behavior). These modes are the ones responsible for the power-law tails of the bath correlation functions that we discussed in 4.3.1, and thus are the main source of "memory" in the dynamics of the system. Their effect can only be observed at late times, since they are off-resonant with respect to the main transition of the system and thus scarcely populated—thus explaining why the crossover time grows with $\omega_0$ in the toy model. Similarly, the effect of these edge modes is suppressed if their density is low ($J(\omega) \sim \omega^\alpha$ with large $\alpha$ at $\omega \to 0^+$). We will see in 4.5.1 and 4.5.2 that there are physically interesting situations in which the system probes the bath spectrum precisely at the edge, resulting in a strongly non-Markovian dynamics.

### 4.4.4  Application: the effect of temperature

In this section, we will apply the conclusions from the toy model to two physically motivated spectral densities, one fermionic and one bosonic in character. In doing this, the effect of the bath temperature will be presented.

The main message that we want to highlight here is that *the existence of a Markovian regime is not determined just by the properties of the bath but also by the frequencies at which the bath spectrum is probed*. In this sense, the only bath that is intrinsically Markovian is the idealized bath with a flat spectral density, for which $G(t) \propto \delta(t)$. Any realistic bath will

---

[42]Indeed, the bath in the toy model (or, equivalently, in the model (20) of spontaneous decay) has been assumed to be continuous, since we are using a continuous spectral density.

have more structure, such as band edges and non-analytic behavior caused by the statistical occupation functions, and the (non-)Markovian nature of its effect on the system will depend on the transition frequency of the latter as well. See, for instance, the discussion in section III.E of [91].

In the preceding discussion on the Markovian approximation, the role of the bath temperature has remained implicit. It determines the shape of the Fourier transform $J(\omega)$ of the bath correlation functions and, hence, in the language of the toy model, the range of frequencies that the system can "probe". In general, a higher temperature makes more transitions in the bath available to the system, so the support of $J(\omega)$ grows in size, and roughly speaking, this usually makes the bath more Markovian. This statement, although widespread [5, 91, 102], should be considered with care and always checked with the model at hand.

**Fermionic baths**   Let us consider the example of a spin-1/2 bath. As it will become clear momentarily, this example will be our prototype of a fermionic bath. We take $H_B = \sum_\lambda \frac{\hbar \omega_\lambda}{2} \sigma_\lambda^z$ and coupling $B = \sum_\lambda g_\lambda \sigma_\lambda^x$, and we consider the thermal state $\rho_B = e^{-\beta H_B}/Z_B$. While the following analysis is independent of the actual shape of the system Hamiltonian, it can be helpful to have in mind a two-level system with frequency $\omega_0$, $H_S = \omega_0 \sigma^z/2$, coupled to the bath through $A = \sigma^x$. Using $\sigma_\lambda(t) = \sigma_\lambda^- e^{-i\omega_\lambda t} + \sigma_\lambda^+ e^{i\omega_\lambda t}$ ($\sigma_\lambda^\pm$ being the raising and lowering spin operators, which in this case are also the eigenoperators of $H_B$) one can compute the correlation function $C(t) = \text{tr}[B(t)B(0)\rho_B]$. Taking the Fourier transform of the latter, we obtain the spectral density

$$J(\omega) = \frac{1}{4}\rho(\omega)[1 - F(\omega)] , \tag{34}$$

where $\rho(\omega) = \hbar^{-1} \sum_\lambda g_\lambda^2 [\delta(\omega - \omega_\lambda) + \delta(\omega + \omega_\lambda)]$ is the spectral function[43] containing the information on all the available excitation energies, while $F(\omega) = (e^{\beta \hbar \omega} + 1)^{-1}$ is the Fermi-Dirac distribution function (at zero chemical potential) accounting for the occupation of the spin states. The appearance of the Fermi-Dirac distribution is to be expected, since in a spin-1/2 bath each spin can be excited only once, analogously to a fermionic system in which states can be occupied by one fermion only. We stress that the "fermionic" behavior of the spectral density is not just an intrinsic property of the bath (i.e., the shape of $H_B$), but also of the type of coupling. Indeed, the Fermi-Dirac function appears because the coupling operator $B$ is "fermionic", namely, it adds or removes only one spin excitation.

We observe in equation (34) the common behavior that the spectral information contained in $\rho(\omega)$ can be disentangled from the statistical information contained in $F(\omega)$. It is the temperature-independent function $\rho(\omega)$ that is often called spectral function in the literature. Let us assume that the bath frequencies $\omega_\lambda$ lie in the interval $[-W, W]$, so that $\rho(\omega)$ will vanish for frequencies $|\omega| > W$. The behavior close to the edges of the band can usually be taken to be algebraic, $\rho(\omega) \sim (\omega \pm W)^\alpha$ for $\omega \to \mp W$, with $\alpha \geq 0$. The form of equation (34) for increasing temperatures is shown in Fig. 3a. We can observe that the Fermi-Dirac function acts as a statistical "filter" superimposed onto the bath spectrum, suppressing the contributions at negative frequencies.

As we discussed in section 4.4, the requirement for the existence of a Markovian time window is that the probing region around the system's transition frequency $\omega_0$ should be as far as possible from any sharp features of $J(\omega)$. In a fermionic bath such as in the present

---

[43]We are assuming that the frequencies $\omega_\lambda$ are densely distributed as the number of spins $N$ increases, and that $g_\lambda \propto N^{-1/2}$, so that there exists a meaningful thermodynamic limit for $N \to \infty$ in which $\rho(\omega)$ converges to a continuous function. This situation is quite different from the ones typically assumed when studying decoherence caused by a spin bath [103–105], i.e., with $H_I = \sigma^z \sum_\lambda g_\lambda \sigma_\lambda^z$. In these models, a meaningful large-size limit is achieved using the central limit theorem. The resulting dynamics is that of a strongly coupled problem, with the coherences of the system displaying a Gaussian decay at early times followed by a power-law tail.

examples, such non-analytic features can come either from $\rho(\omega)$ or from the Fermi function $F(\omega)$. Under the minimal assumption that the bath is "unstructured", the only sharp features in $\rho(\omega)$ occur at the band edges, while the statistical factor $F(\omega)$ may induce strong variations around $\omega = 0$ (the "Fermi energy"). Indeed, we can observe in Fig. 3a (darker curve) that at zero temperature $J(\omega)$ jumps from 0 to a finite value at $\omega = 0$ because $1 - F(\omega) = \theta(\omega)$ (where $\theta$ is the Heaviside step function), and the effective bandwidth is reduced to $W$ only. As discussed in 4.3, $J(\omega)$ vanishes at negative frequencies, since a zero-temperature bath can only absorb energy from the system. For small temperature $k_B T/\hbar \ll W$, the discontinuity is smeared into a sharp increase in the region $|\omega| \lesssim k_B T/\hbar$—compare, e.g., to the $T = 0.05W$ curve in the figure. Then, we can expect that at low or vanishing temperatures there will be a Markovian regime of the dynamics only if[44] the probing region is far from $\omega = 0$ and from the upper band edge, namely $k_B T/\hbar \ll \omega_0 \ll W$.

Upon increasing the temperature, the increase of the Fermi-Dirac function around $\omega = 0$ becomes progressively milder—compare with the lighter-colored curves in Fig. 3a. Thus, the bath becomes more "Markovian", in the sense that the probing region can get closer to 0 without encountering any abrupt behavior, although the large derivative of $J(\omega)$ for $|\omega| \lesssim k_B T/\hbar$ might induce strong non-Markovian corrections to the decay rate. In a finite-bandwidth bath, the ideal limit of infinite temperature is well-defined. In this limit, the Fermi-Dirac function becomes a constant, $F(\omega) \to \frac{1}{2}$, and the full band of excitations $-W < \omega < W$ becomes available. This limit is "as Markovian as possible" for the bath under consideration, since any system frequency that is sufficiently far from the band edges will probe a smoothly varying $J(\omega)$. However, notice that if $\omega_0$ is comparable to $W$, the presence of the edge will induce non-Markovian effects for any temperature. We will discuss a similar scenario in section 4.5.2. Let us stress that, in this example, we are assuming that the bandwidth $W$ is large with respect to the Markovian decay rate $\tau_R^{-1} = J(\omega_0)/2$. In the opposite scenario $W \lesssim \tau_R^{-1}$, the dynamics will be non-Markovian at any temperature.

The previous analysis is essentially applicable also for a noninteracting fermionic reservoir ($H_B = \sum_\lambda (\hbar\omega_\lambda - \mu) c_\lambda^\dagger c_\lambda$) that exchanges particles with a fermionic system, e.g., $H_I = \sum_{j\lambda} (V_{i\lambda} d_j^\dagger c_\lambda + \text{H.c.})$, with $d_j$ being the annihilation operators of the modes of the system. The interplay of band edges and Fermi-Dirac functions provide the spectral densities[45] of this model with a larger set of non-analytic points with a nontrivial dependence on temperature and chemical potential $\mu$. Nevertheless, in complete analogy with the spin bath, increasing the temperature smears out the discontinuous behavior of the Fermi-Dirac functions, leaving only milder band edge effects and thus allowing for a Markovian behavior for a larger range of system frequencies, i.e., without the need of fine tuning in $\omega_0$ (assuming a sufficient bandwidth $W \gg \tau_R^{-1}$).

**Bosonic baths**  In many cases, the environment can be modeled by a set of noninteracting harmonic oscillators $H_B = \sum_\lambda \hbar\omega_\lambda b_\lambda^\dagger b_\lambda$, linearly coupled to the system via $B = \sum_\lambda (g_\lambda b_\lambda + g_\lambda^* b_\lambda^\dagger)$, for certain frequencies $\omega_\lambda$ and couplings $g_\lambda$. This is the case for the electromagnetic field and for phonons in solids, but it can be a good approximation even for interacting environments if the coupling to the system is so weak that the bath is not perturbed beyond the linear response regime in which its excitations can be well described by small oscillations

---

[44]An astute reader will have noticed that the discontinuity at $T = 0$ will cause a very slow decay of $G(t) \sim t^{-1}$ that violates the Davies condition mentioned in 4.3. However, the latter is not a necessary condition for Lindblad to apply, and there still might be a (possibly short) Markovian regime, according to the toy model analysis. A preliminary calculation based on [90] (as well as the statements contained in [86]) seems to support this conclusion.

[45]Re-writing $H_I$ in terms of Hermitian operators introduces two $B_\alpha$, involving the Majorana fermions $V_{i\lambda} c_\lambda + V_{i\lambda}^* c_\lambda^\dagger$ and $i(V_{i\lambda} c_\lambda - V_{i\lambda}^* c_\lambda^\dagger)$.

around an equilibrium configuration[46] [85, 91, 107]. As in the previous case, for concreteness the reader can picture the system to be a simple spin $H_S = \omega_0 \sigma^z/2$, with coupling operator $A = \sigma^x$ to the bath, i.e., the spin-boson model [85]. An explicit calculation shows that

$$
J(\omega) = \begin{cases} \rho(\omega)[1 + B(\omega)] & \omega \geq 0 \,, \\ \rho(-\omega)B(-\omega) & \omega < 0 \,, \end{cases} \tag{35}
$$

where $B(\omega) = (e^{\beta\hbar\omega} - 1)^{-1}$ is the Bose-Einstein distribution and we introduced the spectral function $\rho(\omega) \equiv 2\pi/\hbar \sum_\lambda |g_\lambda|^2 \delta(\omega - \omega_\lambda)$, which is simply $J(\omega)$ at zero temperature. In the previous equation, we have made use of the property that $\rho(\omega) = 0$ for $\omega \leq 0$ because the bath has a ground state only if all frequencies $\omega_\lambda$ are positive (otherwise, condensing an arbitrary number of bosons on a negative frequency would yield arbitrarily low energy). Thus, $\omega = 0$ is the only band edge. The behavior of $J(\omega)$ for the common case of an Ohmic spectral function $\rho(\omega) \propto \omega e^{-\omega/\Lambda}\theta(\omega)$ ($\Lambda$ being a high-frequency cutoff) is shown in Fig. 3b. Contrary to spin and fermionic bath examples, here $J(\omega)$ coincides with $\rho(\omega)$ only at zero temperature (darkest curve in the Fig.). The only non-analytic point is the band edge at $\omega = 0$. For increasing temperatures, one can see that the overall bandwidth increases, similarly to the previous example of a spin bath. However, $J(\omega)$ remains singular at $\omega = 0$, where its derivative has a discontinuous jump. For a super-Ohmic environment with $\rho(\omega) \sim \omega^\alpha$ for $\omega \to 0^+$ and integer $\alpha \geq 2$, the discontinuity would occur in a higher derivative[47]. On the basis of the discussion in the previous section, the most conservative condition for Markovianity would be that the system should probe the spectral density far from $\omega = 0$. Indeed, in 4.5.1, we will present an example of non-Markovian behavior, which can be ascribed to a system probing an Ohmic environment exactly at $\omega_0 = 0$. In general, the persistence of singular behavior at $\omega = 0$ for all temperatures makes the relationship between temperature and Markovianity less intuitive for bosons. For instance, in the case of the damped harmonic oscillator (i.e., a harmonic oscillator in a bath of oscillators) considered in [30], the non-Markovian contributions to the dynamics at times larger than $\tau_B$ are non-monotonic with temperature, with the smallest deviations at exactly zero temperature, rapidly increasing at a finite temperature before decreasing again at large temperatures. This example shows how the issue of non-Markovian effects can be subtle, and their presence should be verified on a case-by-case basis.

## 4.5  Examples: The breakdown of the Born-Markov approximation

In this section, we will present two examples in which the Born and Markov approximations break down *qualitatively*, in the sense that perturbation theory in the system-bath coupling breaks down, leading to strong system-bath correlations. The simultaneous breaking of the Markov and Born approximations is the one encountered more often, and the one that generally points to the emergence of interesting physical effects. In principle, there can be also instances of *quantitative* breakdown of the Born-Markov approximation, corresponding to cases in which perturbation theory might still be valid but in which the second-order approach following from equation (6) yields quantitatively wrong results. Presumably, this would happen to models for which the Lindblad approach can be applied for small coupling when the latter is not small anymore, or when the bath correlation time $\tau_B$ becomes significant [108].

---

[46]There is also an interesting justification of this approximation in terms of a quantum version of the central limit theorem [40, 106]: if a bath operator $B$ coupled to the system is a sum of many independent contributions, then the it converges in an appropriate sense to $a_B + a_B^\dagger$, where $a_B$ is an annihilation operator of an effective bosonic mode, that depends on $B$.

[47]Whereas the case of a non-integer $\alpha$ would yield an intrinsic non-analytic behavior.

### 4.5.1 Kondo model *

The Kondo model is a textbook example of a system strongly correlated with its bath, in which perturbation theory breaks down. From the perspective of this review, it is a case of qualitative breakdown of the Born approximation, driven by non-Markovian effects, namely a diverging bath correlation time. This model exemplifies a typical breakdown of perturbation theory, that occurs when the bath possesses a significant density of *soft* modes, namely excitations with arbitrarily low energy (in the thermodynamic limit). Then, the effect of a system-bath coupling that is apparently small in comparison to the typical energy scales of the system is effectively amplified by the soft modes to yield strong system-bath correlations that violate the Born approximation[48].

The Kondo model describes a spin-1/2 impurity interacting with the local spin of a bath of noninteracting fermions[49] [109] (for its dissipative version see Refs. [110, 111]), which is usually written as

$$H = \lambda H_I + H_B = J\boldsymbol{S} \cdot \boldsymbol{s}(\boldsymbol{0}) + \sum_{\boldsymbol{p},\sigma} (\varepsilon_{\boldsymbol{p}} - \mu) c_{\boldsymbol{p}\sigma}^{\dagger} c_{\boldsymbol{p}\sigma} , \tag{36}$$

where $\boldsymbol{S}$ is the dimensionless spin operator of the magnetic impurity, and $s_\alpha(\boldsymbol{0})$ the spin density by the bath fermions at $\boldsymbol{x} = \boldsymbol{0}$, the position of the impurity. We use a boldface font to indicate vectors, and $\cdot$ to represent the usual scalar product. The term $H_B$ describes noninteracting spinful fermions with dispersion $\varepsilon_{\boldsymbol{p}}$ ($\boldsymbol{p}$ being their momentum) and chemical potential $\mu$, while the impurity spin does not have any dynamics on its own. In terms of the fermion operators, the spin density reads $s_\alpha(\boldsymbol{0}) = \frac{1}{2V} \sum_{\boldsymbol{p},\boldsymbol{k},\sigma,\tau} (\sigma^\alpha)_{\sigma\tau} c_{\boldsymbol{p}\sigma}^{\dagger} c_{\boldsymbol{k}\tau}$, where $\alpha \in \{x, y, z\}$, $\sigma^\alpha$ are the Pauli matrices and $V$ is the (large) volume of the bath. In this paragraph, we limit ourselves to a qualitative discussion of Kondo physics, while the details of the calculation are presented in appendix C. In the notation of the previous paragraphs, the coupling operators are $A_\alpha = S_\alpha$ for the impurity and $B_\alpha = s_\alpha(\boldsymbol{0})$ for the bath. The coupling strength $\lambda = J > 0$ describes an antiferromagnetic interaction which promotes the two spins $\boldsymbol{S}$ and $\boldsymbol{s}(\boldsymbol{0})$ to be oriented in opposite directions[50]. This coupling is the natural perturbative parameter of the problem, in the sense that we are interested in the situation in which $\rho_F J \ll 1$, where $\rho_F$ is the density of fermionic single-particle states per unit energy at the chemical potential. Roughly speaking, $\rho_F$ is proportional to the inverse of the bandwidth of the bath, so $\rho_F J$ is the ratio of the typical energy scales of interaction and bath.

The Kondo model has been extensively studied and is exactly solvable [109, 112]. The dynamics of the impurity spin are deceptively simple: it relaxes exponentially to zero with a temperature-dependent rate $\gamma_K(T)$. The zero-temperature limit of this rate defines a fundamental energy scale of the model known as Kondo temperature, $k_B T_K \equiv \hbar\gamma_K(T \to 0) \sim (2\rho_F J)^{1/2} \exp\left(-\frac{1}{2\rho_F J}\right)$. This rather mundane behavior hides the reality that the Kondo model describes a strongly correlated system for $T \lesssim T_K$, characterized by the formation of a singlet state between the impurity and the bath—in other words, entanglement between the system and bath cannot be neglected. The physical origin of the strong correlation lies in the property that a noninteracting fermionic bath is gapless, meaning that it hosts excitations (particle-hole pairs) of arbitrarily small energy. The abundance of low-energy excitations gives rise to a spectral density that vanishes slowly at low frequency, $J(\omega) \sim \omega$ (an Ohmic spectral density)—see

---

[48]Compare this behavior to time-independent perturbation theory: a perturbation to a Hamiltonian is small only if its matrix elements are small with respect to the energy differences of the eigenstates they connect. If the energy differences (i.e., the excitation energies) can be arbitrarily small, perturbation theory cannot be applied.

[49]Readers with less familiarity with solid state physics might think in terms of a spin-boson model for an Ohmic bath in the limit of zero tunneling, thanks to a well-known mapping between the two models [85].

[50]We notice that, although the coupling between the spins describes an effectively magnetic interaction, its microscopic origin is electrostatic and lies in the strong Coulomb repulsion that two electrons residing on the impurity atom experience. See, for instance, [112]. We remark that in our notation $J$ has units of energy times volume.

Fig. 8. In the language of the toy model in the previous section (compare with Fig. 2b), the system has no dynamics on its own and so it probes the bath at zero frequency $\omega_0 = 0$, where the non-Markovian effects are most prominent[51]. From a real-time perspective, the Ohmic nature of the bath manifests as a slow decay of correlation functions, $\langle s_\alpha(\mathbf{0}, t) s_\alpha(\mathbf{0}, 0) \rangle \sim t^{-2}$, which effectively yields a diverging correlation time $\tau_B$—as mentioned in 4.3.1. The absence of intrinsic dynamics means that also the timescale $\tau_H$ introduced in 3.1 diverges, hence the parameter $\lambda \min(\tau_B, \tau_H)/\hbar \propto \rho_F J \min(\tau_B, \tau_H)$ controlling the perturbative expansion in the bath coupling (see 3) diverges as well—the nominally small coupling $\rho_F J$ is amplified by non-Markovian effects. In this sense, the Kondo model provides an example of the breakdown of the Born approximation, driven by non-Markovian effects.

If we nevertheless tried to approach the problem within the Lindblad framework (as shown in appendix C), we would obtain the equation (1) with jump operators $L_\alpha = S_\alpha$ and $H_S = 0$ (the Lamb shift is trivial in this case). Such an equation would yield the correct stationary state $\rho_S(t \to \infty) = \mathbb{1}/2$ (i.e., a Gibbs state $\rho_S \propto e^{-\beta H_S}$—in the absence of a magnetic field, $H_S = 0$, both spin states are equally populated[52]) while finding a completely different rate of the exponential relaxation, $\gamma_L(T) = \pi J^2 \rho_F^2 k_B T/\hbar$, for small enough temperature (with respect to the chemical potential). So, we find a qualitative discrepancy between the Lindblad equation, that predicts no spin decay at zero temperature, $\gamma_L(0) = 0$, and the exact dynamics, that predicts a decay even at zero temperature, with the rate $\gamma_K(0) = k_B T_K/\hbar$. As shown in the appendix C, using the Born master equation (7) improves the prediction only marginally, in the sense that the system relaxes to $\mathbb{1}/2$ but in a non-exponential fashion. This result confirms that we are dealing with a system strongly correlated with its environment, for which both Born and Markov approximations break down.

The Lindblad dynamics of the Kondo model is not completely worthless. Indeed, the decay rate $\gamma_L(T)$ quoted above coincides with the a well-known result by Korringa [113, 114], and it applies only for sufficiently large temperatures $T \gg T_K$. The linear dependence $\gamma_L(T) \propto T$ is observed in neutron scattering experiments for a large range of temperatures [109] above the Kondo regime. Physically, the non-Markovian effects inducing strong correlations are removed by the smoothening of the bath spectral density at high temperature, similarly to what we discussed in 4.4.4, so that the Lindblad treatment becomes applicable.

There are a few lessons to be learned from this example. Firstly, a blind application of the Lindblad treatment to a non-perturbative problem can result in a dynamics of the system that looks reasonable, despite being inaccurate. In the present example, the Lindblad master equation captures only the thermally activated part of the decay with $\gamma_L(T) \propto T$ because the processes governing the spin decay at $T = 0$ come into play only at the next order in the perturbative expansion in powers of $\rho_F J$ [115]. Indeed, neglecting relevant physical processes is always a risk when stopping at the second order in the Born approximation. In practice, their presence should be checked on a case-by-case basis, and this procedure can be simplified by some intuition regarding the behavior of the system.

In the case of Kondo, the third-order perturbation in $\rho_F J$ leads to a divergence in the relaxation rate $\gamma_L$ at low temperatures $T \ll T_K$ [109, 114, 115]. This indicates that the system is actually strongly interacting. Indeed, the true relaxation rate $k_B T_K/\hbar \propto \exp(-1/(2\rho_F J))$ cannot be expanded around $\rho_F J = 0$, and thus cannot be obtained by any finite-order perturbative approach.

---

[51]Introducing a magnetic field $H_S = -g\mu_B B_z S_z$ (where $g$ is an appropriate g-factor and $\mu_B$ is the Bohr magneton) would provide a finite $\omega_0 = g\mu_B B_z/\hbar$ and tame the non-Markovian effects. Indeed, a large field $g\mu_B B_z \gg k_B T_K$ restores Markovianity. The same happens for $T \gg T_K$, which has the effect of "flattening" the spectral density.

[52]Intriguingly, the maximally mixed state is also the partial trace of the spin-singlet state that is the exact ground state of the Kondo model. Our analysis is not sufficient to understand whether this is just a coincidence.

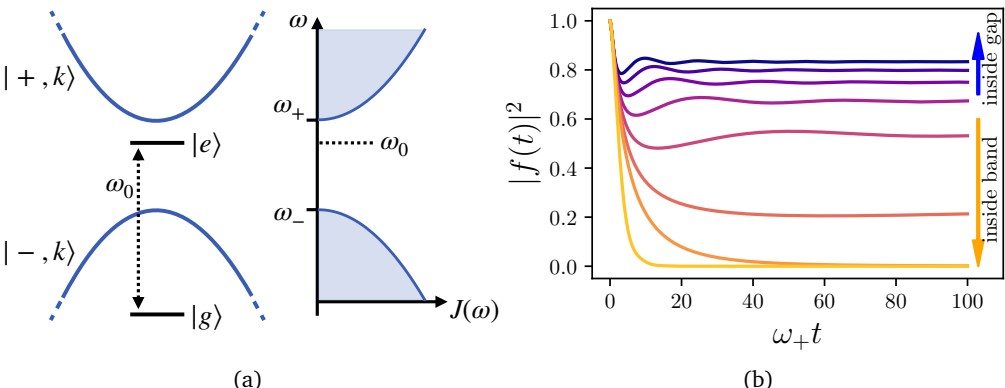

Figure 4: (a) Cartoon of a two-level atom embedded in a photonic crystal, with singularity in the density of states. (b) Non-Markovian spontaneous decay of the atom, calculated with the toy model (19). In the upper curves, the transition energy $\omega_0$ lies within the gap, and the atom cannot fully decay. As $\omega_0$ increases above $\omega_+$ (lower curves), the decay is more pronounced, but only for $\omega_0 \gg \omega_+$ (lowest curve), a Markovian time window for the decay is recovered. The plots have been produced by numerical integration of equation (19) with $J(\omega) = 2\pi\eta(\omega - \omega_+)^{1/2} \exp(-(\omega - \omega_+)/\Lambda)\theta(\omega - \omega_+)$, and parameters $\eta = 0.1\omega_+^{1/2}$, $\Lambda = 2\omega_+$. The curves shown correspond to $\omega_0/\omega_+ = 0.7, 0.8, 0.9, 1, 1.1, 1.2, 1.3, 2$, from top to bottom.

### 4.5.2   Spontaneous emission with structured bath spectra *

In the previous example, the fermionic bath had a rather ordinary Ohmic spectral density, but the Born approximation broke down because the system was "probing" the spectral density around zero frequency, which produced the maximal non-Markovian effects. The resulting behavior of observables was still qualitatively similar to the Markovian case, with a simple exponential decay of the magnetization. In this example, we will explore a more extreme case in which the bath spectral density has a singular behavior, leading to strongly non-Markovian dynamics.

In free space, photons have the usual linear dispersion relation $\omega_{\boldsymbol{k}} = c|\boldsymbol{k}|$, which leads to a super-Ohmic spectral density $J(\omega) \propto \omega^3$. The lack of structure in this spectral density (as well as the smallness of the coupling, namely the fine structure constant) guarantees that an excited two-level atom in contact with the EM vacuum will undergo a spontaneous emission that is effectively Markovian for all measurable times, as long as the frequency $\omega_0$ of the atomic transition is finite. However, there are experimentally relevant conditions under which the photon dispersion is significantly altered [116].

**Cavities**   The simplest example is that of spontaneous emission in a cavity, that is, when the EM field is confined between two closely spaced mirrors [117]. Within the cavity, only certain wavelengths of light are allowed, and the spectral density of the EM field is a series of broadened peaks, whose width is set by the rate at which photons escape from the cavity. Then, according to our analysis of the toy model, if an atomic transition occurs close to one of such peaks, we expect non-Markovian effects, depending on the sharpness of the peak. If the peak is well resolved (the cavity is "high-finesse"—the spacing between the peaks is larger than their width), so that the Markovian decay rate $\tau_R^{-1} = J(\omega_0)/2$ is comparable or smaller than the width, we expect strong non-Markovian effects. Indeed, experiments show

the so-called vacuum Rabi oscillations: the population in the atomic excited states undergoes damped oscillations before decaying completely. Physically, a photon is repeatedly emitted and reabsorbed by the atom because it is "trapped" in the cavity for timescales longer than the spontaneous emission lifetime. In such a situation, the cavity modes of the EM field cannot be treated as a bath to the atom, and they need to be treated as part of the system, with the EM field outside the cavity playing the role of the reservoir [6]. Only in the case of a "bad cavity" with very broad peaks, the condition of smoothness of the spectral density $J(\omega)$ is recovered, and cavity modes can be modeled as part of the bath [118–120].

**Photonic crystals**   There are situations in which the dispersion of the EM field is modified even more radically, to the point of rendering the spectral density singular. A prominent example of this phenomenon is provided by photonic crystals, which are dielectric materials whose refractive index varies periodically in space on a scale comparable to the wavelength of light [121–123]. In the latter materials, in complete analogy with Bloch states of electrons in a crystal, the photon dispersion reorganizes into bands separated by energy gaps in which no light propagation is possible. These gaps open at specific momenta of the order of $|\mathbf{k}_0| = \pi/a$, where $a$ is the lattice constant of the crystal, and close to such gaps, the dispersion becomes parabolic—photons at momenta around $\mathbf{k}_0$ behave as massive particles. The photon dispersion around the gap momentum is split into two bands, $\omega_{\mathbf{k}}^{\pm}$, which can be approximated with $\omega_{\mathbf{k}}^{\pm} \sim \omega_{\pm} \pm A_{\pm}(\mathbf{k} - \mathbf{k}_0)^2$. This quadratic dispersion gives rise to a singular spectral density at the edges $\omega_{\pm}$ of the gap, $J(\omega) \sim (\omega - \omega_+)^{1/2}$ for $\omega > \omega_+$ and $J(\omega) \sim (\omega_- - \omega)^{1/2}$ for $\omega < \omega_-$[53]. This situation is depicted in Fig. 4a, in which we also show a two-level atom embedded in the photonic crystal, such that its transition frequency falls within the frequency gap $\omega_- < \omega_0 < \omega_+$. In this scenario, the atom cannot decay by emitting photons into resonant modes, and emission has to proceed via nonresonant ones at the band edges $\omega \approx \omega_{\pm}$. As we have seen in 4.4, the latter always yields non-Markovian dynamics. This is indeed what has been found, at first in theory, [125] and then in experiments [126–128].

As we mentioned in 4.4, the theoretical description of spontaneous emission in a two-level atom can be based under broad assumptions on the toy model (19), with $f(t)$ being the probability amplitude to find the atom in the excited state. Integrating equation (19) with a kernel $K(t)$ corresponding to the upper band $\omega_{\mathbf{k}}^+$, we obtain the dynamics of decay from the excited level shown in Fig. 4b. The three upper curves correspond to $\omega_0 < \omega_+$, and we observe that decay is hindered, in the sense that the atom remains mostly excited up to arbitrarily long times. As $\omega_0$ increases and enters the photonic band, the availability of resonant modes makes it decay faster, but there remains a significant late-time population of the excited state even when $\omega_0$ is well within the band. Besides the incomplete decay, another signature of non-Markovian decay in this regime is that the approach to the asymptotic value of $f(t)$ is non-exponential. Indeed, as we have seen in 4.4, the presence of a band edge close to the "probing region" around $\omega_0$ causes power-law decay. Only for $\omega_0$ deep enough into the band (lowest curve in Fig. 4b) we recover the more familiar scenario in which the atom can decay completely and there is a sufficiently large Markovian window of times for which $|f(t)|^2 \sim e^{-2t/\tau_R}$.

By comparison, none of the features of the exact dynamics described above can be reproduced by the Lindblad equation. In the latter description, the object to be compared to $|f(t)|^2$ is the population of the system density matrix $\rho_S(t)$ in the excited state $|e\rangle$, $\rho_{ee}^S(t)$. The discrepancy between Lindblad dynamics and the exact solution is particularly stark when the transition frequency of the atom is within the gap, $\omega_0 < \omega_+$, since in that case the former predicts no decay at all, and the atom should stay in the excited state at all times, $\rho_{ee}^S(t) \equiv 1$.

---

[53]For readers familiar with solid-state jargon, these singularities caused by stationary points in the dispersion are simply van Hove singularities [124].

Thus, the small decrease of $|f(t)|^2$ is completely missed by the Markovian approximation. When $\omega_0$ enters the photonic band, the Lindblad equation predicts $\rho_{ee}^S(t) = e^{-2t/\tau_R}$ with a finite decay rate $\tau_R^{-1} = J(\omega_0)/2$. As long as $\omega_0$ is comparable to the band edge, this prediction is qualitatively violated, since the true population does not decay to zero, nor does it approach its asymptotic value exponentially. In this regime, the Markov approximation cannot be applied because it is not true that the spectral density $J(\omega)$ has negligible variation around $\omega_0$, due to the square-root singularity of $J(\omega)$ at $\omega_+$. However, the differences with the Lindblad equation have a deeper physical origin, as the incomplete decay of the excited state is the consequence of the emergence of a bound state involving both the atom and the photon modes [123, 129]—in other words, the atom does not decay to its own ground state, but rather the whole atom-plus-field system decays into the bound state. This bound state has energy within the gap[54], and it is an exact eigenstate of the full system, i.e., it does not decay. Intuitively speaking, it can form because close to the band edge the photons have a vanishing group velocity $\nabla_{\boldsymbol{k}}\omega_{\boldsymbol{k}}^+$ and they can linger on close to the atom for long enough that their interaction with the atom is effectively amplified. The emergence of a bound state is a non-Markovian, non-perturbative phenomenon, and as such, it cannot be reproduced by the Lindblad master equation. In particular, the formation of a bound state between system and bath signals the breakdown of the Born approximation, similarly to the previous example of the Kondo model.

## 5   The Rotating Wave Approximation

- The rotating-wave approximation (RWA) is generally needed to convert the Red-field equation (11) into the Lindblad equation (1) and thus to ensure that positivity of $\rho_S(t)$ is preserved.

- One has to look at the frequencies $\Omega$ of the *transitions* that the bath induces in the system and their differences $\Delta\Omega$, see Fig. 5. RWA is valid if the system contains only perfectly degenerate transitions $\Delta\Omega = 0$ or far-detuned transitions $\Delta\Omega\tau_R \gg 1$.

- Nearly degenerate transitions $0 < \Delta\Omega\tau_R \lesssim 1$ require treatment beyond the Lindblad equation.

### 5.1   General discussion

Following the form of the Born-Markov approximation introduced in Eq. (11) of Sec. 4.1, one arrives at the Redfield master equation

$$
\begin{aligned}
\frac{\mathrm{d}}{\mathrm{d}t}\rho_S(t) = \sum_{\Omega,\Omega'}\sum_{\alpha\beta}\Big\{ & e^{-i(\Omega-\Omega')t}\Gamma_{\beta\alpha}(\Omega)\Big[\mathcal{A}_\alpha(\Omega)\rho_S(t)\mathcal{A}_\beta^\dagger(\Omega') - \mathcal{A}_\beta^\dagger(\Omega')\mathcal{A}_\alpha(\Omega)\rho_S(t)\Big] \\
& + e^{i(\Omega-\Omega')t}\Gamma_{\beta\alpha}^*(\Omega)\Big[\mathcal{A}_\alpha(\Omega')\rho_S(t)\mathcal{A}_\beta^\dagger(\Omega) - \rho_S(t)\mathcal{A}_\beta^\dagger(\Omega)\mathcal{A}_\alpha(\Omega')\Big]\Big\},
\end{aligned}
\tag{37}
$$

---

[54]We briefly discuss these bound states when we present the full solution to the toy model (19) in appendix B. Their energies can be found solving equation (B.10), which requires that it can only occur where the spectral density vanishes. We mention that the present example of a photonic band gap provides an interesting case in which the difference between the bare transition energy $\omega_0$ and the one dressed by the Lamb shift, $\omega_*$, can be be crucial, since the latter may end up in the gap while $\omega_0$ is in the band, and vice-versa [123].

which can be rearranged in a Lindblad-like form

$$\frac{\mathrm{d}}{\mathrm{d}t}\rho_S(t) = -\sum_{\Omega,\Omega'}\sum_{\alpha\beta} e^{-i(\Omega-\Omega')t}\frac{\Gamma_{\beta\alpha}(\Omega)-\Gamma_{\beta\alpha}^*(\Omega')}{2}\Big[\mathcal{A}_\beta^\dagger(\Omega')\mathcal{A}_\alpha(\Omega),\rho_S(t)\Big]$$

$$+\sum_{\Omega,\Omega'}\sum_{\alpha\beta} e^{-i(\Omega-\Omega')t}\frac{\Gamma_{\beta\alpha}(\Omega)+\Gamma_{\beta\alpha}^*(\Omega')}{2}\Big(2\mathcal{A}_\alpha(\Omega)\rho_S(t)\mathcal{A}_\beta^\dagger(\Omega')-\big\{\mathcal{A}_\beta^\dagger(\Omega')\mathcal{A}_\alpha(\Omega),\rho_S(t)\big\}\Big)\,.$$

$$(38)$$

The bath correlation functions can be explicitly written in terms of their real and imaginary components $\Gamma_{\alpha\beta}(\Omega) = \frac{1}{2}\gamma_{\alpha\beta}(\Omega)+iS_{\alpha\beta}(\Omega)$. We will see that in the Lindblad equation, the matrix $\gamma$ contains the information about the dissipation rates of different decay channels, and matrix $S$ encodes the effect of dissipation on the Hamiltonian dynamics. In the Redfield equation, both matrices contribute to the coherent and incoherent dynamics, leaving their role to the energy contributions blurred [94]. Notice that the commutator term in equation (38), which acts as a Hamiltonian-like contribution, couples eigenstates of $H$ with different energies thanks to the terms with $\Omega \neq \Omega'$. In other words, at this level of approximation, the coupling to the bath is able to alter the system's eigenstates[55].

The Redfield master equation, in its unaltered form, is commonly used to model open quantum systems [41, 130–137]. However, it raises concerns about the physical correctness of its solutions, as it does not guarantee the positivity of the density matrix. The property of positivity refers to the physically meaningful condition of the diagonal elements of the density matrix being positive and remaining positive throughout the time evolution. This maintains the interpretation of the density matrix elements as populations on the diagonal and coherences off-diagonally. In general, the terms that are responsible for the unphysical behavior are among the ones that explicitly oscillate in time in the interaction representation and are preceded by $e^{i(\Omega-\Omega')t}$ for $\Omega \neq \Omega'$ (these are the ones departing from the proven form of the most general CPT map, i.e., the Lindblad master equation [2, 16]). In terms of physical processes, the terms with oscillatory factors connect the dynamics of different coherences and those of coherences and populations [138, 139]. In the simplest example of a three-level system in a bosonic bath[56], it can be shown that selected coherences feeding the evolution of a particular density are the sources of positivity breaking [140]. What becomes apparent from this example is that only a subset of the oscillating terms can become negative, depending on the problem at hand. The rotating-wave approximation (RWA, also referred to as "secularization" in the literature) is a radical solution to this issue—in order to deal with physical dissipation channels that contribute disproportionately[57] (in comparison to the underlying physical system) to the master equation, we ignore them completely. As a result, we restore positivity but at the cost of losing access to physical phenomena such as coherent oscillations, which can be observed only through the coupling of coherences to populations [141, 142].

The mathematical justification for this step is the following. Suppose the frequency of these oscillating terms is significantly faster than the system's dissipative dynamics. In that case, on our timescales of interest, the oscillating terms will average to zero. Thus, we can neglect those terms by performing RWA. The result is the Lindblad master equation, which,

---

[55]Notice that an operator $A_\alpha$ might couple degenerate levels in the system, corresponding to transitions with $\Omega = 0$. In this case, the degenerate eigenstates will be altered even after applying RWA.

[56]In the case of a two-level system in a bosonic bath, the dynamics modeled by the Redfield equation remains positive [140].

[57]Note that the time-dependence of these terms is a result of the final Markovian approximation—their coefficients, in general, depart from the values in the Born master equation. The closer the system is to the fully Markovian dynamics, the smaller this variation.

going back to the Schrödinger picture, takes the form

$$
\begin{aligned}
\frac{\mathrm{d}}{\mathrm{d}t}\rho_S(t) = & -\frac{i}{\hbar}[H_S + H_{LS}, \rho_S(t)] \\
& + \sum_{\Omega}\sum_{\alpha\beta}\gamma_{\alpha\beta}(\Omega)\Big[\mathcal{A}_\beta(\Omega)\rho_S(t)\mathcal{A}_\alpha^\dagger(\Omega) - \frac{1}{2}\big\{\mathcal{A}_\alpha^\dagger(\Omega)\mathcal{A}_\beta(\Omega), \rho_S(t)\big\}\Big],
\end{aligned}
\tag{39}
$$

where the commutator contribution from the dissipation takes a Hamiltonian form, a so-called Lamb shift Hamiltonian $H_{LS}$. The name refers to the fact that it simply shifts the eigenstate spectrum of the system, as it commutes with the original system Hamiltonian due to jump operators being its eigenoperators, as discussed in section 4. The Lamb-shift Hamiltonian takes the explicit form

$$
H_{LS} = \hbar\sum_{\Omega}\sum_{\alpha\beta}S_{\alpha\beta}(\Omega)\mathcal{A}_\alpha^\dagger(\Omega)\mathcal{A}_\beta(\Omega).
\tag{40}
$$

As a result, the coherent dynamics are governed by $H_S' = H_S + H_{LS}$ rather than the bare system Hamiltonian. At this stage, we realize that the real part of the bath correlation functions $\gamma$ contributes purely to the incoherent dynamics and, conversely, their imaginary part $S$ only to the coherent dynamics. The above equation can be brought to the standard form (1) by diagonalizing the matrix $\gamma_{\alpha\beta}(\Omega) = \sum_a \gamma_a(\Omega)u_{a\alpha}(\Omega)u_{a\beta}^*(\Omega)$ and defining the new jump operators

$$
L_a(\Omega) \equiv \sum_\alpha u_{a\alpha}(\Omega)\mathcal{A}_\alpha(\Omega).
\tag{41}
$$

The crucial ingredient showing that equation (39) is of the Lindblad form is that the matrix $\gamma_{\alpha\beta}(\Omega)$ can be proven to be positive definite [1,5]—namely, its eigenvalues $\gamma_a(\Omega)$ define positive decay rates[58]. Notice that these rates coincide with the ones given by Fermi's Golden Rule [5,40], namely, they are determined by second-order perturbation theory in the system-bath coupling $\lambda$ (see introductory discussion in Sec. 2).

Throughout this review, we use the term RWA to refer to its standard definition: the omission of all time-oscillating terms in the Redfield master equation. Our discussion of the RWA's validity is based on this interpretation. However, a plethora of modified RWA approaches exists, which we discuss briefly in section 5.4. Bearing this in mind, we can comment more quantitatively on the conditions for the justifiability of the RWA. The typical timescale of the system's intrinsic evolution $\tau_S \sim |\Omega - \Omega'|^{-1}$ has to be smaller than the system's relaxation time, $\tau_S \ll \tau_R$. The hierarchy of the time scales is shown in Fig. 1. The applicability of RWA is thus dependent on the system's spectral properties, and in particular on the difference between the transition energies (rather than simply the energy levels [59]) linked by the jump operators. Based on this criterion, we can distinguish three types of systems, as shown in Fig. 5.

---

[58]Lindblad form master equation was proven to be the most general form of a Markovian CPT map if $\sum_a L_a^\dagger L_a$ is a bounded operator, and each dissipative channel has a time-independent coefficient [17,18]. Positivity is not always guaranteed for a Lindblad-type master equation with time-dependent coefficients [143]. Conversely, there exist Markovian CPT maps which are not of Lindblad form in the case of non-invertible maps [144,145].

[59]Although we emphasize that the differences between the transition energies dictate the internal system's dynamics, if $\Omega = 0$ is one of the possible transition frequencies, the condition $|\Omega - \Omega'|\tau_R \gg 1$ also implies that the individual transition frequencies must be large.

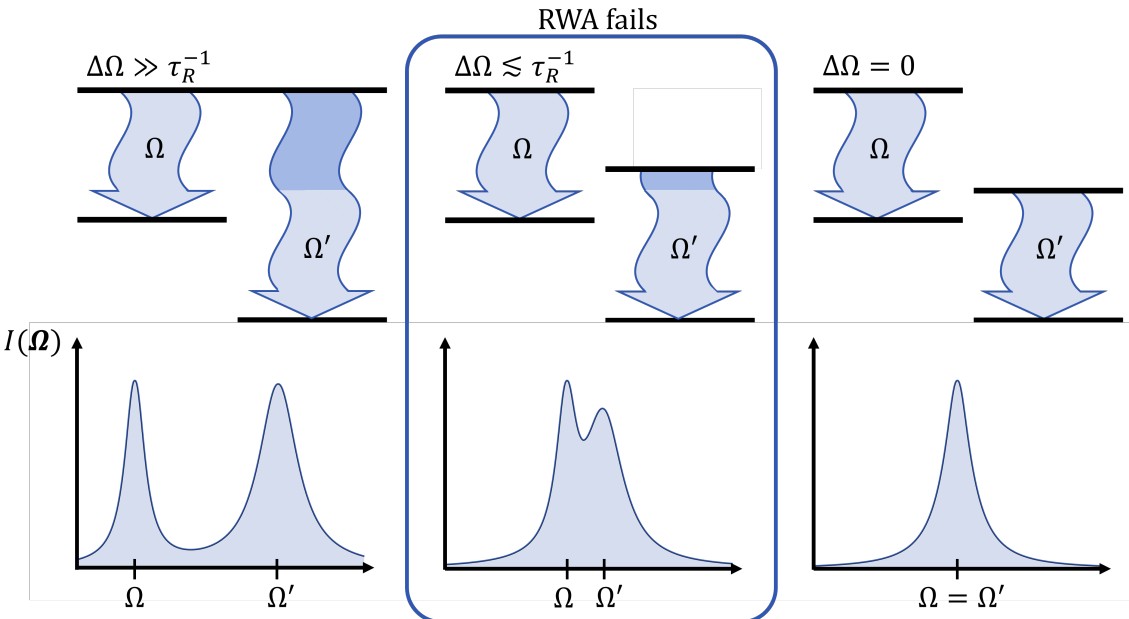

Figure 5: Cartoon representing different types of transitions in open quantum systems, categorized by their transition spectra. Upper Row: Example energy levels connected by the bath. The difference in their transition energies is defined as $\Delta\Omega = |\Omega' - \Omega|$. Lower Row: Corresponding emission spectra with intensity as a function of emission frequency $I(\mathbf{\Omega})$. The diagrams visualize the energy scales relevant in RWA, but in general do not correspond to the actual emission spectra measured in experiments, which can be non-Lorentzian due to interference effects [139, 146]. The columns of the figure represent, in order from left to right, spectra with highly non-degenerate, nearly degenerate, and degenerate transitions. For a nearly-degenerate transition spectrum, the RWA is not valid.

For fully degenerate transitions ($\Omega = \Omega'$), the oscillating prefactors in equation (37) are unity at all times. As a result, these terms directly yield a Lindblad form without invoking the RWA. In particular, it means that the terms $\mathcal{A}_\alpha(\Omega)\rho_S(t)\mathcal{A}_\beta^\dagger(\Omega)$ are in a Lindblad form even for $\alpha \neq \beta$, implying coupling between different dissipation channels. When the level-spacings differ significantly from each other (in comparison to the unperturbed system dynamics), $|\Omega - \Omega'|^{-1} \ll \tau_R$ and the separation of timescales straightforwardly applies. In this regime, only terms corresponding to identical transition frequencies are retained, while the cross terms $\mathcal{A}_\alpha(\Omega)\rho_S(t)\mathcal{A}_\beta^\dagger(\Omega')$ for $\alpha \neq \beta$ average to zero. Each dissipative channel thus evolves independently. The situation is more nuanced for nearly degenerate transitions with $\Omega \simeq \Omega'$. Here, the timescale of the system's internal dynamics $\tau_S$ becomes comparable to, or even exceeds, the relaxation time $\tau_R$, invalidating the RWA. In this case, the terms $\mathcal{A}_\alpha(\Omega)\rho_S(t)\mathcal{A}_\beta^\dagger(\Omega')$ with $\alpha \neq \beta$ are no longer dynamically suppressed, and their influence becomes comparable to that of the diagonal terms with $\alpha = \beta$. The standard Lindblad framework presented here cannot capture such contributions. To correctly include near-degenerate processes in the dynamics one must either forgo the RWA and employ the Redfield equation, or work within dedicated approaches, such as the ones described later in 5.4, that are able to produce a master equation in the Lindblad form by operating different approximations on equation (37).

There is a simple physical intuition for the three cases of transitions and their relationship to RWA, which we can borrow from AMO physics. Let us consider the system to be an excited atom placed in the EM vacuum. The following reasoning can be applied, at least conceptually, to more general scenarios. Each dissipative process identified by $\mathcal{A}_\alpha(\Omega)$ corresponds to the

emission[60] of a photon with frequency $\Omega > 0$. The measurement of the intensity of emitted photons yields spectra like the ones in the bottom row of Fig. 5. Each spectral line is broadened by an amount corresponding to the inverse lifetime of the transition, $\tau_R^{-1}(\Omega)$. Indeed, the coupling to the bath makes the excited eigenstates of the system unstable, and their energy is no longer well-defined. Then, from the perspective of the emitted photons, transitions are deemed nondegenerate when the corresponding emission lines are spectrally resolvable, i.e., when the frequency separation $\Delta\Omega = |\Omega' - \Omega|$ exceeds the linewidth $\tau_R^{-1}$, as visualized in Fig. 5. Conversely, when $\Delta\Omega \lesssim \tau_R^{-1}$, the spectral overlap prevents resolution of individual lines, which is interpreted as a regime where coherent interference between the transitions becomes significant. A possible experimental signature of this interference is a deformation of the emission line with respect to the usual Lorentzian shape [146–148].

It is worth noting that there is a special condition in which the RWA is not needed. Indeed, it is possible to bring equation (37) to the Lindblad form (1) with positive decay rates by defining new jump operators $L_a = \sum_{\alpha,\Omega} u_{a\alpha} A_\alpha(\Omega)$ (note the summation over frequencies $\Omega$) if the $\Gamma_{\alpha\beta}(\Omega)$ coefficients do not depend on $\Omega$ in the first place[61] In the jargon of section 4, this condition requires a bath spectral density $J(\omega)$ which is perfectly flat for all transition frequencies. Indeed, the Lindblad equation becomes exact for such a constant $J(\omega)$ [24,149]. From this perspective, the RWA removes dissipative processes sensitive to the frequency variation of $J(\omega)$. We have already encountered $dJ(\omega)/d\omega$ as controlling the applicability of the Markov approximation in section 4. We can draw the conclusion that the absence of structure (e.g., resonances, non-analytic points) in the bath's spectral density is also beneficial for the derivation of the Lindblad equation from the Redfield one.

### 5.1.1 Hamiltonian RWA

It is crucial to distinguish between the Hamiltonian and Lindblad level RWA. The former is a standard tool in cavity electrodynamics [150–155], and requires its own considerations, separately from the open system dynamics. For the Hamiltonian RWA, the argument of the separation of timescales is the same in spirit as for the Lindblad RWA, although different timescales are under consideration. In this case, one considers only the system dynamics, so in the interaction picture *the most resonant* terms (with respect to the unperturbed dynamics) are kept. The remaining fast oscillating modes average to zero. On the contrary, the terms almost on resonance in the Lindblad equation are the reason for the breakdown of RWA, as they compete with the system's relaxation timescale. A typical case of Hamiltonian RWA is a spin-half $H_0 = \Delta\sigma^z$ interacting with a single photonic mode $H_\beta = \hbar\omega_0 a^\dagger a$. The approximation then amounts to keeping only the counter-rotating terms in the interaction Hamiltonian

$$H_I = g\left(\sigma^+ + \sigma^-\right)\left(a^\dagger + a\right) \longrightarrow H_I^{\text{RWA}} = g\left(\sigma^+ a + \sigma^- a^\dagger\right). \tag{42}$$

We will not comment on the accuracy of the above approximation. The interested reader can see, e.g., [156]. If the study's starting point is an RWA interaction Hamiltonian, an alternative Lindblad form of a master equation for the system's density matrix can be derived without invoking RWA again [27,157].

---

[60]As noticed in 4.3 and 4.4.4, a bath in its ground state can only absorb energy from the system, so only processes with $\Omega > 0$ are allowed—recall that the action of $\mathcal{A}_\alpha(\Omega)$ on the system state decreases its energy by $\hbar\Omega$ (4.1). If a bath is at finite temperature, processes with $\Omega < 0$ become allowed, corresponding to the bath injecting energy into the system.

[61]Or in the very specific case in which there is only one allowed value for $\Omega$, which is essentially equivalent to the case of perfectly degenerate transitions discussed above. For instance, this situation could arise in virtue of a selection rule on the matrix element of $A_\alpha$, or because the *Hamiltonian* RWA has been applied beforehand.

## 5.2   Limitations of the RWA

What kind of error does the RWA introduce? In this section, we formalize the typical reasoning of the averaging out of oscillating terms in the Redfield equation (37) by demonstrating that the RWA stems from a perturbative treatment of the master equation—see [156] for a similar approach. If the separation of timescales required by the secular approximation applies and $\tau_R \gg \tau_S$, the fast-oscillating terms are additionally suppressed by the large frequency difference between the transitions $\Delta\Omega$, and are indeed subleading with respect to other dissipative processes. We can obtain an estimate for the errors introduced by the RWA as follows. Let us re-write the Redfield equation (37) in the Schrödinger picture:

$$
\frac{\mathrm{d}}{\mathrm{d}t}\rho(t) = -\frac{i}{\hbar}[H,\rho(t)] + \sum_{\Omega,\Omega'}\sum_{\alpha\beta}\Big\{\Gamma_{\beta\alpha}(\Omega)\Big[A_\alpha(\Omega)\rho(t)A_\beta^\dagger(\Omega') - A_\beta^\dagger(\Omega')A_\alpha(\Omega)\rho(t)\Big]
$$
$$
+ \Gamma_{\beta\alpha}^*(\Omega)\Big[A_\alpha(\Omega')\rho(t)A_\beta^\dagger(\Omega) - \rho(t)A_\beta^\dagger(\Omega)A_\alpha(\Omega')\Big]\Big\},
\tag{43}
$$

where the subscript "$S$" has been dropped for brevity. Written in this form, the oscillating factors do not appear explicitly anymore since they are absorbed into the coherent (i.e., Hamiltonian) part of the dynamics. The equation above is a linear equation for the system's density matrix $\rho(t)$, and, following the same approach as in section A, it can be written in the form of an ordinary differential equation (ODE) with constant coefficients:

$$
\frac{\mathrm{d}}{\mathrm{d}t}\rho_{mn}(t) = \sum_{pq}\mathcal{R}_{mn;pq}\rho_{pq}(t)
\tag{44}
$$

for the matrix elements of $\rho(t)$ on a chosen basis. We can write the above equation symbolically as the linear ODE $\dot{\boldsymbol{\rho}}(t) = R\boldsymbol{\rho}(t)$ for the $d^2$-dimensional vector $\boldsymbol{\rho}(t)$ containing all elements of the matrix $\rho_{mn}(t)$, and a $d^2 \times d^2$ matrix $R$ whose elements are the quantities $\mathcal{R}_{mn;pq}$ (with $d$ being the dimension of the Hilbert space of the system).

   In order to build some intuition on the above master equation, let us give some details on the coefficients $\mathcal{R}_{mn;pq}$. Let us choose the basis $|m\rangle$ to be the set of eigenstates of the system Hamiltonian $H$. This choice is particularly convenient because the only source of transitions between the states will be the dissipative part. The $(m,n)$ matrix element of the Hamiltonian contribution $-i[H,\rho(t)]/\hbar$ is then $-i(\varepsilon_m - \varepsilon_n)\rho_{mn}(t)/\hbar \equiv -i\Omega_{mn}\rho_{mn}(t)$, where we defined the transition frequency $\Omega_{mn} \equiv (\varepsilon_m - \varepsilon_n)/\hbar$. We observe the existence of two kinds of matrix elements: the populations $\rho_{mm}$ on the diagonal, which are stationary in the absence of the bath, and the off-diagonal coherences $\rho_{mn}$ (with $m \neq n$), whose unperturbed dynamics is a simple oscillation with a frequency $\Omega_{mn}$. We conclude that $\mathcal{R}_{mn;pq}$ has a "diagonal" part $\mathcal{R}_{mn;mn} = -i\Omega_{mn} + \mathcal{O}(\Gamma)$ coming from the Hamiltonian, and "off-diagonal" components $\mathcal{R}_{mn\neq pq}$ that come entirely from the coupling to the bath. The dissipative contributions to both sets are built in terms of the $\Gamma_{mn}$ coefficients and of the matrix elements of the jump operators $A_m(\Omega)$ on the eigenstate basis $|m\rangle$. In general, these quantities also have an imaginary part, which acts as a Hamiltonian contribution $-i\Delta\Omega_{mn}$. The contribution coming from the Lindblad terms (i.e., the ones with $\Omega = \Omega'$) is just the Lamb shift (40) and is a simple renormalization to the unperturbed transition frequencies $-i\Omega_{mn} \to -i\tilde{\Omega}_{mn}$, while the Redfield terms with $\Omega \neq \Omega'$ also generate new Hamiltonian couplings between the unperturbed eigenstates.

   The distinction between populations $\rho_{mm}$ and coherences $\rho_{mn}$, $m \neq n$ is important. In the simplest case in which all transitions are nondegenerate, the Lindblad master equation has the property that the equations governing the evolution of the populations are independent of those determining the coherences [1], and that each coherence is coupled only to coherences having exactly the same transition energy $\Omega$. In particular, in the case of the absence of degenerate transition energies, each coherence evolves on its own. In the language of equation (44),

this property means that the RWA amounts to neglecting all coefficients $\mathcal{R}_{mn;pq}$ that connect each coherence with any other term[62]:

$$
\begin{aligned}
\frac{\mathrm{d}}{\mathrm{d}t}\rho_{mm} &= \sum_n \mathcal{R}_{mm;nn}\rho_{nn} + \sum_{np} \mathcal{R}_{mm;np}\rho_{np} \approx \sum_n \mathcal{R}_{mm;nn}\rho_{nn}\,, \\
\frac{\mathrm{d}}{\mathrm{d}t}\rho_{mn} &= \sum_p \mathcal{R}_{mn;pp}\rho_{pp} + \sum_{pq} \mathcal{R}_{mn;pq}\rho_{pq} \approx \sum_{\substack{pq:\\ \Omega_{mn}=\Omega_{pq}}} \mathcal{R}_{mn;pq}\rho_{pq}\,.
\end{aligned}
\tag{45}
$$

Let us focus on a particular example of two levels, $m = 1$ and $n = 2$. Taking again the full Redfield equation (44), we have

$$
\begin{aligned}
\frac{\mathrm{d}}{\mathrm{d}t}\rho_{11} &= \mathcal{R}_{11;11}\rho_{11} + \mathcal{R}_{11;12}\rho_{12} + \dots\,, \\
\frac{\mathrm{d}}{\mathrm{d}t}\rho_{12} &= \mathcal{R}_{12;11}\rho_{11} + \mathcal{R}_{12;12}\rho_{12} + \dots\,,
\end{aligned}
\tag{46}
$$

where we dropped the couplings to all other matrix elements, indicated with the dots, for the sake of clarity. Then, we obtain a simple matrix-form ODE:

$$
\frac{\mathrm{d}}{\mathrm{d}t}\begin{pmatrix}\rho_{11}\\\rho_{12}\end{pmatrix} = R(g)\begin{pmatrix}\rho_{11}\\\rho_{12}\end{pmatrix} \equiv \begin{pmatrix}\mathcal{R}_{11} & g\\ g^* & \mathcal{R}_{12}\end{pmatrix}\begin{pmatrix}\rho_{11}\\\rho_{12}\end{pmatrix}\,,
\tag{47}
$$

where we defined $g \equiv \mathcal{R}_{11;12} = \mathcal{R}_{12;11}^*$ and we have shortened $\mathcal{R}_{11;11} = \mathcal{R}_{11}$, $\mathcal{R}_{12;12} = \mathcal{R}_{12}$. We also made the assumption that the coupling matrix $R$ is Hermitian; this is generally not the case, but it avoids unnecessary technicalities in the derivation, while leading to the same conclusions as the full treatment with $R^\dagger \neq R$. Following the reasoning preceding equation (45), the Lindblad equation is obtained by dropping the coherence-population couplings, which, in our example, means taking $g \to 0$. Then, the ODE (47) splits into two independent equations whose solution is

$$
\begin{pmatrix}\rho_{11}(t)\\\rho_{12}(t)\end{pmatrix} \stackrel{\text{RWA}}{\approx} \begin{pmatrix}\rho_{11}(0)e^{\mathcal{R}_{11}t}\\\rho_{12}(0)e^{\mathcal{R}_{12}t}\end{pmatrix}\,.
\tag{48}
$$

Our task is to compare this solution with the full solution to equation (47). The latter is solved by:

$$
\begin{pmatrix}\rho_{11}(t)\\\rho_{12}(t)\end{pmatrix} = \boldsymbol{v}_+ e^{\lambda_+ t} + \boldsymbol{v}_- e^{\lambda_- t}\,,
\tag{49}
$$

where $\lambda_\pm$, $\boldsymbol{v}_\pm$ are the eigenvalues and eigenvectors of the coupling matrix $R(g)$, respectively. The two eigenvectors are orthogonal to each other (thanks to the assumption of a Hermitian $R(g)$), and are normalized according to the initial conditions $\boldsymbol{v}_\pm \boldsymbol{\rho}(0) = |\boldsymbol{v}_\pm|^2$, where $\boldsymbol{\rho}(0) = (\rho_{11}(0), \rho_{12}(0))^T$. One could obtain the exact expressions for $\lambda_\pm$ and $\boldsymbol{v}_\pm$, but we only need their expansion around $g = 0$, since the RWA implies that the coupling $g$ has to be small (in the appropriate sense that we now discuss). The expansion can be done by means of the usual Rayleigh-Schrödinger perturbation theory [99], with $R(0)$ as the unperturbed "Hamiltonian" (corresponding to the RWA approximation) and the off-diagonal matrix

$$
V = \begin{pmatrix}0 & g\\ g^* & 0\end{pmatrix}
\tag{50}
$$

---

[62]We notice that in this approximation, the ODE for the populations has the form of a classical master equation for the set of probabilities $\rho_{mm}$ of the states $|m\rangle$, sometimes known as Pauli master equation. For an in-depth discussion of the above equations, we refer the reader to Ref. [1].

as the perturbation. We obtain

$$
\begin{aligned}
\mathbf{v}_+(g) &= \rho_{11}(0)\begin{pmatrix} 1 \\ 0 \end{pmatrix} + \mathcal{O}\left(\frac{g}{\Delta\Omega}\right), \\
\mathbf{v}_-(g) &= \rho_{12}(0)\begin{pmatrix} 0 \\ 1 \end{pmatrix} + \mathcal{O}\left(\frac{g}{\Delta\Omega}\right), \\
\lambda_+ &= \mathcal{R}_{11} + \frac{|g|^2}{\Delta\Omega} + \mathcal{O}\left(\frac{|g|^4}{\Delta\Omega^3}\right), \\
\lambda_- &= \mathcal{R}_{12} - \frac{|g|^2}{\Delta\Omega} + \mathcal{O}\left(\frac{|g|^4}{\Delta\Omega^3}\right),
\end{aligned}
\tag{51}
$$

where $\Delta\Omega \equiv \mathcal{R}_{11} - \mathcal{R}_{12}$.

The question of the validity of the RWA can now be asked quantitatively: under which conditions is equation (48) a good approximation to (49)? By looking at Eqs. (51) we see that the corrections to the unperturbed eigenvalues and eigenvectors should be as small as possible:

$$
|g| \ll |\Delta\Omega| = |\mathcal{R}_{11} - \mathcal{R}_{12}|,
\tag{52}
$$

which is the usual condition for the applicability of perturbation theory [156]. A closer inspection of the matrix elements reveals that this condition is just a slightly refined version of the usual secularity condition $\tau_R^{-1} \ll |\Omega - \Omega'|$. Recalling the discussion of the $\mathcal{R}_{mn;pq}$ coefficients following equation (44), we have $\mathcal{R}_{11} \sim \Gamma_{11}$, $\mathcal{R}_{12} \sim -i\Omega_{12} + \Gamma_{12}$, $g \sim \Gamma_{12}$, and equation (52) becomes $|\Gamma_{12}| \ll |i\Omega_{12} + \Gamma_{11} - \Gamma_{12}|$. Identifying $|\Gamma_{12}| \sim |\Gamma_{11}| \sim \tau_R^{-1}$ with the typical inverse relaxation time, we recover a renormalized version of the above-mentioned secularity condition, with $\Omega = \Omega_{12}$ and $\Omega' = 0$. Of course, this reasoning can be extended to any pair of matrix elements and can be used to obtain a refined set of conditions for the validity of the RWA in cases with multiple decay rates $\gamma(\Omega)$ [141].

The above discussion of the simplified ODE also yields a glimpse of the kind of error that we make by performing the RWA. By plugging the perturbative expansions (51) into the full solution (49), we obtain an expansion in the form

$$
\mathbf{v}_+(g)e^{\lambda_+(g)t} = \left[\begin{pmatrix} \rho_{11}(0) \\ 0 \end{pmatrix} + \mathcal{O}\left(\frac{g}{\Delta\Omega}\right)\right]e^{\mathcal{R}_{11}t + \frac{|g|^2}{\Delta\Omega}t + \mathcal{O}\left(\frac{|g|^4 t}{\Delta\Omega^3}\right)},
\tag{53}
$$

and analogously for the second eigenvector. We see that the RWA introduces two kinds of errors: a fixed amplitude error of order $|g/\Delta\Omega| \sim (|\Delta\Omega|\tau_R)^{-1}$ coming from the modification of the eigenvectors and a *time-dependent* error that comes from the renormalization of the eigenvalues $\lambda_\pm(g)$. Although condition (52) guarantees that the correction to the eigenvalue is small, $|\mathcal{R}_{11}| \gg |g|^2/|\Delta\Omega|$ and cannot overcome the unperturbed result (i.e., there is no secular behavior), the relative error $\|\rho(t) - \rho_{RWA}(t)\|/\|\rho(t)\| \sim |e^{\pm|g|^2 t/\Delta\Omega} - 1|$ may become significant on a timescale proportional to[63] $|\Delta\Omega|/|g|^2 \sim |\Delta\Omega|\tau_R^2$. These results clarify the usual arguments about the averaging out of oscillating terms in the Redfield equation (37): although these terms are formally of the same perturbative order in the system-bath coupling $\Gamma \sim \tau_R^{-1}$ as the non-oscillating (i.e., Lindblad) ones, when the RWA condition $|\Delta\Omega|\tau_R \gg 1$ is satisfied, they are further suppressed by the large frequency difference $|\Delta\Omega|$. Moreover, we obtain a supplemental limitation: the error introduced in the RWA is destined to accumulate in time, and the Lindblad equation ceases to be accurate beyond a timescale $\sim |\Delta\Omega|\tau_R \cdot \tau_R$. Reassuringly, this timescale is much bigger than the relaxation time $\tau_R$ of the Lindblad dynamics under the condition $|\Delta\Omega|\tau_R \gg 1$ for which the latter is valid.

---

[63]Notice that in general $\Delta\Omega$ has both an imaginary part and a real part, so that the deviations from RWA involve both a shift in the oscillation frequency and in the decay rate.

## 5.3 Breakdown of the RWA

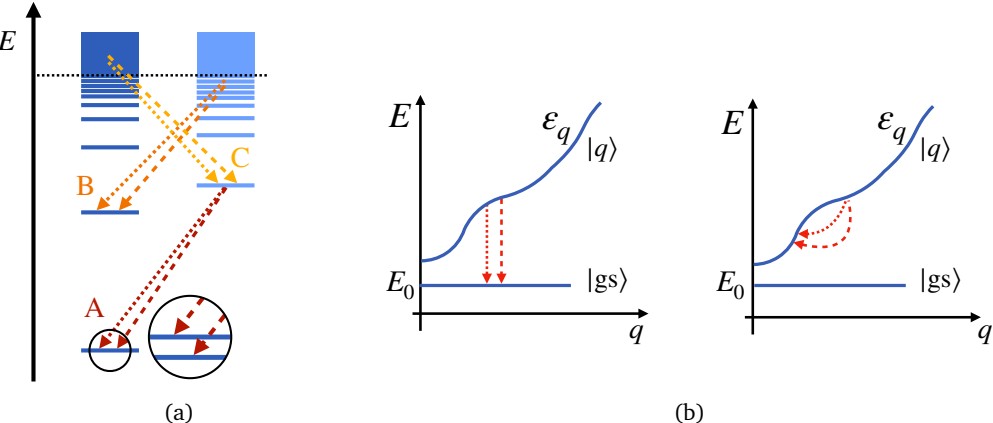

(a)                                                                          (b)

Figure 6: Breakdown of RWA. (a) Cartoon of the spectrum of a few-body system like an atom, featuring both bound states and continuum states (bands above the dotted black line). The colored dashed and dotted arrows represent pairs of "dangerous" transitions for which the RWA might break down. A: splitting of a degenerate manifold into closely spaced levels due to a perturbation (e.g., hyperfine structure or a small external field); B: transitions involving bound states close to the continuum threshold; C: transitions involving continuum states. (b) Cartoon of the energy levels for a single (quasi-) particle in a many body system interacting with a bath. In the example on the left, the effective dissipation annihilates quasiparticles, leaving the system in its ground state, corresponding to jump operators in the form $A(\varepsilon_q) \propto |0\rangle\langle q|$. In the example on the right, the bath changes the quasiparticle momenta, corresponding to jump operators in the form $\mathcal{A}(\varepsilon_q - \varepsilon_{q'}) \propto |q'\rangle\langle q|$. In both cases, the densely spaced momenta $q = 2\pi n/L$ make the possible transitions near-degenerate. In each plot, two such near-degenerate transitions are depicted by the red dotted and dashed arrows.

As discussed in 5.1 and in the previous paragraph, the RWA is inapplicable if the jump operators mediate near-degenerate transitions within the system spectrum, in the sense that $0 < |\Omega - \Omega'|\tau_R \lesssim 1$ (as depicted in the central panel of Fig. 5). We will not discuss here what kind of modifications to the standard derivation of the Lindblad equation are necessary in these situations. We would rather point out in which cases such a near-degeneracy might arise. First, we are going to discuss the safest scenario of systems with a discrete spectrum, and then we will consider the more "dangerous" situation of systems with a dense or even continuous spectrum. The topic is potentially vast and in some points under-studied, so rather than aiming at being exhaustive, we focus on a few examples that should illustrate the limits of RWA in concrete cases. The overall approach will be to identify the physical situation in which it can be expected to find near-degeneracies. When such cases emerge in a concrete example, one should consider other factors, namely: Are the near-degenerate transitions simultaneously allowed, or they never occur in pairs because of selection rules or conservation laws? If they are allowed, is their lifetime sufficiently large to yield $\Delta\Omega\tau_R \gg 1$ (where $\tau_R$ can be taken the minimum of the decay rates of the two transitions), or is that region of the spectrum coupled to the bath sufficiently strongly to have a short lifetime and thus $\Delta\Omega\tau_R \lesssim 1$?

### 5.3.1  Discrete spectra: Few-body systems

The RWA can be generally expected to be accurate for the dynamics of few-body systems in states belonging to the discrete spectrum, since there is no particular reason for two different transitions to have comparable but distinct energies. In these systems, near degeneracy is typically accidental or a result of fine-tuning. This situation is often encountered in AMO physics when considering the spontaneous decay of a small system, such as an atom or a molecule, caused by the interaction with the EM field vacuum. In these systems, different electronic transitions in the low-energy part of the spectrum (e.g., involving different pairs of principal and orbital quantum numbers $(n, l)$ of a single valence electron) involve different energies[64], with frequency differences $\Delta\Omega$ reaching tens of THz, while the corresponding decay rates $\tau_R^{-1}$ are of the order of $10^8$ s or less [158, 159]. Hence, $\Delta\Omega\tau_R \gg 1$ and we can expect a Lindblad treatment to provide an accurate description of the various decay processes[65].

There can be interesting situations in which a form of fine tuning leading to near degeneracies in few-body systems is physically relevant. A possible scenario is that of a set of degenerate states—usually due to a symmetry—is split into a slightly non-degenerate multiplet by a small perturbation, as shown in Fig. 6a. This is the case of fine and hyperfine structure in atoms, which is caused by relativistic effects and interactions with the nuclear multipole moments, respectively [158, 159]. Then, the applicability of the RWA to, e.g., transitions between different members of the multiplet and another state depend on selection rules (i.e., whether the transitions are allowed or not in second-order perturbation theory in the coupling to the EM field) and whether the frequency splitting is comparable with the decay rate of the states. From a practical perspective, the condition $\Delta\Omega\tau_R \gg 1$ amounts to requiring that the spectral lines corresponding to the two transition energies $\Omega$ and $\Omega'$ can be fully resolved. Since the coupling to the EM field is rather small in free space, this is usually the case for atomic transitions[66]. A possibly more delicate situation is when the degeneracy is broken by an external field, such as the Zeeman splitting of atomic levels caused by a static magnetic field. Since the external field can be arbitrarily small, the splitting $\Delta\Omega$ can be small as well. This scenario is known in atomic physics [142], and might also apply to the $m = \pm 1$ states in the ground-state triplet of nitrogen-vacancy centers [160, 161].

A more conventional case of a physically relevant near degeneracy is the celebrated spin-boson model [85],

$$H = -\frac{1}{2}\Delta\sigma^x + \sigma^z \sum_\lambda g_\lambda(b_\lambda + b_\lambda^\dagger) + \sum_\lambda \hbar\omega_\lambda b_\lambda^\dagger b_\lambda \, , \tag{54}$$

in which a single two-level system (a spin $1/2$) is coupled to a bath of bosons $b_\lambda$, such that the absorption or emission of a boson flips the spin. In this model, there is just one transition between the two spin states. If the energy of this transition is large with respect to the Lamb shift and the decay rate, this model can be used to describe the decay of the excited state of an atom in a vacuum [5, 77] and provides a classic example of a successful use of the Lindblad equation. But the model can also describe tunneling in a dissipative environment, in which

---

[64]With the exception of hydrogen, whose nonrelativistic levels only depend on the principal quantum number $n$. This degeneracy is lifted by relativistic effects (fine structure).

[65]As long as the EM vacuum fluctuations can be considered as the only source of decay. In real experiments, the spectral linewidths are increased by other phenomena, such as the Doppler effect and atomic collisions. Moreover, when dealing with multi-electron excitations, or those involving core electrons, the dominant decay channel is the Auger effect [158, 159], in which an electron is emitted instead of a photon. This process is produced by the Coulomb repulsion rather than the vacuum EM fluctuations, and consequently features short lifetimes with $\Delta\Omega\tau_R \lesssim 1$ [147].

[66]As a rough estimate, the hyperfine energy splitting of the ground state of the hydrogen atom—the famed 21 cm line—is about 1.42 GHz, still larger than, e.g., the typical decay rate from the $2p$ to the $1s$ multiplets, which is of the order of $10^8 \, \mathrm{s}^{-1}$.

the two spin states represent the two minima in a double-well potential, and the transition frequency acquires the meaning of a tunneling amplitude. In this case, the interaction with the bath can lead to a complete suppression of tunneling (localization), which is equivalent to the transition frequency becoming effectively zero. Thus, in this scenario, it is important to consider the limit of a small (but finite) transition frequency, and the RWA cannot be applied[67].

### 5.3.2 Dense spectra and many body systems

Near degeneracy is to be expected when a system possesses a continuous spectrum, or a spectrum that becomes densely spaced in some appropriate large-volume limit. In this scenario, one can naturally find transitions that have arbitrarily close energies. For few-body systems, this can be the case for transitions involving states in the continuum at energies above the discrete spectrum[68]—shown as case C in Fig. 6a—or simply for unbound systems. A physically interesting example of the latter is quantum Brownian motion [5, 107], in which an otherwise free quantum particle interacts with a bath of harmonic oscillators. The spectrum of a free particle is continuous; hence, the RWA cannot be applied, in general. Indeed, the master equation describing quantum Brownian motion is not derived following the Lindblad treatment described here, and it is generally not even of the Lindblad form, unless ad hoc terms are introduced [5, 40, 107]. Only if the Brownian particle is confined by a sufficiently deep potential, so that its energy spectrum becomes discrete, the usual Lindblad description can be applied [30, 88, 102].

If the system is many body, in the sense that its number of constituents is proportional to its volume, then a nearly continuous spectrum is to be expected for large enough volumes, and the applicability of RWA must be assessed with care. These situations, involving an extended system perturbed by a reservoir, are outside the scope of the traditional AMO physics and are more related to the field of condensed matter and "modern" AMO physics, namely the field of synthetic quantum matter and quantum simulation. This regime of application of Lindblad has emerged more recently [8]. As we will show in the following, the general expectation for many body systems is that RWA will not be valid [162]. Although this limitation of the Lindblad equation has been long known [20, 24, 39, 163, 164], a full characterization of its regimes of applicability in many body systems is still missing[69]. Therefore, the following discussion will be qualitative and will mostly serve as an encouragement to the reader to approach the problem of Lindbladian dissipation in many body systems with a critical eye. In these situations, the Lindblad master equation can still be useful as a phenomenological approach, i.e., as a way to capture the essential features of dissipation. If quantitative predictions are required, one should resort to other approaches, such as the Redfield equation, alternative master equations (see, e.g., [163], and the next section 5.4), or completely different techniques that take the bath into account, such as Keldysh field theory [23, 149, 165–172].

---

[67]Although for $\Delta \to 0$ also the Markov and Born approximations are expected to break down, depending on the properties of the bath spectral density at low frequencies. Indeed, if the bath is Ohmic, $J(\omega) \sim \omega$, the spin-boson model can be mapped to the Kondo model discussed in 4.5.1.

[68]Or for states in the discrete spectrum close to the continuum threshold, as for instance between highly excited states (Rydberg states) of an atom, which are bound states but densely clustered around the ionization threshold. While the level spacing $\Delta\Omega$ between Rydberg states at consecutive principal quantum numbers $n \gg 1$ scales as $n^{-3}$, they have rather long lifetimes $\tau_R$, scaling as $n^3$ or $n^5$ for low and high angular momentum states, respectively [159]. Then, $\Delta\Omega\tau_R$ does not generally vanish as $n$ increases, and RWA turns out to be applicable. This example illustrates the importance of other factors beyond the simple level spacing in the RWA.

[69]Notice that a continuous spectrum might also become problematic for the Markovian approximation. For instance, the timescale of Hamiltonian dynamics $\tau_H = \Omega^{-1}$, that helps ensuring the validity of the approximation if $\tau_H \ll \tau_R$ (see 4.1) becomes arbitrarily large, and part of the transitions in the system will probe the bath spectrum around zero frequency, where usually band edge effects manifest.

**Single body losses** To illustrate the kind of problems that RWA may run into in many body systems, let us take the simple situation in which the system Hamiltonian is that of free particles with dispersion $\varepsilon_q$: $H = \sum_q \varepsilon_q a_q^\dagger a_q$. We consider a 1D case for simplicity, as space dimensionality does not play a role in what follows. The ladder operators $a_q$ might be bosonic or fermionic, and we consider the system to be of finite but large size $L$, so that the momenta are quantized as $q = 2\pi n/L$, $n = 0, \pm 1, \pm 2, \ldots$ (assuming periodic boundary conditions; if and how the momenta are cut off at high energy is irrelevant to our discussion), but closely spaced. The above Hamiltonian might be describing (approximately) noninteracting electrons in a solid or atoms in an optical lattice, or quasiparticles emerging at low energy, such as phonons, magnons, polaritons, etc. Let us consider the energy levels for a single particle with momentum $q$, $|q\rangle = a_q^\dagger |\text{gs}\rangle$, where $|\text{gs}\rangle$ is the vacuum. A possible effect of the bath may be to annihilate particles, as illustrated in the left-hand side of Fig. 6b. The corresponding eigenoperator would connect $|q\rangle$ with the ground state $|0\rangle$, involving a transition frequency $\Omega = -\varepsilon_q/\hbar$. Since $\varepsilon_q$ is a continuous function of momentum, one can always find transitions with arbitrarily close frequencies $\Omega = -\varepsilon_q/\hbar$ and $\Omega' = -\varepsilon_{q+\delta q}/\hbar$, hence hindering the RWA. Indeed, $|\Omega - \Omega'| \approx |d\varepsilon_q/dq\, \delta q/\hbar| \propto \delta q$, and since the minimum momentum increment is $\delta q \propto L^{-1}$, we have that $|\Omega - \Omega'| \propto L^{-1}$ can be made arbitrarily small at large system sizes. At the same time, the decay rate associated to the two transition only depend on their energies through $\tau_R^{-1}(\Omega) = 2\,\text{Re}\,\Gamma_{\alpha\beta}(\Omega)$ for an appropriate choice of $\alpha$ and $\beta$ (and analogously for the $\Omega'$ transition—see equation (13)) and usually remains finite for $L \to \infty$. Hence, $\Delta\Omega \tau_R(\Omega) \sim L^{-1}$ vanishes[70] in the thermodynamic limit.

There is, however, an interesting possibility related to momentum conservation. In the argument above, we have considered the choice of transition frequencies $\Omega$, $\Omega'$ as free of any constraints, namely, that any combination of the eigenoperators $\mathcal{A}_\alpha(\Omega)$ and $\mathcal{A}_\beta(\Omega')$ is allowed in the Redfield equation (37). However, if translation is a weak symmetry of the system, i.e., total momentum is conserved by the system and environment [173, 174], then the pairs of eigenoperators in (37) are constrained. Indeed, let us consider a term like $\mathcal{A}_q(\Omega)\rho_S(t)\mathcal{A}_{q'}^\dagger(\Omega')$ (the following reasoning works analogously for the other terms), where $\mathcal{A}_q(\Omega)$ removes a particle with momentum $q$ (then, $\Omega = -\varepsilon_q/\hbar$ as described above[71]. Then, the two operators describe dissipative transitions that impart two momentum "kicks" of $-q$ and $-q'$ to the system. However, by momentum conservation, the two kicks must be the same—if $\rho_S(t)$ is diagonal in momentum at $t = 0$, it must remain so during the dynamics. Then, $q = q'$ implies $\varepsilon_q = \varepsilon_{q'}$, i.e., perfect degeneracy $\Omega = \Omega'$, and the RWA does not need to be applied. This argument illustrates how conserved quantities may extend the validity of the Lindblad equation to many body systems.

**Conserved particles** The previous argument becomes insufficient in case the eigenoperators conserve the number of particles but change their energy, as shown in the right-hand part of Fig. 6b. If the energy change is not quantized for other reasons, one finds near-degenerate transitions that invalidate the RWA. In this scenario, we need to consider terms like $\mathcal{A}_{pk}(\Omega)\rho_S(t)\mathcal{A}_{p'k'}^\dagger(\Omega')$ in the Redfield equation (37), with $\mathcal{A}_{pk}(\Omega)$ imparting a momentum change $p - k$ with the associated frequency difference $\Omega = (\varepsilon_p - \varepsilon_k)/\hbar$ (and analogously for $\mathcal{A}_{p'k'}^\dagger(\Omega')$). In the absence of momentum conservation, all four momenta $p$, $k$ $p'$, $k'$ are arbitrary, and so is $\Delta\Omega = (\varepsilon_p - \varepsilon_k - \varepsilon_{p'} + \varepsilon_{k'})/\hbar$—there is a large phase space for near degeneracies $0 < |\Delta\Omega| \lesssim \tau_R^{-1}$. With momentum conservation, the momenta are constrained by

---

[70]The decay time $\tau_R(\Omega)$ is generally a smooth function of $\Omega$, since it is directly related to the spectral density $J(\Omega)$ (see 4.4), which should be smooth for the Markovian approximation to hold. Hence, when evaluating $\Delta\Omega\tau_R$ for closely spaced transitions, $\tau_R(\Omega) \approx \tau_R(\Omega')$ and either of them can be used.

[71]For simplicity, we are assuming a one-to-one correspondence between energy and momentum, because it makes the argument simpler. This assumption can be relaxed without altering the conclusion.

$p - k = p' - k'$, which is generally not sufficient to ensure[72] $\varepsilon_p - \varepsilon_k = \varepsilon_{p'} - \varepsilon_{k'}$. Overall, the present example is completely analogous to that of quantum Brownian motion for unbound particles, with the only conceptual difference that in many body systems, the particle might be an emergent, collective object. Let us remark that the limitation of RWA is not restricted to the single-particle sector considered above: any transition frequency in the many body spectrum of the above Hamiltonian is constructed by summing or subtracting factors of $\varepsilon_q$, which is a continuous function of all the involved momenta.

**Trapped ultra-cold atoms**   As shown above in the case of particle loss, a nearly continuous spectrum does not necessarily imply that a Lindbladian description of the dynamics is in principle impossible. In the above-mentioned example, a conserved quantity imposes a strong constraint on the possible pairs of eigenoperators occurring in the Redfield equation. Other scenarios are possible. For instance, the spectrum might be continuous, but the eigenoperators connect only states with a minimum energy difference that is independent of system size. It is worth mentioning the case of systems trapped by a harmonic potential, which is a common setup for ultra-cold atoms. These systems may contain $\mathcal{O}(10^5)$ atoms, but their spectrum remains discrete (as long as we can neglect interactions within the systems), with a minimum energy spacing proportional to the smallest of the confinement frequencies in the three directions of space, $\omega_0$. Then, regardless of the nature of the coupling with the bath, the minimum energy difference between two bath-induced transitions will be $\hbar\omega_0$. Hence, RWA will be applicable if $\omega_0 \tau_R \gg 1$, namely if the timescale of the dissipative dynamics is much longer than the longest of the oscillation periods in the harmonic potential—a condition for a strongly underdamped motion[73]. The important point of this example is that the minimal energy difference $\omega_0$ is independent of the size of the system, namely the number of trapped particles, so that the validity of RWA is uniform for all particle numbers.

In summary, the RWA can be expected to be a reasonable approximation in few-body systems with a discrete spectrum, while it is expected to fail in generic many body systems, unless special conditions on the energy spectrum and/or jump operators are met. In the latter systems, the validity of RWA has to be verified case by case.

## 5.4  To RWA or not to RWA? *

Numerous studies comparing the accuracy of the Redfield and Lindblad master equations are available in the literature [24–26, 41, 178]. As indicated in Ref. [24], the Redfield equation consistently outperforms the Lindblad equation in its accuracy. As expected, both formalisms produce lower errors in the regions of the parameter space where the Born-Markov approximation holds, and the magnitude of the error grows with the magnitude of the perturbative parameter $\tau_B \lambda / \hbar$ (the product of the bath coherence time $\tau_B$ and the coupling strength $\lambda$). Nevertheless, the Lindblad description breaks down quicker for any system without a perfectly degenerate spectrum. This is expected since, as stated in the equation (52), RWA is invalid outside of the weak coupling regime.

An argument for nevertheless applying RWA is the numerical convenience of working with a Lindblad-like description. As discussed after equation (44), the general form of both master equations is $\dot{\boldsymbol{\rho}}(t) = R\boldsymbol{\rho}(t)$, in the vectorized form that is suitable for numerical computation. In general, the Redfield equation requires constructing and manipulating a $d^2 \times d^2$ matrix $R$

---

[72]Unless the dispersion is strictly linear in momentum, i.e., $\varepsilon_p = vp$. This chiral dispersion can be found as an effective low-energy description in certain 1D systems such as Tomonaga-Luttinger liquids [175] and the edge modes of topological phases [176, 177].

[73]For a typical trap, $\omega_0 \sim 2\pi \cdot 100$ Hz [159], which means that RWA is valid as long as $\tau_R$ is at least a few tens of milliseconds—a long time, but not too long for ultra-cold atom experiments.

(although it is often sparse in physical cases [179]). Thus, the memory resources for storing the matrix $R$ and the computational complexity of determining the time evolution of $\rho(t)$ grow rapidly as a function of $d$. This growth imposes severe limits, especially for many body systems, in which the Hilbert space dimension $d$ increases exponentially with the number of constituents. The situation with the Lindblad master equation is slightly more benign since, in the absence of degenerate transitions, one can separate it into an equation for the $d$ populations (the so-called Pauli master equation [179]), which is governed by a $d \times d$ matrix, and $d(d-1)/2$ independent equations for the coherences. The presence of degenerate transitions couples the coherences and increases the complexity of the vectorized master equation. Still, the overall number of nonzero elements of $R$ is smaller than for the Redfield equation. We notice that the construction of the matrix $R$ itself from the Hamiltonian $H$ and the $n$ jump operators $A_a(\Omega)$ may require a number of operations that scale differently for the two types of master equations [179]: as $n^2$ for the Redfield equation, and as $n$ for the Lindblad one, the difference coming from the need of evaluating all cross terms with $\Omega' \neq \Omega$ in equation (37). The Lindblad master equation has the distinct advantage that it can be (approximately) solved by means of well-established quantum trajectory approaches[74] [102, 172, 180–182]. These approaches are based on so-called stochastic unravelings of the master equation, namely recipes to construct an ensemble of time-evolving *pure* states (the quantum trajectories) $|\psi_j(t)\rangle$ such that the average state $\bar{\rho}(t) = N_{\mathrm{traj}}^{-1} \sum_j |\psi_j(t)\rangle\langle\psi_j(t)|$ estimates, in a statistical sense, the mixed density matrix $\rho(t)$ that solves the master equation. The crucial benefit of these approaches is that they work in the usual Hilbert space of pure states and not in the space of density matrices and thus require much less memory (i.e., only implementing the Hamiltonian and jump operators as $d \times d$ matrices instead of the $d^2 \times d^2$ tensor $R$), and typically less computational time. The extra cost of statistical sampling of trajectories is usually moderate since one can reach convergence with $N_{\mathrm{traj}} \ll d^2$ trajectories [182]. In particular, these methods can become necessary for treating many body systems in which $d$ grows exponentially with the system size.

These advantages motivate a search for less blunt RWA alternatives, which nevertheless lead to a Lindblad form. The most direct approach to systems with nearly-degenerate transitions is to apply the secular approximation only partially and in a controlled manner. This can be done by inspection [141, 183] or by utilizing the piecewise flat spectral-function (PFSF) approximation [184]. The latter method involves approximating the bath spectral functions with a piecewise constant function in the frequency domain, which nevertheless remains smooth on the frequency scales of the coherent time evolution and dissipation. The approximation neglects the cross-terms between transitions belonging to different constant sections of the spectral function, as these correspond to the highly non-degenerate case of the system spectrum, as discussed in section 5.1. The cross-terms between nearly degenerate transitions are kept, but both transitions are assigned an equal transition frequency approximating their actual values. Essentially, this means treating spectral emissions that are below the resolution limit (provided by the inverse linewidth) as the same spectral line. Partial RWA leads to the inclusion of extra processes within the master equation and, thus, potentially qualitatively new behaviors.

A useful tool in error control of the approximations present in the Lindblad equation is the introduction of an explicit coarse-graining timescale, $\Delta t$. In practice, this timescale provides a resolution with which the dynamics are examined. For the validity of RWA, we require that $\tau_S \ll \Delta t$ so that the fast oscillations average to zero. A derivation of the Lindblad equation based on explicit coarse-graining of the dynamics offers control over the error bounds [1]. A Lindblad form positivity-preserving master equation can also be obtained by averaging over the coarse-graining timescale and applying RWA only partially [185] or without invoking it

---

[74]To the best of the authors' knowledge, there is no equivalent technique for the Redfield equation that has found a comparably wide diffusion.

at all [163, 164, 185–188]. Moreover, the explicit presence of the coarse-graining parameter lends itself to optimization procedures [187].

Applying the full RWA treatment is justified if no part of the transition spectrum is nearly degenerate and the phenomena of physical interest are insensitive to variations below the resolution time scales. Otherwise, as discussed in this section, a plethora of more refined techniques are available that simplify the Redfield description without discarding all coherences absent in the traditional Lindblad formalism.

## 6   Is Lindblad for me?

As we reach the end of this review, we are faced with the natural task of providing a concise yet practical synthesis for readers seeking guidance on when the Lindblad equation can—and cannot—be reliably applied to model open quantum dynamics.

To this end, we offer a summary table 6 that distills the core insights of the review into a comparative format. It juxtaposes conventional wisdom with a more nuanced perspective for each of the standard assumptions or commonplace beliefs surrounding the Lindblad formalism. The items in this table do not necessarily follow the sequence in which topics were introduced throughout the review. In assembling Table 6, we have deliberately moved beyond the usual trio of assumptions (Born, Markov, and Rotating Wave), incorporating also pervasive issues that researchers encounter when applying the Lindblad equation to modern research problems.

Our aim is that this final summary serves as cautionary checklist and as a conceptual map for future research work on the breakdown of Lindblad quantum dynamics.

| Conditions on validity of the Lindblad equation | | |
|---|---|---|
| | **Conventional Wisdom** | **Refined View** |
| Dissipative channels | Jump operators take the form of system's operators coupled to the bath. | Jump operators are dictated by both the operators coupled to the bath and the exact eigenstates of the system's Hamiltonian. Hence, they are generally collective for multipartite systems. See 4.2. |
| Weak interactions | System-bath coupling is small. | The combination of the coupling strength $\lambda$ and bath correlation time $\tau_B$ need to be small $\lambda\tau_B/\hbar \ll 1$. See 3.1. |
| Perturbation theory | Small system-bath coupling ensures that the Lindblad master equation can be derived in a perturbative manner. | The system-bath coupling can become renormalized by the bath correlation functions and become effectively a large parameter. A famous example of such a case is the Kondo model. See 4.5.1. |
| Born approximation | Density matrix is approximately separable, $\rho \approx \rho_S \otimes \rho_B$. | Density matrix is not fully separable, but the dynamics of $\rho_S$ are mostly dictated by the separable part. See 3.1. |
| Markovian treatment | The dynamics of the system are either Markovian or not. | Markovianity is always a matter of timescales. Markovian dynamics holds at intermediate times, while the initial and final transient are generally non-Markovian (due to correlation build-up and finite bath bandwidth, respectively). See 4.4. |
| Markovian bath I | Bath spectral density $J(\omega)$ needs to be smooth. | The system probes $J(\omega)$ around the energies of the transitions induced by the bath. The Markov approximation is valid if $J(\omega)$ does not have sharp features in this specific probing window. See 4.4. |
| Markovian bath II | Bath correlation functions $G(t)$ need to decay exponentially to be Markovian. | Bath correlation functions need to decay "fast enough". For instance, in the case of a non-interacting fermionic bath, it means just faster than $1/t$. See 4.3.1. |
| Markovianity vs. temperature | The hotter the bath, the more Markovian the dynamics. | The dependence of Markovianity on temperature is highly non-trivial and needs to be checked on a case-by-case basis. In particular, for the popular choice of a boson bath, the Markovianity of the system varies non-monotonically with temperature. See 4.4.4. |
| RWA I | Timescale of the internal system dynamics $\tau_S$ needs to be much smaller than the relaxation timescale $\tau_R$, $\tau_S \ll \tau_R$. | RWA requires a separation of energy scales, so that perturbation theory in the dissipative rates around the Hamiltonian dynamics of the system is valid. See 5.2. |
| RWA II | RWA introduces only a relative error to the dynamics. | RWA completely removes certain processes, making it impossible to study their dynamics. RWA may not be applicable in systems with near-degenerate transitions (like many body systems). See 5.3. |

## Acknowledgments

We thank F. Balducci, N. Bar-Gill, M. Buchhold, O. Chelpanova, J. Keeling, H. Hosseinabadi, and X. Hu for helpful discussions. JM gratefully acknowledges the hospitality of the Pauli Center at ETH Zurich. During the Spring Semester of 2024, he delivered at ETH a short course on open quantum systems, which inspired this pedagogical review. AZ acknowledges support from the Alexander von Humboldt Foundation. We acknowledge support from the European Union's Horizon 2020 research and innovation programme under Grant Agreement No 101017733 "QuSiED") and by the DFG (project number 499037529), and by the DFG through the grant HADEQUAM-MA7003/3-1.

## A   What to expect from the Lindblad equation?

The formal properties of the Lindblad equation (1), such as the characterization of its stationary states and symmetries, have been studied extensively and can be found in different references [1, 5, 16, 28, 55, 173, 189]. In order to offer a self-consistent presentation, we summarize in this Appendix some formal aspects of the solution of the Lindblad equation.

The most direct way of solving equation (1) is to obtain the equations for the elements of the density matrix in a given basis[75], $\{|n\rangle\}$ (often, the eigenstates of the Hamiltonian $H$):

$$\frac{\mathrm{d}}{\mathrm{d}t}[\rho_S(t)]_{nm} = \sum_{n',m'} L_{nm,n'm'}[\rho_S(t)]_{n'm'} \,, \tag{A.1}$$

where $L_{nm,n'm'}$ is a set of constant coefficients depending on the system Hamiltonian $H$, the jump operators $L_a$, and the rates $\gamma_a$. If the Hilbert space of the system has size $d$, the equations above form a system of $d^2$ linear, ordinary differential equations (ODEs) for the matrix elements $[\rho_S(t)]_{nm}$. The conditions $\rho_S^\dagger(t) = \rho_S(t)$ and $\mathrm{Tr}\,\rho_S(t) = 1$ reduce the number of independent equations to $d_e \equiv d(d+1)/2 - 1$, with $d(d-1)/2$ equations for the off-diagonal elements $[\rho_S(t)]_{nm}$, $m > n$, and $d-1$ for the diagonal ones, $[\rho_S(t)]_{nn}$. In practice, one considers the composite index $(n, m)$ as a single index and organizes the set of matrix elements into a $d_e$-dimensional vector $\boldsymbol{\rho}_S(t)$. This procedure, known as vectorization, can be performed by, for instance, "stacking up" the columns of $[\rho_S(t)]_{nm}$ to form $\boldsymbol{\rho}_S(t)$ and correspondingly rearranging the set of coefficients $L_{nm,n'm'}$ into a $d_e \times d_e$ matrix $\hat{L}$. The result is a recasting of the Lindblad equation in the familiar form $\dot{\boldsymbol{\rho}}_S(t) = \hat{L}\boldsymbol{\rho}_S(t)$[76]. The solution to this linear ODE amounts to computing a matrix exponential, $\boldsymbol{\rho}_S(t) = \exp(\hat{L}t)\boldsymbol{\rho}_S(0)$, which can be compared with the unitary dynamics $|\psi(t)\rangle = \exp(-iHt/\hbar)|\psi(0)\rangle$ of the solution to the linear Schrödinger equation.

The conceptually most straightforward way of computing the matrix exponential is through the diagonalization of the matrix $\hat{L}$. Unlike a Hamiltonian, the matrix $\hat{L}$ is generally neither Hermitian nor symmetric, so in general[77] one must find a so-called bi-orthogonal basis [190], which consists of a set of right eigenvectors $\hat{L}\boldsymbol{r}_\mu = \lambda_\mu \boldsymbol{r}_\mu$ and left eigenvectors

---

[75]Here, we assume the basis to be finite-dimensional. The infinite-dimensional case of the Lindblad equation can be considered analogously, but in the case of a continuous spectrum, it is harder to justify its validity from first principles.

[76]This is at times referred to as a superoperator form of a master equation and the matrix $\hat{L}$ as a superoperator Lindbladian. The "super-" prefix refers to the fact that the Lindbladian is a linear operator acting on other operators (the density matrices).

[77]We are considering the simplest case of a Lindbladian matrix $\hat{L}$ that is diagonalizable, i.e., in which one can find $d_e$ distinct eigenvectors. The most general scenario includes the possibility of having non-diagonalizable Jordan blocks [55], whose matrix exponential can be computed nevertheless. This scenario is a complication that does not alter the conclusions of this section.

$l_\mu^\dagger \hat{L} = \lambda_\mu l_\mu^\dagger \iff \hat{L}^\dagger l_\mu = \lambda_\mu^* l_\mu$ corresponding to complex eigenvalues $\lambda_\mu$. If $\hat{L}$ was Hermitian (or at least symmetric), the spectral theorem would guarantee that $\lambda_\mu$ would be real and that $r_\mu = l_\mu$ would form an orthogonal set. For a non-Hermitian matrix, one can impose biorthogonality, $\langle l_\mu, r_\nu \rangle = \delta_{\mu\nu}$, while each of the sets of right or left eigenvectors themselves does not form an orthogonal basis. Here, the product $\langle \, , \, \rangle$ is an inner product on the Hilbert space of the problem, $\langle a, b \rangle \equiv \sum_i a_i^* b_i$. In terms of this basis, the solution to the vectorized Lindblad equation is

$$\rho_S(t) = \sum_\mu e^{\lambda_\mu t} \langle l_\mu, \rho_S(0) \rangle r_\mu \,. \tag{A.2}$$

In words, each matrix element of $\rho_S$ is a sum of exponentials[78] with different, complex rates $\lambda_\mu$, with weights proportional to the initial state $\rho_S(0)$. Since the Lindblad equation must generate a physically sensible density matrix at all times, we can anticipate that $\text{Re}\,\lambda_\mu \leq 0$, so that $\rho(t)$ is finite for $t \to \infty$. Indeed, this is guaranteed by the specific form of the Lindblad equation, and in particular by the requirement that the rates $\gamma_a$ in equation (1) are positive. The quantities $\text{Re}\,\lambda_\mu$ represent the decay rates of the components of the initial density matrix due to dissipation. After a sufficiently long time, all exponentials with $\text{Re}\,\lambda_\mu < 0$ will have decayed completely, leaving only those with a vanishing decay rate $\text{Re}\,\lambda_\mu = 0$. The Lindblad equation always has at least one stationary state, corresponding to $\lambda_{\mu_0} = 0$ [173, 189, 191, 192]. In many cases in which the environment is initially in thermodynamic equilibrium, this state is simply a thermal state,[79] $\rho_S^\infty \propto \exp(-\beta H_S)$, at the (inverse) temperature of the bath $\beta = (k_B T)^{-1}$. [5, 55], but it is possible to engineer the jump operators and rates to yield interesting stationary states, including pure states [173, 193], and non-equilibrium steady states [194–196].

# B   Solution to the toy model of non-Markovian dynamics*

In this appendix, we derive the solution to the toy model that we employed in section 4.4 to understand the limits of the Markovian approximation. For clarity, we report the equation here:

$$\frac{\text{d}}{\text{d}t} f(t) = -i\omega_0 f(t) + \int_0^t \text{d}\bar{t}\, K(t - \bar{t}) f(\bar{t}) \,. \tag{B.1}$$

We changed the notation of the integration variable to $\bar{t}$ because, throughout this appendix, we will reserve the symbol $s$ for the variable of Laplace transforms. The integro-differential equation above can be solved via the Laplace transform [90]. For a generic function $\varphi(t)$, the latter is defined as

$$\varphi(s) = \int_0^\infty \text{d}t\, e^{-st} \varphi(t) \,, \tag{B.2}$$

where $s$ is a complex number. The transform $\varphi(s)$ is usually defined for $\text{Re}\,s < a$ or $\text{Re}\,s \leq a$, where $a$ is determined by the properties of $\varphi(t)$. Applying the Laplace transform to the toy model (B.1), we obtain $s f(s) - f_0 = [-i\omega_0 + K(s)] f(s)$, where $f_0 = f(t=0)$ is the initial value of the function $f(t)$. We obtain

$$f(s) = \frac{f_0}{s + i\omega_0 - K(s)} \,. \tag{B.3}$$

---

[78] In the most general scenario that includes Jordan blocks, some of the exponentials can be multiplied by polynomial functions of time.

[79] One may notice that such state $\rho_S^\infty$ is independent of any property of the bath except for its temperature. This occurs because equation (1) is derived for an infinitesimal coupling to the bath. Higher-order master equations would renormalize $\rho_S^\infty$ [53].

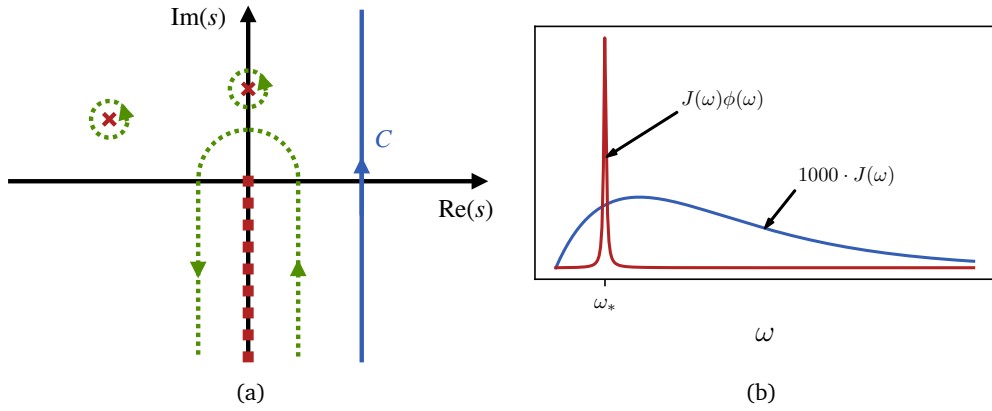

Figure 7: Solution to the toy model of non-Markovian dynamics. (a) Contour of integration for the inversion of the Laplace transform $f(s)$. The original contour $\mathcal{C}$ (blue, continuous line) is deformed to the one (green, dashed lines) that goes around all singularities: a branch cut (thick dashed, red line on the imaginary axis) and two poles (red crosses). (b) The action of the filter function $\phi(\omega)$ on the spectral density $J(\omega)$ in the branch cut contribution (B.11) is to select only a narrow region around the renormalized frequency $\omega_* \approx \omega_0$. Notice that the spectral density has been multiplied by a factor of 1000 to facilitate the comparison. We chose an Ohmic spectral density $J(\omega) = \eta \omega e^{-\omega/\Lambda}$, with $\eta = 0.1$ and $\omega_0 = 0.6$ in units of $\Lambda$.

The real-time solution $f(t)$ is obtained by inverting the Laplace transform via the formula

$$f(t) = \int_{\mathcal{C}} \frac{\mathrm{d}s}{2\pi i} e^{st} f(s) = \int_{\mathcal{C}} \frac{\mathrm{d}s}{2\pi i} e^{st} \frac{f_0}{s + i\omega_0 - K(s)} , \qquad (B.4)$$

where the integration contour $\mathcal{C}$ runs vertically in the complex plane (i.e., parallel to the imaginary axis) and has to be located to the right of all singularities of $f(s)$, as indicated by the blue continuous line in Fig. 7a. The solution $f(t)$ will be obtained by pushing the contour towards $\operatorname{Re} s \to -\infty$, leaving only the contributions from the non-analytic points of $f(s)$, as sketched by the green, dashed lines of Fig. 7a. The singularities of $f(s)$ are either poles, corresponding to zeroes of $s + i\omega_0 - K(s)$, or branch cuts. We are going to show that $f(s)$ always has a branch cut on the negative imaginary axis, $s = -i\omega$, for all values of $\omega$ where the spectral density $J(\omega)$ is nonzero. Then, we can write the solution as

$$f(t) = f_0 \left[ \sum_p Z_p e^{s_p t} + \int_{J(\omega) \neq 0} \frac{\mathrm{d}\omega}{2\pi} e^{-i\omega t} [f(s = -i\omega + 0^+) - f(s = -i\omega - 0^+)] \right], \qquad (B.5)$$

where the sum on the right-hand side runs on all poles $s_p$ of $f(s)$, with residues $Z_p = [1 - \mathrm{d}K(s_p)/\mathrm{d}s]^{-1}$, while the integral is the contribution from the branch cut. We will see that the former terms are non-Markovian contributions, which are either very small or vanishing, while the Markovian dynamics originate from the branch-cut contribution.

To proceed further, we need to characterize the Laplace transform of the kernel $K(t)$. Expressing the latter in terms of $J(\omega)$, equation (27), we have

$$K(s) = -\int \frac{\mathrm{d}\omega}{2\pi} \frac{J(\omega)}{i\omega + s} . \qquad (B.6)$$

Then, we find that

$$K(s = -i\omega \pm 0^+) = -iR(\omega) \mp \frac{1}{2} J(\omega) , \qquad (B.7)$$

where

$$R(\omega) \equiv \mathcal{P} \int \frac{d\varepsilon}{2\pi} \frac{J(\varepsilon)}{\omega - \varepsilon} . \tag{B.8}$$

As stated before, $K(s)$—and therefore $f(s)$—has a branch cut on the imaginary axis, wherever the spectral density is nonzero. Assuming that $J(\omega)$ vanishes for $\omega < 0$, the branch cut is located on the negative part of the imaginary axis.

Let us now consider the occurrence of poles of $f(s)$. Writing $s_p = x - iy$ and taking the real and imaginary parts of $s_p = -i\omega_0 + K(s_p)$, we obtain

$$x = -\int \frac{d\omega}{2\pi} J(\omega) \frac{x}{x^2 + (\omega - y)^2} ,$$
$$y = \omega_0 - \int \frac{d\omega}{2\pi} J(\omega) \frac{\omega - y}{x^2 + (\omega - y)^2} . \tag{B.9}$$

In particular, the equation for the real part $x$ is of the form $x = F(x; y)$, where $F(x; y)$ is an odd function of $x$. However, since $J(\omega)$ is positive, the sign of $F(x; y)$ is opposite to that of $x$. Then, the only solution $x$ to $x = F(x; y)$ has to be $x = 0$. The latter corresponds to a pole on the imaginary axis, and we have just proven that the real part of $K(s)$ (which is just $F(x; y)$) is discontinuous for all $s = -iy$ for which $J(y) \neq 0$—namely, $F(x; y > 0)$ changes sign at $x = 0$ without going through zero, and there cannot be a solution to $x = F(x; y)$ for positive $y$. Then, all poles must be of the form $s_p = -i\omega_p$, where $\omega_p$ solves [90]

$$\omega_p = \omega_0 + R(\omega_p), \quad J(\omega_p) = 0 . \tag{B.10}$$

According to the general solution (B.5), such poles would correspond to undamped modes $Z_p e^{-i\omega_p t}$ in $f(t)$, which are rather unexpected for a dissipative system, and that would correspond to a non-Markovian solution. These poles signal the presence of bound states in the coupled system-bath spectrum. In general, their existence requires a strong enough coupling [90], or they come with a highly suppressed residue $Z_p$. Indeed, with our convention that $J(\omega)$ vanishes for $\omega < 0$, equation (B.10) implies that $\omega_p$ must be negative. In our toy model, we are implicitly assuming that $\omega_0$ is positive so that we obtain a finite (Markovian) decay rate $\tau_R^{-1} = J(\omega_0)/2$. If this were not the case, the dynamics would be generally non-Markovian, as shown in the example of a photonic crystal (Sec. 4.5.2). Then, in order $\omega_0 + R(\omega_p)$ to be negative, the function $R(\omega)$ should attain values of the order of $\omega_0$. However, we are assuming that $K(t) \propto \lambda^2$, and hence $R(\omega)$, is perturbatively small, and in general we can expect that $|R(\omega)| \ll \omega_0$ for all frequencies $\omega$. Therefore, in the perturbative regime of weak coupling, $f(s)$ does not have any poles, and the solution $f(t)$ is determined by the branch cut contribution only[80].

Let us turn to the branch cut contribution, which reads

$$f(t) = f_0 \int_0^\infty \frac{d\omega}{2\pi} e^{-i\omega t} \frac{J(\omega)}{[\omega - \omega_0 - R(\omega)]^2 + [J(\omega)/2]^2} . \tag{B.11}$$

This integral is the Fourier transform of the spectral density multiplied by a "filter" function $\phi(\omega) = \{[\omega - \omega_0 - R(\omega)]^2 + [J(\omega)/2]^2\}^{-1}$. We will argue that in the weak-coupling regime $J(\omega) \ll \omega_0$, $\phi(\omega)$ selects a narrow window of frequencies around the Lamb-shifted value of the system frequency $\omega_0$, and that this "probing region" determines the Markovian part of the

---

[80]A case in which a bound state exists for all $\lambda \neq 0$ is when $J(\omega)$ has a discontinuous jump at $\omega = 0$, because then $R(\omega)$ diverges logarithmically there. Even in this pathological case, Markovian dynamics is protected because $|\omega_p|$ is exponentially small, $|R'(\omega_p)| \sim |\omega_p|^{-1} \gg 1$, and so the residue $Z_p$ is highly suppressed.

dynamics. This situation is depicted in Fig. 7b. We can determine the location of the maximum of $\phi(\omega)$ by minimizing $[\omega - \omega_0 - R(\omega)]^2 + [J(\omega)/2]^2$, obtaining

$$\omega_* = \omega_0 + R(\omega_*) - \frac{J(\omega_*)}{4} \frac{J'(\omega_*)}{1 - R'(\omega_*)} , \qquad (B.12)$$

where the primes stand for the first derivatives. We notice that terms on the right-hand side are of increasing perturbative order: $R(\omega)$ is of order $\mathcal{O}(\lambda^2)$, while the third term is of order $\mathcal{O}(\lambda^4)$. Then, in the weak-coupling regime that we are interested in, the location of the maximum is approximately $\omega_* = \omega_0 + R(\omega_0)$ (to order $\mathcal{O}(\lambda^4)$), which is the frequency of the Markovian solution (29) once we recognize $R(\omega_0)$ as the Lamb shift.

Close to the maximum[81] $\omega = \omega_*$, the filter function can be approximated by a Lorentzian function $Z_*^2/[(\omega - \omega_*)^2 + Z_*^2 J_*^2/4]$, where $J_* \equiv J(\omega_*)$, and $Z_* \equiv [1 - R'(\omega_*)]^{-1}$. On the scale of the bandwidth of $J(\omega)$, this Lorentzian function is extremely narrow since its width $Z_* J_*/2$ is perturbatively small. (We can reverse the argument: we assume that $J(\omega)$ has no feature on a scale smaller than $Z_* J_*/2$ close to $\omega_*$). Then, we can estimate equation (B.11) by

$$f(t) \approx f_0 \int_{-\infty}^{\infty} \frac{d\omega}{2\pi} e^{-i\omega t} \frac{Z_*^2 J_*}{(\omega - \omega_*)^2 + (Z_* J_*/2)^2} = Z_* f_0 e^{-i\omega_* t - Z_* J_* t/2} , \qquad (B.13)$$

in which we recognize the Markovian solution (29) with decay rate $\gamma_* = Z_* J(\omega_*)/2 = J(\omega_0)/2 + \mathcal{O}(\lambda^4) = \tau_R^{-1} + \mathcal{O}(\lambda^4)$. We see that the above solution coincides with the one provided by the Markovian approximation as long as $Z_* \approx 1$, $\omega_* \approx \omega_0 + R(\omega_0)$ and $\gamma_* \approx J(\omega_0)/2$. From the definitions of $\omega_*$ and $Z_*$, we see that these three conditions are well satisfied if $|R'(\omega_0)| \ll 1$ and $|J'(\omega_0)| \ll 1$, which imply that $\lambda^2 \ll 1$ and that $J(\omega)$ should not vary steeply around[82] $\omega_0$. In this sense, the smallness of $J'(\omega)$ around $\omega_0$ controls the non-Markovian corrections to $f(t)$ in the Markovian regime in which equation (B.13) is valid, as we claimed in the main text.

The approximation (B.13) cannot be valid for all times. It is not valid at very early times since it predicts $f(0) = Z_* f_0 \neq f_0$. Indeed, we know that for times $t \ll \tau_B$ $|f(t)|$ has a parabolic behavior. However, it cannot be valid for very late times, either, since the asymptotic properties of Fourier transforms depend crucially on the boundaries of integration. Indeed, by repeated integration by parts, one can prove the useful result that, for a smooth function $q(\omega)$,

$$\int_a^{\infty} \frac{d\omega}{2\pi} e^{-i\omega t} q(\omega) \sim e^{-iat} \frac{1}{2\pi} \sum_{n \geq 0} \frac{q^{(n)}(a)}{(it)^{n+1}} \qquad (B.14)$$

for $t \to \infty$. The above equation is valid under rather mild conditions on $q(\omega)$, and its derivatives $q^{(n)}(\omega)$ [83, 84]. From the equation above, we can draw two important conclusions: first, the asymptotic behavior of Fourier transforms of smooth functions that vanish below some minimum frequency is algebraic (as per the Paley-Wiener theorem [79–81]), and second, that the amplitude of the various powers of $t^{-1}$ only depend on the properties of the function at its lower edge[83]. Let us apply equation (B.14) to the full solution (B.11) in the common case [85] $J(\omega) = 2\pi\lambda^2 \omega_c (\omega/\omega_c)^\alpha + \mathcal{O}(\omega^{\alpha+1})$ for $\omega \to 0^+$, where $\omega_c$ is a frequency scale of the order of the bandwidth and $\alpha$ is a positive integer[84]:

$$f_{\text{asymp}}(t) \sim \frac{f_0 \lambda^2 \omega_c^{1-\alpha}}{[\omega_0 + R(0)]^2} \frac{\alpha!}{(it)^{\alpha+1}} + \mathcal{O}(t^{-\alpha-2}) \qquad (B.15)$$

---

[81]We are assuming that there is only one maximum $\omega_*$. A multiplicity of maxima would probably rule out a Markovian solution.

[82]Indeed, if $J(0) = J(\infty) = 0$, $R'(\omega) = \mathcal{P}\int_0^\infty \frac{d\varepsilon}{2\pi} J'(\varepsilon)/(\omega - \varepsilon)$, so the amplitude of $J'(\omega)$ controls that of $R'(\omega)$.

[83]If the frequency integral had an upper limit $\omega = b$, there would be a similar contribution from the latter, depending on $q^{(n)}(b)$.

[84]This result holds even if $\alpha$ is not integer, with minor modifications.

The full solution $f(t)$ can be well approximated for most times by summing equation (B.13) with the asymptotic result:

$$f(t) \sim Z_* f_0 e^{-i\omega_* t - \gamma_* t} + \frac{f_0 \lambda^2 \omega_c^{1-\alpha}}{[\omega_0 + R(0)]^2} \frac{\alpha!}{(it)^{\alpha+1}} + \mathcal{O}(t^{-\alpha-2}) . \qquad (B.16)$$

We can now better understand the existence of a time window for Markovian dynamics. The main reason is that the asymptotic expression (B.15) is highly suppressed, not just because of the negative power of time it contains, but also because the coefficient that it has in front is perturbatively small ($f_{\text{asymp}}(t) \propto \lambda^2$) and further decreased by the filter function $\phi(0) = 1/(\omega_0 + R(0))^2 \approx 1/\omega_0^2$. Then, at times of the order of $\gamma_*^{-1} \approx \tau_R$ the first terms dominates, since $Z_* \approx 1$ for $\lambda^2 \ll 1$. Only when the exponential term has completely decayed does the algebraic term become dominant, and the dynamics become fully non-Markovian. The crossover time $t_{\text{NM}}$ at which this happens occurs when both terms become comparable in magnitude and are usually much bigger than the Markovian decay time $\tau_R$—so, essentially, $f(t)$ has already decayed to zero for all practical purposes. From equation (B.15), we can read out the conditions that would decrease $f_{\text{asymp}}(t)$, thus ensuring a wider window of Markovian evolution. These are the weak coupling condition $\lambda^2 \ll 1$, a large value of $\alpha$ (i.e., the presence of fewer low-frequency excitations in the bath), and a large value of $\omega_0$, i.e., a faster intrinsic dynamics of the system.

Until now we have implicitly assumed that $J(\omega)$ is smooth everywhere, except possibly at its lower edge $\omega = 0$. However, it can happen that it has some singular behavior, often induced by Fermi-Dirac or Bose-Einstein functions (see 4.4.4). In particular, equation (B.14) shows that any discontinuity in the $n$th derivative of $J(\omega)$ at a frequency $\omega_{\text{NM}}$ would yield a non-Markovian term decaying as $t^{-n-1}$ to the function $f(t)$. However, the presence of the filter function $\phi(\omega)$ helps to preserve the Markovian regime: as long as the singularity is far from $\omega_* \approx \omega_0$ these algebraic terms will have a very small prefactor $\phi(\omega_{\text{NM}})$, and the Markovian term (B.13) will be the leading one at intermediate times. In this sense, the filter function $\phi(\omega)$ defines the "probing region" mentioned in the main text. The behavior of $f(t)$ for most of the interesting part of the dynamics (i.e., when $f(t)$ is appreciably large) depends only on the spectral density around the renormalized frequency $\omega_*$ and is essentially blind to the features of $J(\omega)$ for frequencies that are further than a few times the (Markovian) decay rate $Z_* J_*/2 \approx \tau_R^{-1}$ from $\omega_*$. Vice versa, if the system frequency $\omega_0$ happens to be close to a sharp feature of $J(\omega)$, the latter will dominate the dynamics even for intermediate times since the filter function will amplify rather than suppress the effect of those features.

## C  Breakdown of Lindblad in the Kondo model *

In this appendix, we derive the Lindblad master equation for the Kondo model we commented on in 4.5.1. At a practical level, we are going to compute the correlation functions for the spin density in a noninteracting fermionic gas, and then we will use them to compute the $\Gamma_{\alpha\beta}(\Omega)$ coefficients (13) appearing in the master equation, from which we will extract the Markovian decay rate $\gamma_L(T)$. The analysis of the spectral density of the bath will reveal why the Lindblad prediction $\gamma_L(T \to 0) = 0$ cannot be trusted. Finally, we will derive the full Born equation (10) for the Kondo problem, and show that it still cannot reproduce the correct dynamics of the impurity spin, confirming that the Kondo model is an example of the breakdown of the Born approximation.

The Kondo model is described by the Hamiltonian [109, 112, 124]

$$H = \lambda H_I + H_B = J \boldsymbol{S} \cdot \boldsymbol{s}(\boldsymbol{0}) + \sum_{\boldsymbol{p},\sigma} (\varepsilon_{\boldsymbol{p}} - \mu) c_{\boldsymbol{p}\sigma}^\dagger c_{\boldsymbol{p}\sigma} , \qquad (C.1)$$

where $S$ is the spin of the impurity[85], which is coupled with interaction strength $\lambda = J > 0$ to the spin density $s(0)$ of the fermionic bath at the impurity position $x = 0$. The latter is defined as $s_\alpha(0) \equiv \frac{1}{2} \sum_{\sigma,\tau} (\sigma^\alpha)_{\sigma\tau} c^\dagger_{x=0,\sigma} c_{x=0,\tau}$, where $c_{x\sigma}$ is the annihilation operator of the bath fermions at position $x$ and spin projection $\sigma \in \{\uparrow, \downarrow\}$, and $\sigma^\alpha$ are the Pauli matrices ($\alpha \in \{x, y, z\}$). The fermionic bath is noninteracting, and its Hamiltonian can be diagonalized in momentum space in terms of the annihilation operators of momentum states $c_{p\sigma}$, as shown in the last term of equation (C.1). For concreteness, we assume that the fermions move in a three-dimensional lattice[86] with a large total volume $V$ with periodic (Born-von Kármán) boundary conditions, giving rise to a certain dispersion relation $\varepsilon_p$. We could equally assume that the fermions are not subject to any potential. In terms of the momentum modes that diagonalize $H_B$ we have $c_{x\sigma} = V^{-1/2} \sum_p e^{ipx} c_{p\sigma}$, hence $s_\alpha(0) = \frac{1}{2V} \sum_{p,k,\sigma,\tau} (\sigma^\alpha)_{\sigma\tau} c^\dagger_{p\sigma} c_{k\tau}$.

We aim to implement the derivation of the Lindblad equation described in the main text. The coupled operators $A_\alpha$, $B_\alpha$ are simply the spin components $S_\alpha$ and $s_\alpha(0)$, respectively. Since the impurity has no intrinsic dynamics ($H_S = 0$)[87], the $A_\alpha = S_\alpha$ are conserved in the interaction picture—namely, there is only one transition frequency $\Omega = 0$, and the $S_\alpha$ are the eigenoperators of $H_S = 0$. Next, we need to compute the bath correlation functions $G_{\alpha\beta}(t) = \text{Tr}[B_\alpha(t)B_\beta \rho_B(0)]$. We take the initial state of the bath to be a thermal state $\rho_B \propto e^{-\beta H_B}$ with chemical potential $\mu$ and temperature $T = (k_B \beta)^{-1}$. Since $H_B$ is quadratic, the state $\rho_B$ is Gaussian and completely defined by the two-point functions $\langle c^\dagger_{p\sigma} c_{p'\sigma'} \rangle = \delta_{p,p'} \delta_{\sigma,\sigma'} F(\varepsilon_p - \mu)$, where $F(\varepsilon) = (e^{\beta\varepsilon} + 1)^{-1}$ is the Fermi-Dirac distribution. All higher-order correlation functions of the fermions can be expressed in terms of the two-point functions by applying Wick's theorem [112, 124]. Then

$$
\begin{aligned}
G_{\alpha\beta}(t) &= \frac{J^2}{4} \sum_{\sigma\tau,\sigma'\tau'} (\sigma^\alpha)_{\sigma\tau}(\sigma_\beta)_{\sigma'\tau'} \langle c^\dagger_{0\sigma}(t) c_{0\tau}(t) c^\dagger_{0\sigma'}(0) c_{0\tau'}(0) \rangle \\
&= \frac{J^2}{4} \sum_{\sigma\tau,\sigma'\tau'} (\sigma^\alpha)_{\sigma\tau}(\sigma_\beta)_{\sigma'\tau'} [\, \langle c^\dagger_{0\sigma}(t) c_{0\tau}(t) \rangle \, \langle c^\dagger_{0\sigma'}(0) c_{0\tau'}(0) \rangle \\
&\quad + \langle c^\dagger_{0\sigma}(t) c_{0\tau'}(0) \rangle \, \langle c_{0\tau}(t) c^\dagger_{0\sigma'}(0) \rangle\,] \\
&= \langle B_\alpha \rangle \, \langle B_\beta \rangle + \frac{J^2}{4\hbar^2} \sum_{\sigma\tau} (\sigma^\alpha)_{\sigma\tau}(\sigma^\beta)_{\sigma\tau} g^<_\sigma(-t) g^>_\tau(t) \, ,
\end{aligned}
\tag{C.2}
$$

where we used the notation $c_{0\sigma}(t) \equiv e^{iH_B t/\hbar} c_{x=0,\sigma} e^{-iH_B t/\hbar}$ for the fermion operators in the interaction picture, and we introduced the so-called local Green's functions

$$
\begin{aligned}
g^>_\sigma(t) &\equiv -i\hbar \langle c_{0\sigma}(t) c^\dagger_{0\sigma}(0) \rangle = -\frac{i\hbar}{V} \sum_p e^{-i\varepsilon_p t/\hbar} [1 - F(\varepsilon_p - \mu)] \, , \\
g^<_\sigma(t) &\equiv i\hbar \langle c^\dagger_{0\sigma}(0) c_{0\sigma}(t) \rangle = \frac{i\hbar}{V} \sum_p e^{-i\varepsilon_p t/\hbar} F(\varepsilon_p - \mu) \, .
\end{aligned}
\tag{C.3}
$$

We recall that $c_{p\sigma}(t) = e^{-i\varepsilon_p t/\hbar} c_{p\sigma}$ under $H_B$, and we have used the property that $\rho_B$ is a stationary state of the unperturbed bath, so that $\langle B_\alpha(t) \rangle = \langle B_\alpha \rangle$ and $g^<_\sigma(-t) \equiv i\hbar \langle c^\dagger_{0\sigma}(t) c_{0\sigma}(0) \rangle$. The expectation value of the $B_\alpha = s_\alpha(0)$ operators read

$$
\langle B_\alpha \rangle = -\frac{i}{2\hbar} \sum_\sigma (\sigma^\alpha)_{\sigma\sigma} g^<_\sigma(0) = 0 \, ,
\tag{C.4}
$$

---

[85]We are employing spin operators expressed in units of $\hbar$, so that $J$ has the dimensions of energy times volume. The physical spins are $\hbar S$ and $\hbar s(0)$ (the latter being a spin density per volume).

[86]Any simple lattice with one site per unit cell will suffice, since we do not need to consider the effects of multiple bands.

[87]Including a magnetic field is slightly more involved, and does not change the overall conclusion.

since in the absence of a magnetic field the Green's functions do not depend on the spin projection—in other words, each momentum state has equal ↑ and ↓ spin occupation. For the same reason, the bath correlation function $G_{\alpha\beta}(t)$ is diagonal in $\alpha, \beta$:

$$G_{\alpha\beta}(t) = \frac{J^2}{4\hbar^2} \text{Tr}(\sigma^\alpha \sigma^\beta) g^<(-t) g^>(t) = \delta_{\alpha\beta} \frac{J^2}{2\hbar^2} g^<(-t) g^>(t) \equiv \delta_{\alpha\beta} G(t) , \quad \text{(C.5)}$$

where we used $\text{Tr}(\sigma^\alpha \sigma^\beta) = 2\delta_{\alpha\beta}$ and omitted the $\sigma$ in the Green's functions to stress that they do not depend on spin. To have an intuition of the behavior of $G(t)$, we need to specify the band dispersion $\varepsilon_{\bm{p}}$. This is most easily done by introducing the appropriate density of single-fermion states[88]

$$\rho(\varepsilon) \equiv \frac{1}{V} \sum_{\bm{p}} \delta(\hbar\varepsilon - \varepsilon_{\bm{p}}) , \quad \text{(C.6)}$$

so that

$$g^>(t) = -i\hbar \int d(\hbar\varepsilon) \, e^{-i\varepsilon t} \rho(\varepsilon)[1 - F(\hbar\varepsilon - \mu)] ,$$
$$g^<(t) = i\hbar \int d(\hbar\varepsilon) \, e^{-i\varepsilon t} \rho(\varepsilon) F(\hbar\varepsilon - \mu) . \quad \text{(C.7)}$$

These expressions are completely analogous to the ones considered in 4.4.4 for the spin$-1/2$ bath, and indeed, the same arguments about the behavior of the Green's functions apply here. The spectral density $\rho(\varepsilon)$ is nonzero only within a certain frequency bandwidth $|\varepsilon| < W$ (we can always choose the zero of the energy to fall in the middle of the band), and it is generally some smoothly varying function. In most metals, the chemical potential and the bandwidth correspond to temperatures of the order of $10^4$ K [112, 124], so that even at room temperature they can be considered effectively at low temperature—namely, there is a sharply defined Fermi surface at $\varepsilon_{\bm{p}} = \mu$ dividing occupied states at $\varepsilon_{\bm{p}} \lesssim \mu$ from empty ones at $\varepsilon_{\bm{p}} \gtrsim \mu$, the transition occurring in a layer of thickness $\sim k_B T$. Since the scattering of electrons off the impurity spin described by the Kondo Hamiltonian (C.1) can only occur if the initial momentum states are occupied and the final ones are empty, low-energy scattering events are confined to the vicinity of the Fermi surface. In this regime, the behavior of the bath correlation functions (and, therefore, of the impurity spin) is determined by the Fermi-Dirac functions rather than the spectral function.

Substituting Eqs. (C.7) into equation (C.5) and changing the variables appropriately, we can obtain the bath spectral density

$$J(\omega) = \pi J^2 \hbar \int d(\hbar\varepsilon) \, \rho(\varepsilon) \rho(\varepsilon + \omega) F(\hbar\varepsilon - \mu)[1 - F(\hbar(\varepsilon + \omega) - \mu)] , \quad \text{(C.8)}$$

with $G(t) = \int \frac{d\omega}{2\pi} e^{-i\omega t} J(\omega)$. As we noticed in 4.3, $J(\omega)$ measures the amount of excitations available to the system at the frequency $\omega$. Indeed, the above expression can be interpreted as counting all possible transitions from occupied states at energy $\hbar\varepsilon$ (the factor $\rho(\varepsilon) F(\hbar\varepsilon - \mu)$) to unoccupied states at energy $\hbar(\varepsilon + \omega)$ (factor $\rho(\varepsilon + \omega) F(\hbar(\varepsilon + \omega) - \mu)$). At low temperature, the exclusion principle constrains the frequency to lie in the range $\mu/\hbar - \omega \lesssim \varepsilon \lesssim \mu/\hbar$. At zero temperature, the inequality becomes strict, thus implying that $J(\omega)$ vanishes for $\omega \leq 0$, in accordance with the general theory. At low frequency and temperature $|\omega| \ll k_B T/\hbar \ll W$, the spectral density can be approximated by simply neglecting the variation of the density of states within the relevant interval $\mu/\hbar - \omega \lesssim \varepsilon \lesssim \mu/\hbar$, obtaining

$$J(\omega) \approx \pi\hbar(J\rho_F)^2 \int_{-\infty}^{\infty} d\varepsilon \, F(\hbar\varepsilon - \mu)[1 - F(\hbar(\varepsilon + \omega) - \mu)] = \pi(J\rho_F)^2[1 + B(\hbar\omega)]\hbar\omega , \quad \text{(C.9)}$$

---

[88]Notice that for mathematical convenience, the variable $\varepsilon$ in the following formulas has the dimensions of a frequency rather than an energy.

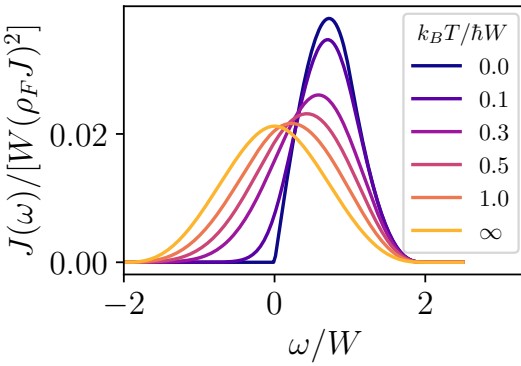

Figure 8: Spectral function equation (C.8) computed numerically for the spectral function $\rho(\varepsilon) = \rho_F(1 - \varepsilon^2/W^2)\theta(W - |\varepsilon|)$ for increasing temperatures. The spectral density displays a bosonic (Ohmic) character (compare with Fig. 3b) at low temperature while revealing a more fermionic character at very high temperature (compare with Fig. 3a).

where we have introduced the density of states at the chemical potential $\rho_F \equiv \rho(\mu/\hbar)$ and the Bose-Einstein distribution $B(\varepsilon) = (e^{\beta\varepsilon} - 1)^{-1}$. The above expression contains three interesting points. First, the spectral density depends on the Kondo coupling only through the dimensionless product $J\rho_F$. Hence, the latter is the natural small parameter of the theory. Second, the above spectral density is bosonic in nature. Indeed, the Kondo interaction does not change the number of fermions in the bath, but rather moves fermions across the Fermi surface. The resulting particle-hole excitations are effectively bosonic, since they are formed out of a pair of fermions. This analogy can be further formalized into a mapping of the Kondo model to the spin-boson model [85]. Third, the spectral density is Ohmic, i.e., linear in frequency, for $\omega \to 0$. This is a consequence of the fact that the spectral function is nonzero around the Fermi surface, $\rho_F \neq 0$, so that there is an abundance of low-energy particle-hole pairs. In turn, this slow decrease of $J(\omega)$ for $\omega \to 0$ is reflected in the algebraic decay of $G(t) \sim t^{-2}$ at late times, as it can be verified by taking the previous formula and applying equation (B.14). The full behavior of $J(\omega)$ (computed numerically) is shown in Fig. 8 for increasing temperatures. A comparison with Fig. 3 shows that the low-temperature behavior of $J(\omega)$ is indeed analogous to that of an Ohmic bosonic bath, while at high temperature $k_B T \gtrsim \hbar W$ it becomes more fermionic in nature, in the sense that the Fermi-Dirac function becomes flat and the shape of $J(\omega)$ is fully dictated by $\rho(\varepsilon)$.

We can now continue with the implementation of the Lindblad formalism. The $\Gamma_{\alpha\beta}(\Omega)$ matrix (equation (13)) is diagonal, $\Gamma_{\alpha\beta}(\Omega) = \delta_{\alpha\beta}\Gamma(\Omega)$, and needs to be evaluated at the only transition frequency $\Omega = 0^+$. The Lamb shift Hamiltonian amounts to a mere constant shift of the energy, $H_{LS} = \hbar \operatorname{Im}\Gamma(0^+)\sum_\alpha S_\alpha^2 \propto \mathbb{1}$, since $S_\alpha^2 = \mathbb{1}/4$. The decay rates are independent of the direction $\alpha$, and are given by

$$\gamma_L(T) = 2\operatorname{Re}\Gamma(0^+) = \frac{1}{\hbar^2}J(\omega = 0) = \pi J^2 \int d\varepsilon \, \frac{\rho^2(\varepsilon)}{4\cosh^2[\beta(\hbar\varepsilon - \mu)/2]} \; . \tag{C.10}$$

Finally, we obtain the Lindblad master equation for the Kondo model,

$$\frac{d}{dt}\rho_S(t) = \gamma_L(T)\sum_\alpha \left(S_\alpha \rho_S S_\alpha - \frac{1}{4}\rho_S\right) = -\gamma_L(T)\left(\rho_S - \frac{\mathbb{1}}{2}\right), \tag{C.11}$$

where the second equality has been obtained by using the general decomposition $\rho_S = \mathbb{1}/2 + \sum_\alpha \operatorname{Tr}(\rho_S\sigma^\alpha)\sigma^\alpha/2$ and the property $\sum_\alpha \sigma^\alpha\sigma^\beta\sigma^\alpha = -\sigma^\beta$. The solution of the above equation

is

$$\rho_S(t) = \frac{\mathbb{1}}{2} + e^{-\gamma_L(T)t}\left[\rho_S(0) - \frac{\mathbb{1}}{2}\right]. \tag{C.12}$$

This equation describes an exponential relaxation of the spin towards the maximally mixed state $\mathbb{1}/2$, with both the magnetization $\langle S_z(t)\rangle \propto (\rho_S)_{\uparrow\uparrow}(t) - (\rho_S)_{\downarrow\downarrow}(t)$ and the coherences $(\rho_S)_{\uparrow\downarrow}(t) = [(\rho_S)_{\downarrow\uparrow}(t)]^*$ decaying exponentially with the same rate, $\gamma_L(T)$.

As we have seen before, $\gamma_L(T = 0) \propto J(\omega = 0)$ vanishes at zero temperature, so we recover the result that we quoted in the main text—the Lindblad treatment of the Kondo model predicts that the impurity spin does not relax at absolute zero, in contrast with the exact solution and with experimental data [109]. We recover relaxation only at finite temperature. As long as $k_B T \ll \mu$, the $1/\cosh^2$ factor in the integral is strongly peaked around $\hbar\varepsilon = \mu$, so that it is sufficient to expand $\rho(\varepsilon)$ around the chemical potential:

$$\gamma_L(T) \sim \pi J^2 \int_{-\infty}^{\infty} d\varepsilon \, \frac{\rho_F^2 + 2\rho_F(\varepsilon - \mu/\hbar) + \mathcal{O}\big((\varepsilon - \mu/\hbar)^2\big)}{4\cosh^2[\beta(\hbar\varepsilon - \mu)/2]} = \pi(J\rho_F)^2 \frac{k_B T}{\hbar} + \mathcal{O}(T^3), \tag{C.13}$$

where we used $\int_{-\infty}^{\infty} dx / [4\cosh^2(x/2)] = -\int_{-\infty}^{\infty} dx \, dF(x)/dx = 1$. The above expression, known as the Korringa rate [109], is the one quoted in the main text.

We wish to highlight that the Lindblad treatment of the Kondo model is not entirely worthless. In fact, the linear dependence of the spin relaxation rate on the temperature is what is observed in experiments on dilute magnetic alloys (e.g., see chapter 9.5 of [109]) for sufficiently high temperatures (with respect to the Kondo temperature $T_K$). The predicted stationary state is also the correct Gibbs state $\propto e^{-\beta H_S} = \mathbb{1}$ for a spin without a magnetic field. The crucial point is that the Lindblad approach fails in the low temperature regime $T \ll T_K$, when entanglement between the spin and its bath becomes relevant. Indeed, in section 3.1 we have argued that, to the leading order in the coupling $\lambda$, the dynamics of $\rho_S$ is dictated by the separable part of the total state $\rho(t)$, $\rho_S(t) \otimes \rho_B$, with the nonseparable part being a higher-order perturbation—this is the essence of the Born approximation. However, in the present case of the Kondo model, the separable part leads to no dynamics at all at zero temperature. Hence, the dynamics at $T = 0$ is entirely determined by the nonseparable terms in $\rho$, beyond the Born approximation.

## C.1  Failure of Born approximation in the Kondo model

Despite the previous qualitative argument about the role of non-separable contributions of the system-bath state to the dynamics of the spin, one might still wonder whether the unphysical absence of dynamics at zero temperature might actually be caused by the Markovian approximation, rather than the Born one. After all, the bath correlation function has a rather slow algebraic decay $G(t) \sim t^{-2}$ in time. We will now show that including non-Markovian effects in the Born approximation leads to a marginal improvement: the impurity spin relaxes even at zero temperature, but the decay is not exponential, in contrast with the known dynamics.

Let us take the Born master equation (7) in the present model, with $A_\alpha(t) = S_\alpha$ and $\langle B_\alpha(t)B_\beta(\bar{t})\rangle = \delta_{\alpha\beta} G(t - \bar{t})$:

$$\begin{aligned}
\frac{d}{dt}\rho_S(t) &= \int_0^t d\bar{t} \, 2\,\mathrm{Re}\, G(t - \bar{t}) \sum_\alpha \left(S_\alpha \rho_S(\bar{t}) S_\alpha - \frac{1}{4}\rho_S(\bar{t})\right) \\
&= -\int_0^t d\bar{t} \, 2\,\mathrm{Re}\, G(t - \bar{t})\left(\rho_S(\bar{t}) - \frac{\mathbb{1}}{2}\right).
\end{aligned} \tag{C.14}$$

Since $H_S = 0$, there is no difference between the Schrödinger and the interaction pictures for the impurity. The maximally mixed state $\rho_S^\infty = \mathbb{1}/2$ is still a stationary solution, and our task

is to understand if the dynamics will bring the state towards it even at $T = 0$. Taking matrix elements of the above equation, we recognize the familiar form of the toy model (19),

$$\frac{\mathrm{d}}{\mathrm{d}t} f(t) = -\int_0^t \mathrm{d}\bar{t}\, 2\,\mathrm{Re}\, G(t - \bar{t}) f(\bar{t})\,, \tag{C.15}$$

with $\omega_0$, integral kernel $K(t) = -2\,\mathrm{Re}\, G(t)$ and $f(t) = [\rho_S(t)]_{\sigma\tau} - \delta_{\sigma\tau}/2$ being either a coherence $\rho_{\sigma\bar{\sigma}}(t)$ or the distance between population $\rho_{\sigma\sigma}(t)$ and the stationary population, $[\rho_S^\infty]_{\sigma\sigma} = 1/2$. Then, we can apply the same machinery as in appendix B. The fundamental object that we need to compute is the spectral density associated with the kernel $-2\,\mathrm{Re}\, G(t)$, which reads $K(\omega) = J(\omega) + J(-\omega)$ (we are going to denote it as $K(\omega)$, since we already use $J(\omega)$ for the spectral density of $G(t)$). The other function that we need is the Hilbert transform [83] of $K(\omega)$, $R(\omega) \equiv \mathcal{P} \int \frac{\mathrm{d}\varepsilon}{2\pi} \frac{K(\varepsilon)}{\omega - \varepsilon}$. Let us notice that, if the spectral function $\rho(\omega)$ is nonzero for $|\omega| < W$, then $J(\omega)$ and $K(\omega)$ are finite for $|\omega| < 2W$—in other words, the lower band edge is at $\omega = -2W$ rather than 0. Following (B.5) and (B.11), the solution to the differential equation (C.15) can be written as a Fourier transform,

$$\begin{aligned}
f(t) &= f(t = 0) \int_{-\infty}^{\infty} \frac{\mathrm{d}\omega}{2\pi} e^{-i\omega t} \frac{K(\omega)}{[\omega - R(\omega)]^2 + [K(\omega)/2]^2} \\
&= f(t = 0)\, 2\,\mathrm{Re} \int_0^{\infty} \frac{\mathrm{d}\omega}{2\pi} e^{-i\omega t} \frac{K(\omega)}{[\omega - R(\omega)]^2 + [K(\omega)/2]^2} \\
&\equiv f(t = 0)\, 2\,\mathrm{Re} \int_0^{\infty} \frac{\mathrm{d}\omega}{2\pi} e^{-i\omega t} K(\omega) \phi(\omega)\,,
\end{aligned} \tag{C.16}$$

where in the second equality we have exploited the even symmetry of the integrand function and we have introduced the filter function $\phi(\omega)$. As mentioned in B, the filter function selects the "probing region" of $K(\omega)$ that dominates the dynamics at intermediate times.

Let us focus on the case of zero temperature. From the discussion in B, we expect the probing region to be around $\omega_0 + R(\omega_0)$ with width $\sim K(\omega_0)$. In the present case, $\omega_0 = 0$ and $R(\omega_0) = 0$ (it is an odd function, because $K(\omega)$ is even), and $K(\omega_0) = 0$ as well[89]. These conditions already hint that the resulting dynamics cannot be Markovian, since we need at least a nonvanishing spectral function in the probing region and, moreover, $\omega = 0$ is a band edge ($\omega = 0$ is the band edge for the particle-hole excitations quantified by $J(\omega)$). In the words of section 4.4, there cannot be a Markovian time window, because $f(t)$ will be dominated by the band edge at all times—the dynamics is always in the non-Markovian regime, which usually appears only at late times. Further analysis reveals that the filter function selects the "probing window" at $\omega = 0$ in a highly singular manner. Indeed, since $J(\omega) \sim \omega \theta(\omega)$ at low frequencies, we have $K(\omega) \sim |\omega|$. The non-analytic behavior of $K(\omega)$ at $\omega = 0$ is reflected in a singular behavior of $R(\omega)$, too. With a shift of integration variables, we can rewrite $R(\omega)$ as

$$R(\omega) = \int_0^{\infty} \frac{\mathrm{d}\varepsilon}{2\pi} \frac{K(\omega - \varepsilon) - K(\omega + \varepsilon)}{\varepsilon}\,, \tag{C.17}$$

and, taking a small but finite $|\omega|$, we can extract its most singular contribution by focusing on a neighborhood of the origin, $\varepsilon \in ]0, \Lambda[$ (where $\Lambda > |\omega|$ is a cutoff much smaller than $W$), in which we can approximate $K(\omega) \sim \eta|\omega|$:

$$R(\omega) \sim \int_0^{\Lambda} \frac{\mathrm{d}\varepsilon}{2\pi} \eta \frac{|\omega - \varepsilon| - |\omega + \varepsilon|}{\varepsilon} = \frac{\eta}{2\pi}\left(\omega \ln\frac{|\omega|}{\Lambda} + \mathcal{O}(\omega)\right). \tag{C.18}$$

---

[89]The attentive reader will have noticed that these are the conditions (B.10) for a pole of the Laplace transform of $f(t)$. However, we will see that $R(\omega) \sim \omega \ln|\omega|$ at $\omega \to 0$, so the residue of the pole $Z_p = [1 - R'(0)]^{-1}$ vanishes and does not contribute to $f(t)$ (compare equation (B.5)). Poles above and below the band might appear, $|\omega_p| \gtrsim 2W$, but these occur only if the coupling $J\rho_F$ is large enough. We assume that this is not the case.

Therefore, at small frequencies $\phi(\omega) \sim (\omega \ln \omega)^{-2}$, and the resulting behavior of $K(\omega)\phi(\omega)$ is diverging at small frequencies as $1/(\omega(\ln \omega)^2)$, which is quite far from the Lorentzian profile $\propto 1/(\omega^2 + \gamma^2)$ that is necessary for the Markovian approximation to hold. The asymptotics of Fourier transforms of functions possessing such logarithmic singularities generally involve an expansion in powers of $(\ln t)^{-1}$ ( [84], chapter II.2), hinting at an extremely slow decay of $f(t)$. However, we emphasize that $f(t)$ does decay to 0, because it is a Fourier transform of an integrable function (by the Riemann-Lebesgue lemma [82]). Thus, avoiding the Markovian approximation does improve the master equation, because it guarantees that $\rho_S(t)$ converges to the correct steady state, $\mathbb{1}/2$, even at zero temperature. Indeed, the ground state of the Kondo model is known to be a singlet $|\text{gs}\rangle = (|\uparrow\rangle_S |\varphi_\downarrow\rangle_B - |\downarrow\rangle_S |\varphi_\uparrow\rangle_B)/\sqrt{2}$ [109,112,124], where $|\varphi_\uparrow\rangle_B$ ($|\varphi_\downarrow\rangle_B$) is an appropriate state of the bath fermions having total spin $1/2$ ($-1/2$). Then, the *exact* stationary density matrix of the spin at zero temperature is the maximally mixed state, $\rho_S = \text{Tr}_B |\text{gs}\rangle\langle\text{gs}| = \mathbb{1}/2$. Whether this is just a coincidence cannot be answered within the present approach. Notwithstanding the improvement over the Lindblad approach at $T = 0$ with regards to the equilibration to the correct stationary state, the Born master equation still yields an incorrect dynamics with a non-exponential decay. We have verified this statement by direct numerical integration of equation (C.15).

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
