# Peer review of "Is Lindblad for me?"

_SciPost Physics Reviews_

## Round 1 · Referee Report · Anonymous (Referee 1) · 2025-11-14

Report

This paper presents a detailed lecture on intricacies and technicalities associated with Lindblad quantum master equation (QME), a standard tool for calculating dynamics of open systems. Specifically, the authors delve into Born, Markov and Rotating Wave Approximations that go into derivation in order to obtain an autonomous differential equation for the density operator. Besides doing a comprehensive detailing and discussion of standard calculations, there are a couple of technical examples where authors offer some new and several useful insights. With regards to new insights, I particularly took note of Sec. 4.4.3 on the effect of bath temperature studied in the context of central qubit coupled to a Bosonic/Fermionic bath, and Sec. 4.5.1 on the effect of system-bath entanglement in the context of a spin-Fermion (Kondo) model, and found both of them quite instructive for the intended reader.

Overall, I find the exposition clear and interesting. Given the resurgence of interest in revisiting the standard quantum optics descriptions, triggered by the growing need for tools to describe dynamics of noisy quantum systems, this is a timely and useful exposition for the quantum community including both theorists and experimentalists. Nonetheless, there are some technical issues that should be ironed out in order to make the discussion clearer and self-consistent, even if not exhaustive (I realize it is hard for authors to do cover all developments even after deciding to focus on a specific and limited description of open quantum systems, such as Lindblad QME.)

I detail below my questions and suggestions to address some of these key concerns. A lot of them should hopefully help towards the very task authors set out for themselves -- i.e. clarifying distinctions between and technicalities associated with Born and Markov approximations:

  1. One of my primary concerns is the small parameter authors quote first on Sec. 2/page 6. Here the authors comment on the validity of Born and Markov approximations, invoking comparisons of the system relaxation time $\tau_{R}$ (essentially the time scale that will be calculated from the prefactor of the dissipator in the Lindblad QME) being longer than bath time scale $\tau_{B}$ (for Born or weak coupling) and system time scale $\tau_{B}$ (for Markov) respectively. Later on page 12, in the discussion following Eq. (10), they discuss perturbation theory validity and quote how this is governed by the calculated dissipation rate to be perturbative on the scale set by ${\rm min}{\tau_{B}, \tau_{S}}$.

Besides finding this separation of validity of perturbation theory and weak coupling artificial, especially since the authors are always considering microscopic description of the universe (system + bath) to be unitary (no decays to begin with!), I am not convinced this is entirely justified. In Hamiltonian perturbation theory, the perturbative parameter is not set by minimum of dynamics scales of the system/bath but rather the difference i.e. $\lambda < 1/|\tau_{S} - \tau_{B}|$ which for unitary case boils down to system-bath detunings. Thus a more nuanced discussion of this point which is consistent across Sections 2 and 3 is needed.

  1. A related issue also makes an appearance on page 42, Eq. (52) where authors do mention the explicit perturbative limit on the system to determine the validity of RWA. This is nothing but the limit discussed for motivating Markov approximation earlier and highlights the insidious connection between how Markovianity/RWA implies weak coupling but not the other way around. If authors agree, they should make these clarifications clear by including an explicit time scale hierarchy in row 5 of the very useful table presented on page 50 -- and explicitly state the connections and distinctions between Markovian treatment and RWA. Some recent works, such as https://arxiv.org/pdf/2403.18907 that have explicitly made such comparisons for linear and nonlinear system-bath interactions, could also be worth including as citations.

  2. In the same vein, for the discussion on page 6 regarding self-consistency of Born approximation, it will probably be useful for the authors to specify -- maybe in a footnote -- that consistency of neglecting terms second-order corrections to the state, post Born, is a result of interaction being normal-ordered (the so-called `centered bath' approximation for linear interactions such as Jaynes-Cummings and spin-Boson models) which prevents generation of spurious higher-order moments.

A related point pertains to applicability of footnote 26 on page 18 where authors make a statement about demanding all higher-order correlations generated via Wick's contraction to be Delta-correlated. This is a tricky point and does not automatically follow since, for nonlinear interactions, normal ordering the interaction is not equivalent to assuming a 'centered' bath.

  1. Page 15, footnote 19: this is a fairly important and often overlooked point. While it is nice that authors are including this discussion, this is not really limited to the issue of thermalization. In fact, it is worth specifying that coherence or off-diagonal elements vs populations are predicted to different orders in $\lambda$ using QMEs. I would encourage the authors to expand this discussion and, if needed, elevate it to main text.

  2. At the end of Sec. 4.2.3 on thermalization, where authors makes an important distinction between calculating the jump operator including or excluding the off-diagonal unitary interaction. I am not sure I fully understand the assertion: are authors saying that the linewidth inherited by the qubit transition should \emph{not} be included in the calculation? What is the issue with having $\gamma^{2}$ in the denominator? Furthermore, as they explained earlier in the paper, validity of Markovian approximation necessitates $\tau_{R}^{-1}\ll \tau_{H}^{-1}$ which boils down to their desired condition of $\gamma < \Delta$. Please clarify.

  3. Sec 4.2.4, many-body systems: I got a little confused by the title and the relevance of it to ensuing discussion. The point regarding nonlocal jump operators is not confined to many-body systems by the conventional definition of this phrase. In fact, the problem of a bipartite system $H_{S} = H_{A} + H_{B}$ with individual reservoirs is the archetypal setting for several reservoir-engineered schemes, cavity/circuit-QED platforms, two-qubit interactions etc. The issue of global vs local master equations has been well studied as noted in references cited by the author and several missing ones [e.g. Cattaneo et al., New J Phys. 21, 113045 (2019)]. I recommend changing the name to local vs global QMEs unless authors want the reader to construe anything bigger than a single system degree of freedom (including bi-/two-) as "many-".

  4. Under the cavities subsection of Sec. 4.5.2, since the authors are now alluding to a lossy cavity, in the context of Rabi oscillations and also for making the connection with earlier discussion of Born/weak coupling approximation more apparent for "not-so-bad"(!) cavity, it is worth mentioning that now the relevant bath time scale $\tau_{B}$ determining weak coupling is the photon decay rate (say $\kappa_{B}$) and not some unitary frequency scale associated with the cavity. Also, the phrase 'bad cavity' should be clarified in the context of the article. In terms of the symbols introduced in this paper, this should correspond to $\Omega < \kappa_{B}$, where $\Omega$ denotes system transition frequencies.

  5. It will be worthwhile to revisit or move the brief discussion of singular coupling limit under Sec. 5, either in the context of Figure 5 or later as a separate subsection. This is because it is a special limit to obtain Lindblad form without making an RWA approximation in nearly degenerate settings by imposing exact degeneracy. A more detailed discussion of entailing issues may have been useful to the reader too, but I leave this to authors' discretion.

  6. At the end of Sec. 5.3.1, authors make an important point about the suppression of tunneling and qubit transition frequency crashing to zero. If I understand correctly, the effect authors have in mind maps to ultra-strong coupling regime when $|g_\lambda| \approx \Delta$. If yes, this provides a clear(er) reason as to why RWA approximation cannot be made -- as dressing due to two-photon terms cannot be neglected in the calculation of Lamb shifts or rates, and should be mentioned in the text.

  7. Minor text change: In the line above Eq. (19), it will be better to say ``we will focus on a toy model that corresponds to the simplest form of Eq. (18) with a scalar memory Kernel", instead of the current sentence.

Recommendation

Ask for major revision

---

## Editorial Decision

awaiting_resubmission